# GWAS of random glucose in 476,326 individuals provide insights into diabetes pathophysiology, complications and treatment stratification

Conventional measurements of fasting and postprandial blood glucose levels investigated in genome-wide association studies (GWAS) cannot capture the effects of DNA variability on 'around the clock' glucoregulatory processes. Here we show that GWAS meta-analysis of glucose measurements under nonstandardized conditions (random glucose (RG)) in 476,326 individuals of diverse ancestries and without diabetes enables locus discovery and innovative pathophysiological observations. We discovered 120 RG loci represented by 150 distinct signals, including 13 with sex-dimorphic effects, two cross-ancestry and seven rare frequency signals. Of these, 44 loci are new for glycemic traits. Regulatory, glycosylation and metagenomic annotations highlight ileum and colon tissues, indicating an underappreciated role of the gastrointestinal tract in controlling blood glucose. Functional follow-up and molecular dynamics simulations of lower frequency coding variants in glucagon-like peptide-1 receptor (*GLP1R*), a type 2 diabetes treatment target, reveal that optimal selection of GLP-1R agonist therapy will benefit from tailored genetic stratification. We also provide evidence from Mendelian randomization that lung function is modulated by blood glucose and that pulmonary dysfunction is a diabetes complication. Our investigation yields new insights into the biology of glucose regulation, diabetes complications and pathways for treatment stratification.

Genetic factors are important determinants of glucose homeostasis and type 2 diabetes (T2D) susceptibility. Heritability of both fasting glucose (FG) and T2D is high, at 35–40%[1] and 30–60%[2], respectively. To date, more than 400 genetic loci have been associated with T2D[3,4]. Genome-wide association studies (GWAS) for glycemic traits in individuals without diabetes have identified genetic predictors of blood glucose, insulin and other metabolic responses during fasting or after oral or intravenous glucose challenge tests[5–8]. However, physiological glucose regulation involves responses to diverse nutritional and other stimuli that were, by design, omitted from such studies. Blood glucose is frequently measured at different times throughout the day in clinical practice and research studies (random glucose (RG)). While RG is inherently more variable than standardized measures, we reasoned that, across a very large number of individuals, it gives a more comprehensive representation of complex glucoregulatory processes occurring in different organ systems. Therefore, to identify and functionally validate genetic effects influencing RG, explore its relationships with other traits and diseases, and use these data to provide pathways for T2D treatment stratification, we performed a large-scale cross-ancestry GWAS meta-analysis for RG in individuals without diabetes.

✉e-mail: m.kaakinen@imperial.ac.uk; ben.jones@imperial.ac.uk; i.prokopenko@surrey.ac.uk

## Results

### RG GWAS expands the catalog of glycemia-related genetic associations

We undertook RG GWAS in 476,326 individuals without diabetes of European ($n = 459,772$) and other ancestries ($n = 16,554$) with adjustment for age, sex and time since last meal (where available), along with the exclusion of extreme hyperglycemia (RG > 20 mmol l$^{-1}$) and individuals with diabetes (Supplementary Table 1). The covariate selection was done upon extensive phenotype modeling (Supplementary Note, Supplementary Table 2 and Extended Data Fig. 1a). We identified 150 distinct signals ($P < 10^{-5}$) by fine mapping through conditional analysis within 120 loci reaching genome-wide significance ($P < 5.0 \times 10^{-8}$; Fig. 1a and Supplementary Tables 3 and 4). Fifty-three RG signals are reported for glycemic traits for the first time, greatly expanding our knowledge about the genetics of glycemia (Tables 1 and 2 and Supplementary Table 3). Adjustment for last meal timing (Extended Data Fig. 1b) did not change effect size estimates while enabling better power for the analysis. Application of glycated hemoglobin (HbA1c) cut point for diagnosing diabetes (HbA1c ≥ 6.5%) highlighted stronger associations at *G6PC2* and *GCK* lead RG loci (Extended Data Fig. 1c), suggesting their roles in glucose set-point in normoglycemia[9]. Neither adjustment for body mass index (BMI), nor a more stringent hyperglycemia cut-off (RG > 11.1 mmol l$^{-1}$; Extended Data Fig. 1d,e) materially changed the magnitude and significance of the RG effect estimates, although when all covariate models were individually applied, 11 additional signals at genome-wide significance were identified (Table 2 and Supplementary Table 5). Despite previous misconceptions that RG is of limited value for genetic discovery because of its inherent variability, our RG GWAS demonstrates that this trait variability has a clear genetic component.

A number of signals identified in individuals of European ancestry showed nominal significance ($P < 0.05$) in other ancestry groups, including new loci *MANSC4/KLHL42* in African, *FAM46C* and *ACVR1C* in Indian and *RBMS1* in Chinese ancestry groups (Supplementary Table 3). All such signals, except rs540524 at *G6PC2*, rs183606969 at *GCK* and rs6006399 at *MTMR3/HORMAD2*, were directionally concordant across ancestries. At *GCK*, rs2908286 ($r^2_{\text{1000GenomesAllAncestries}} = 0.83$ with rs2971670 lead in European ancestry individuals) was genome-wide significant in the African ancestry individuals alone (Supplementary Table 6). Cross-ancestry meta-analyses combining European and the other four ancestral groups revealed two new RG signals at *RRNAD1* and *PROX1* (Table 2 and Supplementary Table 6). Overall, while being only 16,554 individuals larger in sample size than the European ancestry meta-analysis, the cross-ancestry analysis expanded the new locus discovery for RG, confirming the potential of cross-ancestry studies for complex trait genetics.

The strongest associations with RG were detected at *G6PC2* ($P < 1.0 \times 10^{-746}$) and *GCK* ($P < 3.7 \times 10^{-277}$), established loci for FG and with key roles in gluconeogenesis[10] and glucose sensing[11], respectively (Supplementary Table 3). Notably, only two-thirds of RG signals overlapped with T2D-associated loci (Extended Data Fig. 1f), including three new loci for glycemia (*SCD5*, *RNF6* and *TSHZ2*). The direction of effects

at these loci between RG, T2D and homeostasis model assessment of β-cell function/insulin resistance (HOMA-B/HOMA-IR)[6] (Extended Data Figs. 1f,g and 2 and Supplementary Table 7) were consistent with their epidemiological correlation. We also discovered sex dimorphism at 13 RG loci, including male-specific *PRDM16* and *RSPO3*, and female-specific *SGIP1*, *SRRM3* and *SLC43A2* (Table 2, Fig. 1a and Supplementary Tables 3 and 8). We conclude that sex dimorphism, characterizing over one-tenth of RG-associated loci, is a widespread feature of glucose metabolism.

### Coding, rare and causal variants in RG variability

The lead variants at two new RG loci (*NMT1* and *RFX1*) and three previously reported loci for FG (*TET2*, *THADA* and *RREB1*) were all coding common (minor allele frequency (MAF) ≥ 5%) variants (Supplementary Table 3 and Extended Data Fig. 3). Additionally, lead RG-associated SNPs at glucagon-like peptide-1 receptor (*GLP1R*), neuronal differentiation 1 (*NEUROD1*) and ER degradation enhancing α-mannosidase like protein 3 (*EDEM3*) loci in our analysis were low-frequency (5% > MAF ≥ 1%) coding variants (Table 1, Supplementary Table 3 and Extended Data Fig. 3). *NEUROD1* and *EDEM3* are plausible candidates for glucose homeostasis, with the former reported for glucosuria[12] and the latter linked to renal function[13,14]. Within the rare allele frequency range (1% > MAF ≥ 0.001%), we first identified 30 RG loci and validated seven in whole-exome sequencing (WES) UK Biobank (UKBB) data (Supplementary Note). These included noncoding, such as rs2096313127 at *CAMK2B* (Supplementary Table 9) and synonymous rs2232324 in *G6PC2* variant associations (Table 2 and Supplementary Table 9). We expanded the annotation of coding nonsynonymous independent ($r^2_{\text{1000GenomesAllAncestries}} < 0.0010$) rare variant signals associated with RG to nondeleterious new rs146886108 (Arg187Gln) in *ANKH*[15], and deleterious, including three in *G6PC2* with predicted (rs2232326) and established (rs138726309, rs2232323)[16] effects (Supplementary Table 9). Thus, a range of coding and rare variants contributes to RG level variability and can be detected in very large genetic studies.

Next, we sought to pinpoint the most plausible set of causal variants by calculating 99% credible sets for each RG locus. In the European ancestry-only analysis, 15 RG signals were explained by one variant with a posterior probability of ≥99% of being causal, including low-frequency variants in *GLP1R*, *G6PC2*, *MECOM* and *CCND2* (ref. 17), and common variants in *LMO1* and *CACNA2D3* (Fig. 1b and Supplementary Table 10a). For another 16 signals, such as at *RMST*, *FOXN3* and *ADRA2A*, a lead variant had a posterior probability ≥80%. Credible sets at *WIPI1*, *GCKR*, *TET2*, *RREB1* and *RFX6* included coding common variants. *RREB1* and *RFX6* encode transcription factors implicated in the development and function of pancreatic β cells[18,19]. The credible sets were narrowed down for several signals in cross-ancestry RG meta-analysis (European ancestry median credible set size = 12.0 and cross-ancestry = 12.0), with improvements observed at *DGKB* and *TP53INP1* lead signals (Supplementary Table 10b,c). These analyses highlight examples of validated and potential targets for therapeutic development[15].

**Fig. 1 | Summary of all RG loci identified in this study. a**, Circular Manhattan plot summarizing findings from this study. In the outermost layer, gene names of the 133 distinct RG signals are labeled with different colors indicating the following three clusters defined in cluster analysis: 1a/1b, metabolic syndrome; 2a/2b, insulin release versus insulin action (with additional effects on inflammatory bowel disease for cluster 2a) and 3, defects of insulin secretion. Asterisks annotate RG signals that are new for glycemic traits. Track 1 shows RG Manhattan plot reporting −log$_{10}$($P$ value) for RG GWAS meta-analysis. Signals reaching genome-wide significance ($P < 5.0 \times 10^{-8}$) are colored in red. Crosses annotate loci that show evidence of sex heterogeneity ($P_{\text{sex-dimorphic}} < 5.0 \times 10^{-8}$ and $P_{\text{sex-heterogeneity}} < 0.05$); blue crosses for larger effects in men, green crosses for larger effects in women. Track 2 shows the effects of the 133 independent RG signals on four GIP/GLP-1-related traits GWAS. The colors of the dotted lines indicate four GIP/GLP-1-related traits: gray dot, signals reaching $P < 0.010$ for

a GIP/GLP-1-related trait; red dot, lead SNP has a significant effect on GIP/GLP-1-related trait (Bonferroni corrected $P < 1.0 \times 10^{-4}$). Track 3 shows the effects (−log$_{10}$($P$ value)) of the 133 independent RG signals on 113 glycan PheWAS. Track 4 shows the effects (−log$_{10}$($P$ value)) of the 133 independent RG signals on 210 gut-microbiome PheWAS. Track 5 shows MetaXcan results for ten selected tissues for RG GWAS meta-analysis; signals colocalizing with genes (Bonferroni corrected $P < 9.0 \times 10^{-7}$) are plotted for each tissue. All $P$ values were calculated from the two-sided $z$ statistics computed by dividing the estimated coefficients by the estimated standard error, without adjustment. **b**, Credible set analysis of RG associations in the European ancestry meta-analysis. Variants from each of the RG signal credible sets are grouped based on their posterior probability (the percentiles labeled on the sides of the bar). SNP variants with posterior probability >80%, along with their locus names, are provided. All variants from the credible set of lead signals are highlighted in bold.

**Characterization of RG-associated *GLP1R* coding variants provides a framework for T2D treatment stratification**

Following annotation and definition of likely causal variants, for functional studies, we prioritized *GLP1R*, which encodes a class B1 GPCR (GLP-1R) important in blood glucose and appetite regulation and a

well-established target of the T2D drugs exenatide (exendin-4) and semaglutide[20]. We used RG data to validate an experimental framework for predicting individual responses to GLP-1R agonists, as this would be a major asset in clinical practice and is currently lacking. Within *GLP1R*, the lead missense variant at rs10305492 (A316T) has a strong

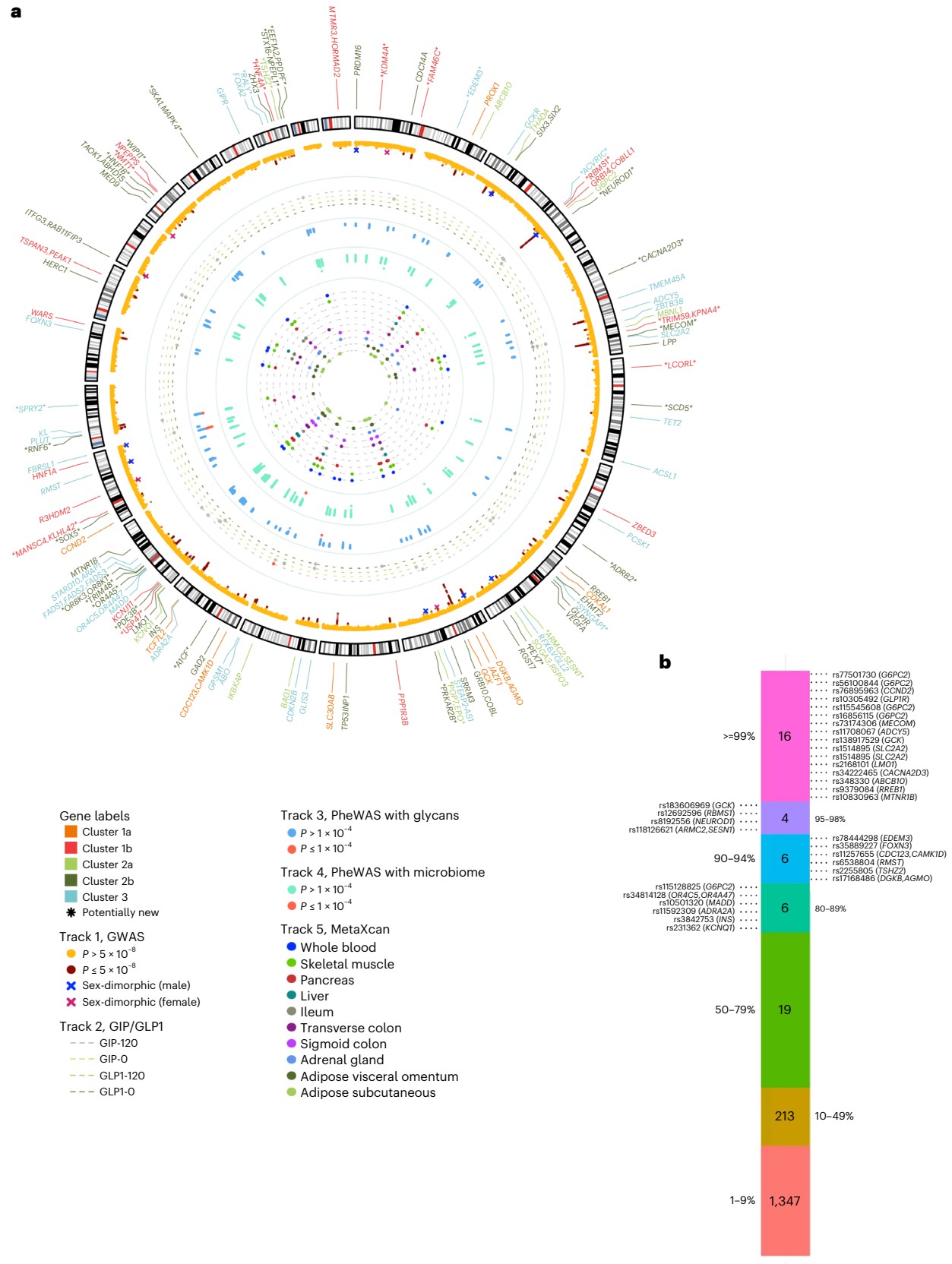

**Table 1 | New signals for glycemic traits discovered in GWAS meta-analysis of RG levels in up to 459,772 individuals of European ancestries without diabetes**

| Signal | Nearest gene(s) | Variants | Chr | Position | Type/model | Alleles | EAF | Effect | SE | P value | P het | n |
|---|---|---|---|---|---|---|---|---|---|---|---|---|
| EUR | KDM4A | rs3791033 | 1 | 44,134,077 | lead/7 | T/C | 0.67 | −0.0017 | 0.00031 | $3.9×10^{-8}$ | 0.58 | 455,267 |
| EUR | FAM46C | rs1966228 | 1 | 118,144,332 | additional/5 | A/G | 0.75 | 0.0032 | 0.00034 | $1.3×10^{-20}$ | 0.98 | 412,368 |
| EUR | FAM46C | rs17656269 | 1 | 118,162,139 | lead/7 | T/C | 0.33 | 0.0030 | 0.00032 | $4.3×10^{-21}$ | 0.075 | 455,647 |
| EUR[a] | EDEM3 | rs78444298 | 1 | 184,672,098 | lead/5 | A/G | 0.020 | 0.0076 | 0.0011 | $2.8×10^{-12}$ | 0.68 | 398,925 |
| EUR | ACVR1C | rs58288813 | 2 | 158473008 | lead/5 | T/C | 0.95 | 0.0037 | 0.00066 | $2.3×10^{-8}$ | 0.0073 | 415,629 |
| EUR | ACVR1C | rs2848657 | 2 | 158495349 | additional/7 | A/T | 0.13 | 0.0026 | 0.00044 | $2.4×10^{-9}$ | 0.13 | 454,031 |
| EUR | RBMS1 | rs12692596 | 2 | 161,265,910 | lead/7 | T/C | 0.37 | 0.0019 | 0.00030 | $1.2×10^{-9}$ | 0.84 | 457,182 |
| EUR[a] | NEUROD1 | rs8192556 | 2 | 182,542,998 | lead/7 | T/G | 0.024 | 0.0053 | 0.00096 | $3.0×10^{-8}$ | 0.50 | 418,468 |
| EUR | CACNA2D3 | rs34222465 | 3 | 55,123,055 | lead/1 | A/G | 0.56 | −0.0019 | 0.00030 | $3.7×10^{-10}$ | 0.055 | 418,498 |
| EUR | TRIM59, KPNA4 | rs9799314 | 3 | 160,082,071 | lead/7 | T/C | 0.47 | 0.0018 | 0.00030 | $1.1×10^{-9}$ | 0.025 | 439,182 |
| EUR | MECOM | rs73174306 | 3 | 169,194,244 | lead/5 | A/T | 0.96 | −0.0059 | 0.00074 | $1.3×10^{-15}$ | 1.00 | 393,841 |
| EUR | LCORL | rs1503884 | 4 | 18,207,538 | lead/5 | T/G | 0.56 | −0.0018 | 0.00030 | $8.8×10^{-10}$ | 0.65 | 414,134 |
| EUR | SCD5 | rs4693043 | 4 | 83,563,582 | lead/7 | A/G | 0.14 | 0.0023 | 0.00042 | $2.9×10^{-8}$ | 0.66 | 456,696 |
| EUR | ADRB2 | rs71584073 | 5 | 148,149,418 | lead/5 | T/C | 0.92 | 0.0038 | 0.00056 | $1.7×10^{-11}$ | 0.91 | 398,925 |
| EUR | SYNGAP1 | rs9461856 | 6 | 33,395,199 | lead/1 | A/G | 0.49 | −0.0017 | 0.00030 | $4.9×10^{-9}$ | 0.29 | 436,654 |
| EUR | ARMC2, SESN1 | rs118126621 | 6 | 109,304,170 | lead/5 | A/G | 0.025 | 0.0055 | 0.00098 | $2.3×10^{-8}$ | 0.029 | 393,841 |
| EUR | PEX7 | rs7756291 | 6 | 137,235,325 | lead/7 | T/C | 0.55 | −0.0016 | 0.00030 | $4.4×10^{-8}$ | 0.47 | 434,769 |
| EUR | POP7, EPO | rs221798 | 7 | 100,287,495 | lead/5 | C/G | 0.11 | −0.0030 | 0.00047 | $7.1×10^{-11}$ | 0.78 | 415,738 |
| EUR | PRKAR2B | rs3801969 | 7 | 106,711,492 | lead/1 | T/G | 0.44 | 0.0017 | 0.00030 | $1.2×10^{-8}$ | 0.47 | 458,102 |
| EUR | A1CF | rs61856594 | 10 | 52,637,925 | lead/7 | A/G | 0.70 | 0.0022 | 0.00032 | $7.3×10^{-12}$ | 0.59 | 451,966 |
| EUR | ADRA2A | rs11195538 | 10 | 113,117,650 | additional/5 | T/C | 0.93 | 0.0031 | 0.00060 | $2.3×10^{-7}$ | 0.21 | 403,260 |
| EUR | TCF7L2 | rs144155527 | 10 | 114,737,633 | additional/5 | T/C | 0.019 | −0.0061 | 0.0011 | $3.5×10^{-8}$ | 0.33 | 398,925 |
| EUR | USP47 | rs11022029 | 11 | 11,806,317 | lead/5 | T/C | 0.85 | 0.0023 | 0.00042 | $3.4×10^{-8}$ | 0.75 | 414,134 |
| EUR | PDE3B | rs141521721 | 11 | 14,763,828 | lead/5 | A/C | 0.024 | 0.0054 | 0.00098 | $2.6×10^{-8}$ | 0.38 | 398,925 |
| EUR | OR4A5 | rs72913090 | 11 | 50,653,357 | lead/5 | A/C | 0.92 | 0.0033 | 0.00055 | $2.7×10^{-9}$ | 1.0 | 380,422 |
| EUR | TRIM48 | rs150587121 | 11 | 55,036,391 | lead/5 | T/C | 0.91 | 0.0030 | 0.00054 | $3.3×10^{-8}$ | 0.12 | 396,388 |
| EUR | OR8K3, OR8K1 | rs2170441 | 11 | 56,095,739 | lead/5 | A/G | 0.078 | −0.0032 | 0.00056 | $9.5×10^{-9}$ | 0.57 | 398,925 |
| EUR | CCND2 | rs3217791 | 12 | 4,384,669 | additional/7 | T/C | 0.074 | −0.0032 | 0.00059 | $8.2×10^{-8}$ | 0.69 | 393,841 |
| EUR | SOX5 | rs12581677 | 12 | 24,060,732 | lead/5 | A/G | 0.91 | 0.0032 | 0.00053 | $3.1×10^{-9}$ | 0.10 | 414,063 |
| EUR | MANSC4, KLHL42 | rs11049144 | 12 | 27,931,511 | lead/5 | A/C | 0.22 | −0.0022 | 0.00036 | $1.2×10^{-9}$ | 0.012 | 413,498 |
| EUR | RNF6 | rs12874929 | 13 | 26,781,607 | lead/1 | A/G | 0.77 | −0.0026 | 0.00035 | $5.5×10^{-14}$ | 1.0 | 456,162 |
| EUR | SPRY2 | rs4884144 | 13 | 80,678,136 | lead/5 | A/G | 0.67 | 0.0019 | 0.00032 | $1.2×10^{-9}$ | 0.38 | 411,619 |
| EUR | HERC1 | rs67507374 | 15 | 64,038,340 | additional/5 | A/T | 0.31 | −0.0024 | 0.00032 | $8.9×10^{-14}$ | 0.28 | 415,015 |
| EUR | HNF1B | rs10908278 | 17 | 36,099,952 | lead/5 | A/T | 0.52 | −0.0019 | 0.00030 | $2.3×10^{-10}$ | 0.39 | 398,925 |
| EUR[b] | NMT1 | rs2239923 | 17 | 43,176,804 | lead/1 | T/C | 0.29 | 0.0020 | 0.00030 | $1.1×10^{-9}$ | 0.54 | 458,104 |
| EUR | WIPI1 | rs2952295 | 17 | 66,447,421 | lead/5 | A/T | 0.23 | 0.0024 | 0.00035 | $4.5×10^{-12}$ | 0.14 | 398,925 |
| EUR | SKA1, MAPK4 | rs2957989 | 18 | 48,075,733 | lead/1 | A/G | 0.82 | 0.0021 | 0.00039 | $3.4×10^{-8}$ | 0.67 | 437,935 |
| EUR | RALY | rs7274168 | 20 | 32,435,978 | lead/1 | T/C | 0.48 | 0.0018 | 0.00030 | $4.5×10^{-9}$ | 0.75 | 443,728 |
| EUR | HNF4A | rs2267850 | 20 | 43,524,963 | lead/7 | T/C | 0.27 | −0.0021 | 0.00033 | $6.2×10^{-10}$ | 0.92 | 437,057 |
| EUR | TSHZ2 | rs2255805 | 20 | 51,627,634 | lead/5 | T/C | 0.58 | −0.0019 | 0.00030 | $1.5×10^{-10}$ | 0.90 | 414,134 |
| EUR | STX16–NPEPL1 | rs61285514 | 20 | 57,283,828 | lead/7 | A/G | 0.77 | 0.0021 | 0.00035 | $2.3×10^{-9}$ | 0.24 | 451,642 |
| EUR | EEF1A2, PPDPF | rs6122466 | 20 | 62,139,177 | lead/5 | A/G | 0.86 | −0.0026 | 0.00043 | $7.8×10^{-10}$ | 0.70 | 405,111 |

A lead signal was annotated as 'EUR' if it reached genome-wide significance ($P<5.0×10^{-8}$) in the meta-analysis of European ancestry cohorts in either of our two models of interest with adjustment for age, sex with or without time since last meal (where available) along with the exclusion of extreme hyperglycemia (RG > 20 mmol l$^{-1}$) or in their combination. Additional distinct signals with a region-wide threshold of $P≤1.0×10^{-5}$ are also reported. Effects and P values reported are from the model indicated in column 'type/model' (1, AS20; 5, AST20; 7, AS20+AST20). Heterogeneity among studies was assessed using the $I^2$ index. [a]Nonsynonymous variants. [b]Synonymous variants. Alleles, effect/other; Chr, chromosome; EAF, effect allele frequency (frequency of allele, for which beta is reported); EUR, individuals of European ancestry; Pos, position GRCh37.

**Table 2 | New signals for glycemic traits discovered through UK Biobank (UKBB) (European ancestry only) GWAS in other RG models, UKBB (European ancestry only) GWAS on rare variants and cross-ancestry meta-analysis of up to 476,326 individuals of European or other ancestries (Black, Indian, Pakistani and Chinese) in UKBB**

| Signal | Nearest gene(s) | Variants | Chr | Position | Type/model | Alleles | EAF | Effect | SE | P value | P het | n |
|---|---|---|---|---|---|---|---|---|---|---|---|---|
| UKBB | PEX7 | rs7756291 | 6 | 13,7235,325 | lead/6 | C/T | 0.45 | 0.0018 | 0.00030 | $3.0\times10^{-9}$ | – | 379,291 |
| UKBB | INAFM2 | rs882829 | 15 | 40,607,689 | lead/2 | C/G | 0.92 | 0.0032 | 0.00057 | $1.6\times10^{-8}$ | – | 379,301 |
| UKBB | INAFM2, C15orf52 | rs4143838 | 15 | 40,622,374 | lead/3 | T/C | 0.95 | −0.0039 | 0.00070 | $1.8\times10^{-8}$ | – | 379,947 |
| UKBB | ADCY9, SRL | rs2018506 | 16 | 4,227,922 | lead/6 | C/G | 0.85 | −0.0023 | 0.00042 | $2.2\times10^{-8}$ | – | 379,291 |
| UKBB | ERN1 | rs58642235 | 17 | 62,202,689 | lead/5 | T/C | 0.86 | −0.0024 | 0.00044 | $4.5\times10^{-8}$ | – | 380,422 |
| UKBB[a] | WIPI1 | rs883541 | 17 | 66,449,122 | In LD with lead/6 | G/A | 0.23 | 0.0023 | 0.00036 | $5.5\times10^{-11}$ | – | 380,422 |
| UKBB[b] | RFX1 | rs2305780 | 19 | 14,083,761 | lead/4 | T/C | 0.54 | 0.0016 | 0.00029 | $1.5\times10^{-8}$ | – | 378,819 |
| UKBB, rare[a] | ANKH | rs146886108 | 5 | 14,751,305 | rare/1 | T/C | 0.0072 | −0.012 | 0.0018 | $3.2\times10^{-12}$ | – | 380,432 |
| Cross-anc | RRNAD1 | rs3806415 | 1 | 156,698,265 | lead/5 | T/C | 0.32 | −0.0017 | 0.00031 | $3.6\times10^{-8}$ | 0.51 | 476,326 |
| Sex-dim (w) | SGIP1 | rs7544505 | 1 | 66,998,618 | lead/5 | T/C | 0.84 | −0.0030 | 0.00053 | $1.8\times10^{-8}$ | 0.019 | 207,903 |
| Sex-dim (m) | SGIP1 | rs7544505 | 1 | 66,998,618 | lead/5 | T/C | 0.84 | −0.0010 | 0.00063 | 0.10 | | 172,529 |
| Sex-dim (w) | POP7, EPO | rs534043 | 7 | 100,312,724 | lead/5 | A/G | 0.11 | −0.0018 | 0.00061 | 0.0029 | 0.0040 | 207,903 |
| Sex-dim (m) | POP7, EPO | rs534043 | 7 | 100,312,724 | lead/5 | A/G | 0.11 | −0.0046 | 0.00073 | $4.8\times10^{-10}$ | | 172,529 |
| Sex-dim (w) | SLC43A2 | rs56405641 | 17 | 1,528,464 | lead/5 | C/T | 0.91 | −0.0040 | 0.00067 | $2.0\times10^{-9}$ | $1.4\times10^{-4}$ | 207,903 |
| Sex-dim (m) | SLC43A2 | rs56405641 | 17 | 1,528,464 | lead/5 | C/T | 0.91 | $-4.1\times10^{-5}$ | 0.00081 | 0.96 | | 172,529 |

Loci showing sex-dimorphic effects on glycemic trait levels for the first time are also shown. A signal was annotated as 'UKBB' if it reached genome-wide significance ($P<5.0\times10^{-8}$) in UKBB (European ancestry) in any of the six RG models. A signal was annotated as 'UKBB, rare' if it reached genome-wide significance ($P<5.0\times10^{-8}$) in UKBB (European ancestry) analysis for rare variants. Additional distinct signals with a region-wide threshold of $P\leq1.0\times10^{-5}$ are also reported. Effects and P values reported are from the model indicated in column 'type/model' (2, ASB20; 3, AS11; 4, ASB11; 5, AST20; 6, ASTB20). Heterogeneity among studies was assessed using the $I^2$ index. P het values for the sex-dimorphic variants are from Cochran's Q test (for sex heterogeneity representing the differences in allelic effects between sexes). Sex-dimorphic P values (2 degrees of freedom test of association assuming different effect sizes between the sexes) for the SGIP1, POP7/EPO and SLC43A2 variants were $3.2\times10^{-8}$, $4.3\times10^{-11}$ and $1.5\times10^{-8}$, respectively. [a]Nonsynonymous variants. [b]Synonymous variants. Cross-anc, cross-ancestry; Sex-dim (m), sex-dimorphic results for men; Sex-dim (w), sex-dimorphic results for women.

(0.058 mmol l⁻¹ per allele) RG-lowering effect, second by size only to *G6PC2* locus variants, and is also associated with FG/T2D[21,22].

We functionally tested the impact of rs10305492 (A316T) and 16 other *GLP1R* coding variants detected in the UKBB dataset, with effect allele frequency ranging from common (G168S, rs6923761, $P_{RG\ GWAS\ meta-analysis} = 5.20\times10^{-5}$) to rare (R421W, rs146868158, $P_{RG\ GWAS\ meta-analysis} = 0.036$), by measuring GLP-1-induced recruitment of mini-G$\alpha_s$[23] in HEK293 cells stably expressing wild-type (WT) or variant GLP-1R. This approach captures the most proximal part of the G$\alpha_s$-adenylate cyclase-cyclic adenosine monophosphate pathway, which links GLP-1R activation to insulin secretion. With correction for differences in cell surface expression determined using SNAP-tag labeling[24], mini-G$_s$-coupling efficiency was indeed predictive of the RG effect for these variants (Fig. 2a and Supplementary Table 11), thereby linking experimentally measured GLP-1R function in vitro to blood glucose homeostasis. This relationship was assessed in UKBB WES data (Supplementary Note and Extended Data Fig. 4).

Focusing on the two directly genotyped *GLP1R* missense variants in UKBB, we also measured mini-G$_s$ responses to several endogenous and pharmacological GLP-1R agonists, observing that A316T (rs10305492-A) showed increased responses and R421W (rs146868158-T) showed reduced responses, to all ligands except exendin-4 (both variants) and semaglutide (A316T only), in line with their RG effects (Fig. 2b). Interestingly, for late-stage T2D candidate tirzepatide, which has pronounced 'biased agonism' at GLP-1R[25], the difference between A316T and R421W amounted to nearly tenfold difference in activity. The common G168S variant, with a relatively small RG-lowering effect ($\beta = -0.0013$, s.e. $= 3.1\times10^{-4}$), also showed increases in function with pharmacological agonist stimulation. As GLP-1R undergoes extensive agonist-induced endocytosis, a process that modulates the subcellular origin and temporal dynamics of receptor signaling[26], we also assessed

the endocytic characteristics of A316T, G168S and R421W variants using high content microscopy. Here the most notable observation was that agonist-induced GLP-1R endocytosis with R421W was normal despite its signaling deficit, suggesting a specific alteration to how this variant couples to downstream effectors[24]. These results, supported by RG data and clinical observations[27,28], suggest that in vitro assessments can provide valuable insights into the optimal selection of GLP-1R treatment according to genotype.

Next, we performed molecular dynamics (MD) simulations of human GLP-1R bound to oxyntomodulin (OXM)[29] to gain structural insights into the above-described *GLP1R* variant effects. A316T has a single amino acid substitution in the core of the receptor transmembrane (TM) domain (Fig. 2c) that leads to an alteration of the nearby hydrogen bond network that normally serves to stabilize the GLP-1R inactive state (Supplementary Video 1). Specifically, in A316T, residue T316^5.46 replaces Y242^3.45 (superscripts follow the study discussed in ref. 30 generic GPCR class B1 numbering system, where the number before the dot indicates the TM helix and the number after the dot refers to the sequence distance from the most conserved residue indicated by 50) in a persistent hydrogen bond with the backbone of P312^5.42, one turn of the helix above T316^5.46 (Fig. 2d,e and Supplementary Video 1). This triggers a local structural rearrangement that could transmit to the intracellular G-protein-binding site through TM3 and TM5, thereby enhancing G-protein coupling. A water molecule is close to position 5.46 in both A316T and WT (water cluster α5; Fig. 2f). Notably, the same water bridges the backbone of Y241^3.44 and A316^5.46 in WT or the backbone of Y241^3.44 and the side chain of T316^5.46 in A316T. Given the importance of conserved water networks in the activation of class A GPCRs[31,32], the stability of the hydrated spot close to position 5.46 corroborates the importance of this site for GLP-1R effects. In analogy with A316T, simulations with the G168S variant indicated the formation of a stable new hydrogen bond between

the side chain of residue S168[1.63] and A164[1.59], one turn above on the same helix (Fig. 2g and Supplementary Video 2). This moves the C-terminal end of TM1 closer to TM2 and reduces the overall flexibility of intracellular loop 1 (ICL1; Fig. 2h), altering the role of ICL1 in G-protein activation. In contrast to A316T and G168S, the site of variant R421W is consistent with persistent interactions with the G protein, and simulations predicted a propensity of R421W to interact with a different region of the G-protein β-subunit compared to WT (Fig. 2i). These results capture the full range of structural features in the current active GLP-1R models and provide clear clues about the dynamics of A316T and other GLP-1R variants, compared to early models that did not benefit from the structural insights obtained from cryo-electron microscopy[22].

For a broader view of the impact of *GLP1R* coding variation, we screened an additional 178 missense variants identified from exome sequencing[33] for exendin-4-induced mini-G_s coupling and endocytosis by transient transfection in HEK293 cells (Supplementary Note, Fig. 2j,k and Supplementary Table 12). In total, 110 variants showed a reduced response in either or both pathways ('LoF1') and 67 displayed a specific response deficit that was not fully explained by differences in GLP-1R surface expression ('LoF2'). Many of these defects were larger than in the analysis in Fig. 2a, with a major loss of GLP-1R function a likely consequence, meaning that patients carrying these variants are less likely to benefit from GLP-1R agonist drug treatment.

## Functional annotation of RG associations and intestinal health

Previous T2D and glycemic trait GWAS have primarily implicated pancreatic, adipose and liver tissues[3]. We performed a range of complementary functional annotation analyses by leveraging our RG GWAS results to identify additional cell and tissue types with etiological roles in glucose metabolism. Data-driven expression prioritized integration for complex traits (DEPICT)[34], which predicts enriched tissue types from prioritized gene sets, highlighted intestinal tissues including ileum and colon, as well as pancreas, adrenal glands[5], adrenal cortex and cartilage (false discovery rate, FDR < 0.20; Fig. 3a,b and Supplementary Table 13). Similarly, CELL type expression-specific integration for complex traits (CELLECT)[35], which facilitates cell type prioritization based on single-cell RNA-sequencing (scRNA-seq) datasets, identified large intestinal tissue as second-ranked only to pancreatic cell types (Fig. 4 and Supplementary Table 14). Interestingly, RG variants were related particularly to enriched expression in pancreatic polypeptide cells, exceeding even the more conventionally implicated insulin-secreting β cells. Supporting evidence was obtained from transcriptome-wide

association study (TWAS) analysis, where we identified a total of 216 (119 unique) significant genetically driven associations across the ten tested tissues (Supplementary Table 15a); 51 (25 unique) of highlighted genes are located at genome-wide significant RG loci (Supplementary Table 15). TWAS signals in skeletal muscle[5] showed the largest overlap with RG signals, such as *GPSM1* (ref. 36) and *WARS*. The combined results from ileum and colon also showed high enrichment, including the new *NMT1* and the established *FADS1/3* and *MADD* genes (Fig. 1a and Supplementary Table 15). Expression quantitative trait locus (eQTL) colocalization analyses, using eQTLgen whole blood expression data from 31,684 individuals[37,38] and the COLOC2 approach, identified 14 loci with strong links (posterior probability >70%) to gene expression data, including *TET2* (ref. 39), *KCNJ11*, *KLHL42*, *IKBKAP* and *CAMK1D*, with transcriptional effects in pancreatic islets and kidney mesangial cells (Supplementary Table 16). Similar analyses of human pancreatic islets regulatory variation in the translational human pancreatic islet geno-type tissue-expression resource (TIGER) dataset[38] defined 58 loci with strong statistical support for colocalization of the effects on RG and tissue expression of *ADCY5*, *RNF6*, *FADS1*, *MADD* and *STARD10* (ref. 40), in addition to *KLHL42* and *CAMK1D*, with the latter overlapping in whole blood. Moreover, epigenetic annotations using the GARFIELD tool highlighted significant ($P < 2.5 \times 10^{-5}$) enrichment of RG-associated variants in the fetal large intestine, as well as blood, liver and other tissues (Extended Data Fig. 5 and Supplementary Table 17). Adult intestinal tissues are not available in GARFIELD except for colon. Prompted by multiple analyses highlighting a potential role for the digestive tract in glucose regulation, we assessed the overlap between our signals and those from the latest gut-microbiome GWAS[41] and identified two genera sharing signals and direction of effect with RG at one locus: *Collinsella* and *Lachnospiraceae*-FCS020 at ABO-*FUT2* (Fig. 1a and Supplementary Table 18). The ABO-*FUT2* locus effects on RG could be mediated by the abundance of *Collinsella*/*Lachnospiraceae*-FCS02, producing glucose from lactose and galactose[42]. *Collinsella* genus affects gut permeability via interleukin-17A[43] and shows higher abundance in individuals with T2D compared to those with normal glucose tolerance and individuals with prediabetes[44]. Moreover, weight loss decreases *Collinsella* among obese individuals with T2D[45]. Higher prevalence of *the Lachnospiraceae* family is associated with metabolic disorders, while genus *Lachnospiraceae*-FCS02 abundance shows an inverse correlation with serum triglycerides[46]. However, the mechanism of their enrichment has yet to be studied. This multi-omics annotation provided strong evidence for links between RG and intestinal health.

**Fig. 2 | Functional and structural analysis of coding *GLP1R* variants. a**, Minor allele frequency-weighted linear regression was used to test if mini-G_s response to GLP-1 stimulation substantially predicted point estimates of *GLP1R* variant effect on RG levels (AST20 $\beta_{RG}$ as estimated in the UKBB study, $n_{max}$ = 401,810). Mini-G_s response to GLP-1 stimulation was corrected for variant surface expression ($n_{max}$ = 22, exact *n* for each variant is provided in Supplementary Table 11). Error bars extend one standard error above and below the point estimate. Size of the dots is proportional to the weight applied in the regression model. The regression results (coefficient of determination $R^2$ = 0.74, $F(1, 15)$ = 47.5, $P = 5.1 \times 10^{-6}$) suggest that mini-G_s coupling in response to GLP-1 stimulation predicts the effect of these coding variants on RG levels (AST20 $\beta_{RG}$ = −0.030; 95% confidence interval (CI) = −0.039 to −0.020; $P = 5.1 \times 10^{-6}$). The gray shaded area around the regression line corresponds to the 95% CI of predictions from the model. Variants in red showed no detectable surface expression (NDE) and are not included in regression analysis. **b**, Mean *GLP1R* variant mini-G_s coupling and receptor endocytosis, with surface expression correction, in response to GLP-1, OXM, glucagon (GCG), exendin-4 (Ex4), semaglutide (Sema) and tirzepatide (TZP), *n* = 6. Positive deviation indicates variant gain-of-function, with statistical significance inferred when the 95% CIs shown do not cross zero. Responses are also compared between pathways by unpaired *t* test, with an asterisk indicating statistically significant differences. **c**, Architecture of the complex formed between the agonist-bound GLP-1R and G_s; the likely effect triggered by residues involved in GLP-1R isoforms A316T, G168S and R421W (in magenta) are reported.

**d**, Distributions of the distance between Y242[3.45] side chain and P312[5.42] backbone computed during molecular dynamics simulations of GLP-1R WT and A316T; the cut-off distance for hydrogen bond is shown. **e**, Difference in the hydrogen bond network between GLP-1R WT and A316T. **f**, Analysis of water molecules within the TMD of GLP-1R WT and A316T suggests minor changes in the local hydration of position 5.46 (unperturbed structural water molecule). Also, a stabilizing role for the water molecules at the binding site of the G protein (water cluster apha5) cannot be ruled out. **g**, Distributions of the distance between position 168[1.63] and Y178[2.48] during molecular dynamics simulations of GLP-1R WT and G168S. **h**, During molecular dynamics simulations, the GLP-1R isoform S168G showed increased flexibility of ICL1 and H8 compared to WT, suggesting a different influence on G-protein intermediate states. **i**, Contact differences between G_s and GLP-1R WT or W421R; the C terminal of W421R H8 made more interactions with the N terminal segment of the G_s β subunit. **j**, Mini-G_s and GLP-1R endocytosis responses to 20 nM exendin-4, plotted against surface GLP-1R expression, from 196 missense *GLP1R* variants transiently transfected in HEK293T cells (*n* = 5 repeats per assay), with data represented as mean ± s.e.m. after normalization to WT response and $\log_{10}$-transformation. Variants are categorized as 'LoF1' when the response 95% CI falls below zero or 'LoF2' where the expression-normalized 95% CI falls below zero. **k**, GLP-1R snake plot created using gpcr.com summarizing the functional impact of missense variants; for residues with >1 variant, classification is applied as LoF2 > LoF1 > tolerated.

Finally, we observed associations at *HNF1A*[47] with nine total plasma N-glycome traits[48] at a Bonferroni-corrected threshold (Fig. 1a and Supplementary Table 19). These traits represent highly branched galactosylated sialylated glycans (attached to an α1-acid protein, an acute-phase protein[49]), known to lead to chronic low-grade inflammation[50,51] and an increased risk of T2D[52–54] that might be explained by the role of N-glycan branching of the glucagon receptor in glucose homeostasis[55]. In addition, ten glycans showed association with five RG loci (*HNF1A*, *BAG1*, *PLUT*) at a

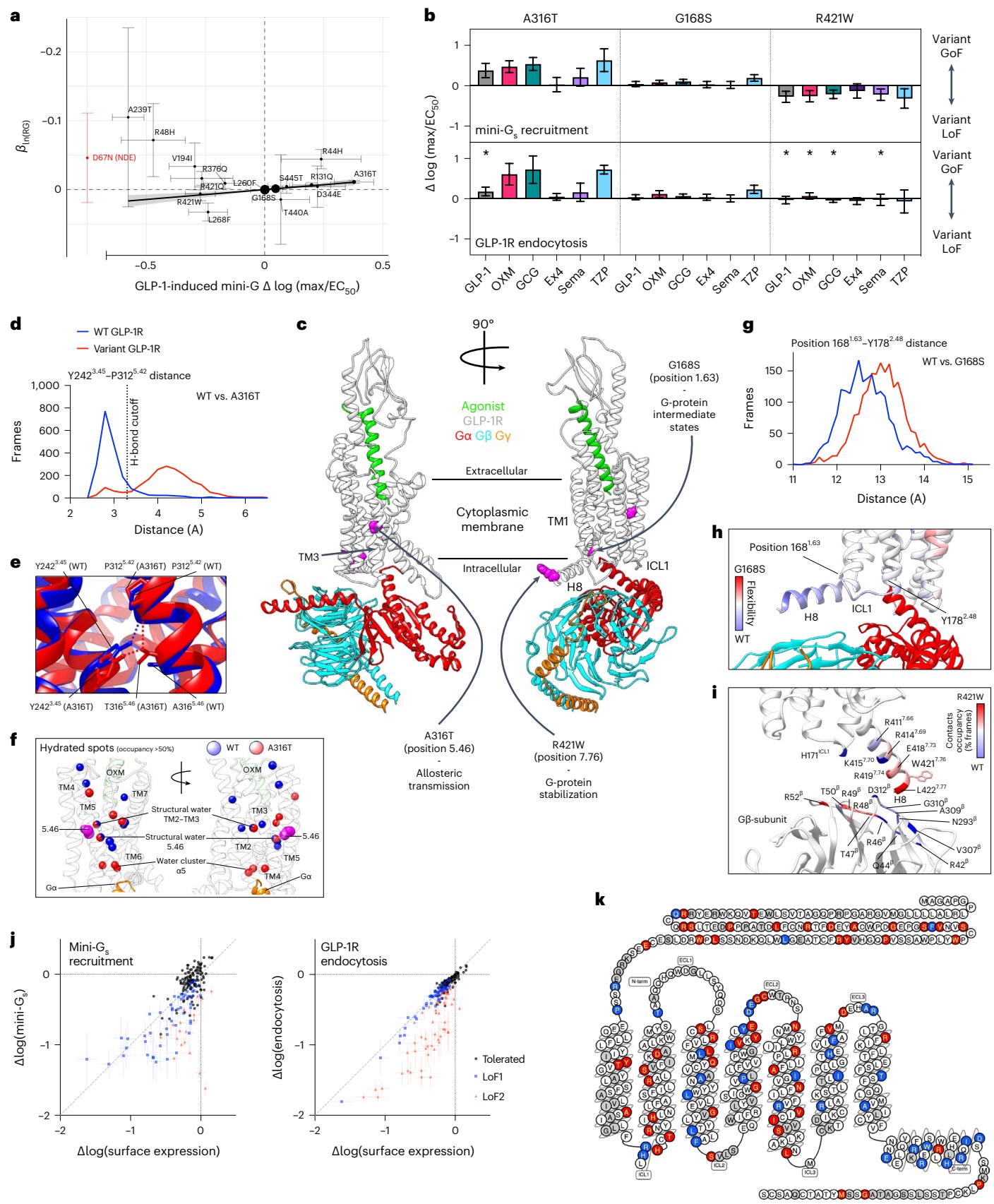

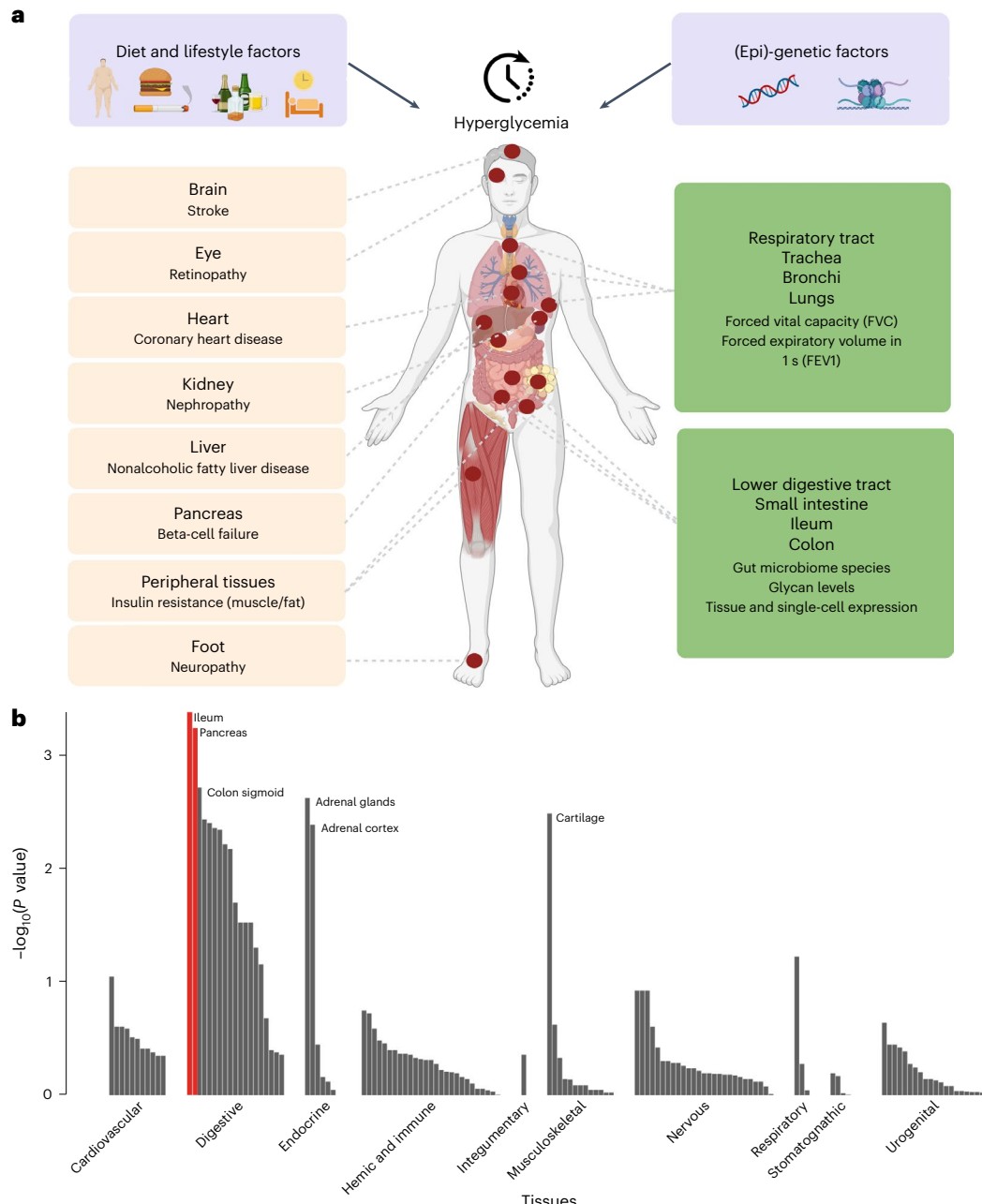

**Fig. 3 | Deterioration of glucose homeostasis progressing into T2D and leading to complications in multiple organs and tissues.** Established (left, in peach) and new (right, in green). **a**, A human figure illustrating the main causes of hyperglycemia (a combination of lifestyle and genetic factors), and how hyperglycemia affects many organs and tissues. Complications on the left panel are well-established for T2D. Those on the right panel are emerging ones and are supported by our current analyses. Figure created with BioRender.com. **b**, DEPICT prioritization of 134 tissues from the GTEx Project highlights the ileum and pancreas (shown in red, one-sided empirical $P$ value with FDR < 0.05 determined against randomized phenotypes in a null GWAS).

suggestive level of significance (Fig. 1a). Among them, three are attached to immunoglobulin G molecules[49], and their increased relative abundances are associated with a lower risk of T2D[56] and diminished inflammation status[57]. These observations suggest an overlap between networks regulating RG homeostasis and plasma-protein N-glycosylation.

## Genetic relationships between RG and other metabolic or nonmetabolic traits

Using linkage-disequilibrium score regression analyses, we estimated the genetic correlations between RG and other phenotypes to quantify the shared genetic contribution. We detected positive genetic correlations between RG and squamous cell lung cancer ($r_g = 0.28$, $P = 0.0015$)

and lung cancer ($r_g = 0.12$, $P = 0.037$; Fig. 5 and Supplementary Table 20), as well as inverse genetic correlations with lung function related traits, such as forced vital capacity (FVC, $r_g = -0.090$, $P = 0.0059$) and forced expiratory volume in 1 second (FEV1, $r_g = -0.054$, $P = 0.017$; Figs. 3a and 5 and Supplementary Table 20). To investigate this further, we conducted bidirectional Mendelian randomization (MR) analysis, which suggested a causal effect of RG and T2D on lung function, including FEV1 ($\beta_{MR-RG} = -0.66$, $P = 9.6 \times 10^{-5}$; $\beta_{MR-T2D} = -0.049$, $P = 1.3 \times 10^{-13}$) and FVC ($\beta_{MR-RG} = -0.60$, $P = 1.5 \times 10^{-4}$; $\beta_{MR-T2D} = -0.062$, $P = 1.4 \times 10^{-21}$), but not vice versa (RG $\beta_{MR-FEV1} = -0.0048$, $P = 0.42$; $\beta_{MR-FVC} = -0.01$, $P = 0.17$ and T2D ($\beta_{MR-FEV1} = -0.18$, $P = 0.040$; $\beta_{MR-FVC} = -0.21$, $P = 0.040$; Supplementary Table 21a,b). External factors, such as smoking or sedentary lifestyle,

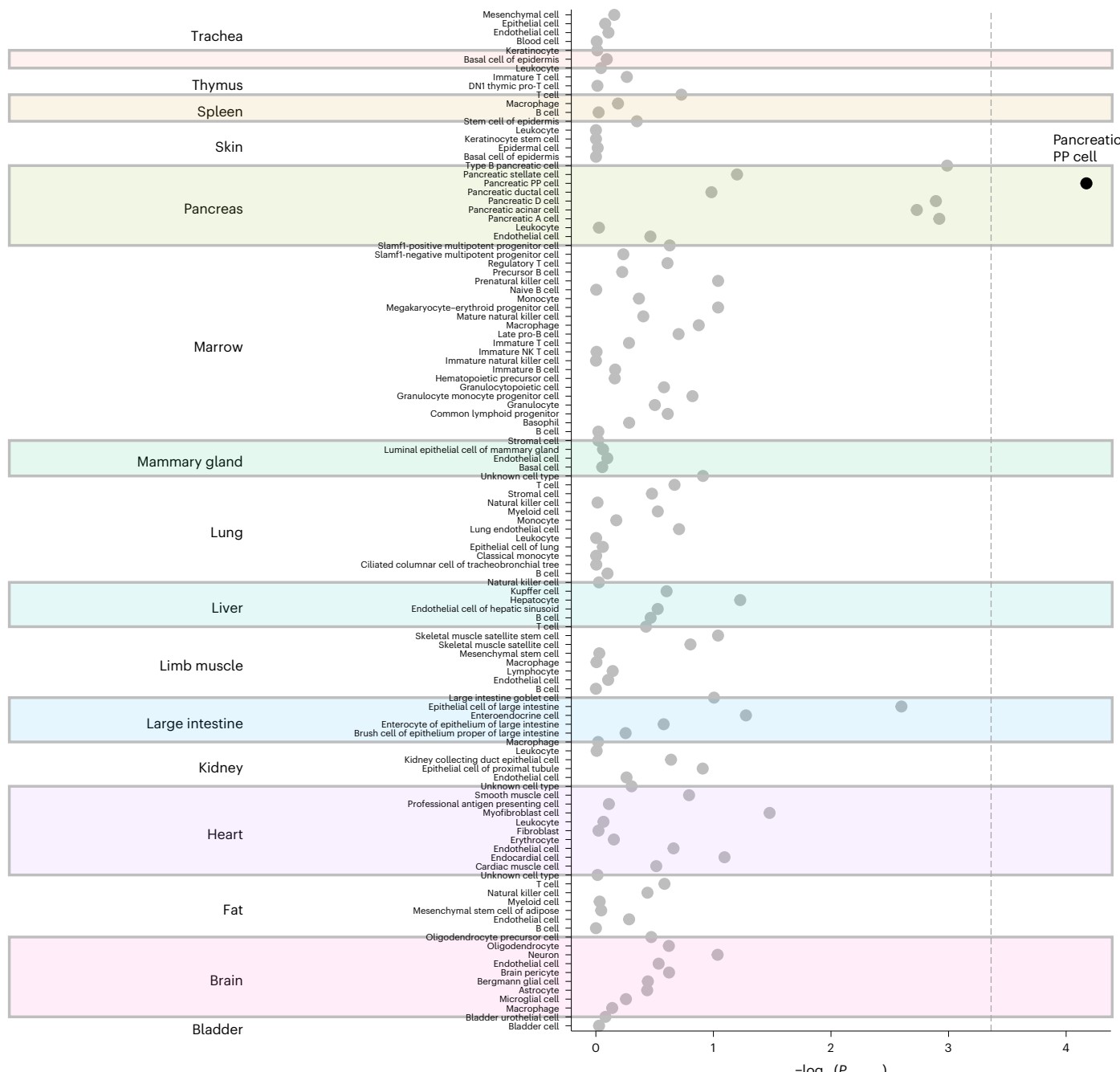

**Fig. 4 | Cell type prioritization across 17 tissues identified large intestinal tissue ranked second only to pancreatic cell types.** CELLECT prioritization of 115 cell types from Tabula Muris highlights pancreatic polypeptide (PP) cells (shown in black, one-sided Wilcoxon rank-sum test with significance threshold depicted by a dotted line indicating cell types with a nominal $P_{S\text{-}LDSC}} < 4.3 \times 10^{-4}$).

could cause lung function to decline, independent of RG and T2D effects. We implemented multivariable MR (MVMR) and found (Supplementary Table 21c) that RG and T2D causal effects on FVC are independent of both cigarettes smoked per day (CPD; that is, proxy for smoking[58]) and leisure screen time (LST; that is, proxy for physical activity[59]). This is important as previous observational studies have highlighted worsening lung function, as defined by FVC, in patients with T2D, but whether this was a causal relationship was not clear[60,61]. More recently, it was shown that patients with diabetes are at an increased risk of death from the viral infection COVID-19 (ref. 62), with pulmonary dysfunction contributing to mortality[63]. Our data confirm the causal effect of glycemic dysregulation on a decline in lung function as a new complication of diabetes.

Genome-wide genetic correlation analyses also showed a strong positive genetic correlation of RG with FG ($r_g = 0.88$, $P = 6.93 \times 10^{-61}$; Fig. 4 and Supplementary Table 20). We meta-analyzed RG studies other than UKBB with FG GWAS summary statistics[64], observing 79 signals reaching nominal significance that were directionally consistent in both UKBB and RG + FG (Supplementary Table 3), providing additional support to our RG findings. Given the large genetic overlap between RG, other glycemic traits and T2D, we evaluated the ability of a trait-specific polygenic risk score (PRS) to predict RG, T2D and HbA1c levels using UKBB effect estimates and the Vanderbilt cohort. The RG PRS explained 0.58% of the variance in RG levels when individuals with T2D were included (Supplementary Table 22), and 0.71% of the variance

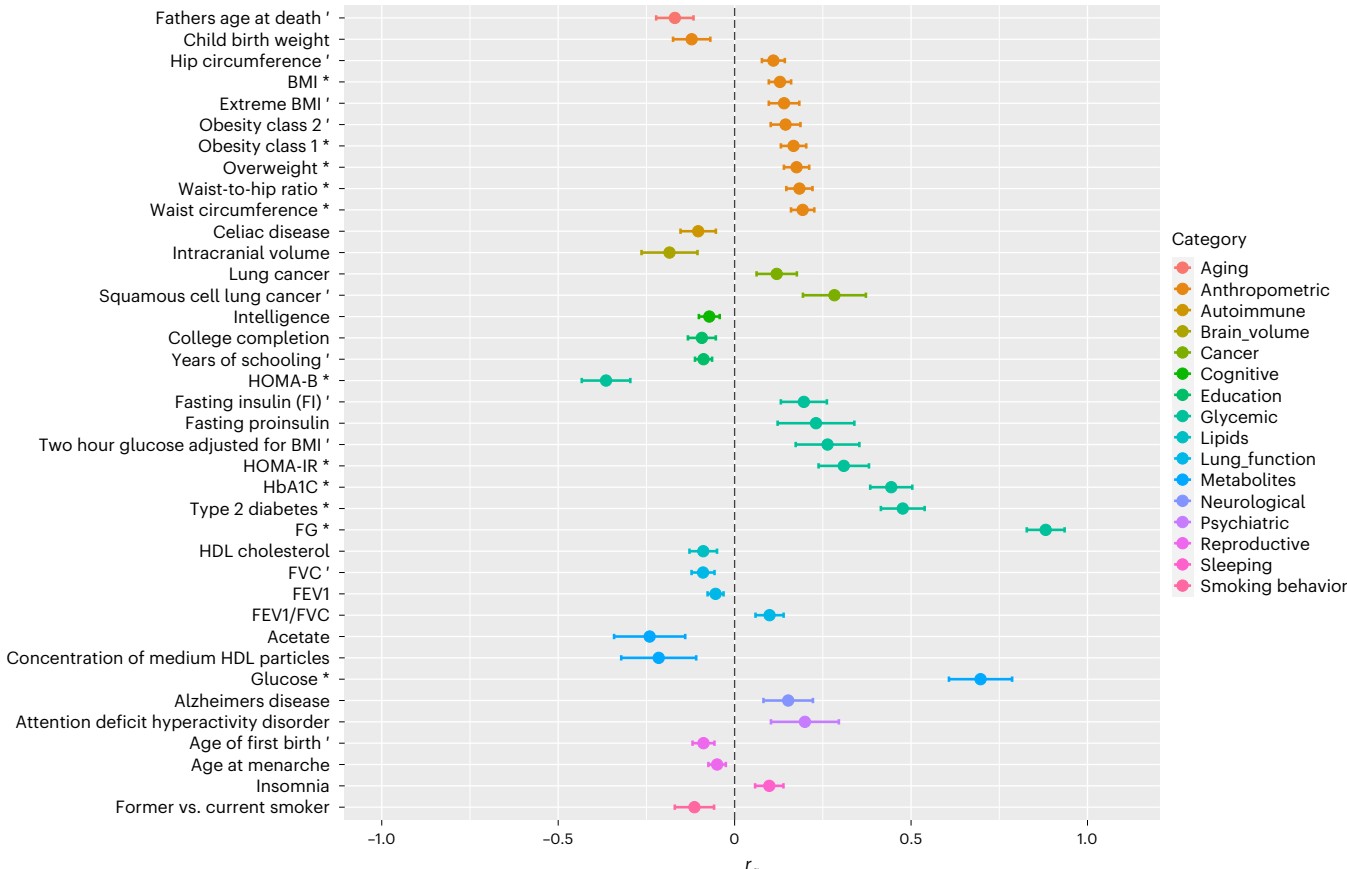

**Fig. 5 | Genome-wide genetic correlation between RG and a range of traits and diseases.** The *x* axis provides the estimated $r_g$ genetic correlation values for traits or diseases (*y* axis) reaching at least nominal significance (*P* < 0.05). Correlations reaching *P* < 0.010 are labeled with the prime symbol, and those *P* < 2.1 × 10⁻⁴ are labeled with the asterisk symbol. *P* values were calculated from the two-sided *z* statistics computed by dividing the estimated $r_g$ by the estimated standard error, without adjustment. Each error bar represents the standard error of the estimate.

after excluding those who developed T2D within 1 year of their last RG measurement. The RG PRS performance was comparable to that of the FG loci PRS (0.38% versus 0.42% for T2D; 0.40% versus 0.44% for HbA1c), indicating shared genetic variability determining glycemic traits.

We previously highlighted diverse effects of FG and T2D loci on pathophysiological processes related to T2D development by grouping associated loci in relation to their effects on multiple phenotypes[6]. Cluster analysis of the RG signals with 45 related phenotypes identified three separate clusters (Fig. 1a, Supplementary Table 23 and Extended Data Figs. 6 and 7), including 'metabolic syndrome' cluster 1, with 28 loci also leading to higher waist-to-hip ratio, blood pressure, plasma triglycerides, insulin resistance (HOMA-IR) and coronary artery disease risk, as well as lower sex hormone binding globulin levels in both sexes and testosterone in males. Cluster 3 was characterized, in particular, by insulin secretory defects[6]. Cluster 2 showed a primary effect on insulin release versus insulin action[3], but included a subcluster of 11 loci, which exert protective effects on inflammatory bowel disease, a relationship not previously reported. Moreover, cluster 2 was notable for generally reduced T2D risk in comparison to clusters 1 and 3, shaping the partial overlap between genetic determinants of glycemia and T2D that is known to exist[65]. This RG loci grouping gave innovative insights into the etiology of glucose regulation and associated disease states.

## Discussion

Leveraging data from 476,326 individuals, we have expanded by 44 the number of loci associated with glycemic traits. By using RG, our analysis integrates genetic contributions into a wider range of physiological stages, which thus far was not possible with standardized glycemic measures. Moreover, the greater statistical power obtained from large cross-ancestry meta-analysis improves confidence in identifying potentially causal variants, thereby helping to prioritize genes for more detailed functional analyses in the future. Our comprehensive functional characterization of *GLP1R* coding variation validates its role in blood glucose regulation and, more importantly, shows how GLP-1R-targeting drug responses depend on genetic variation. Notably, additional islet-expressed class B1 GPCRs identified in our current analysis and other glycemic trait/T2D GWAS, including *GIPR*, *GLP2R* (refs. 3,66) and *SCTR*[21], are investigational targets for T2D treatment, which should be subjected to similar analysis. Our functional annotation analyses point to underexplored tissue mediators of glycemic regulation, with new evidence highlighting the role of the intestine. This observation supports the profound effects of gastric bypass surgery on T2D resolution[67], as well as links between the intestinal microbiome and responses to several diabetes drugs[68]. In the near future, larger well-phenotyped datasets will enable high-dimensional GWAS investigations, disentangling the role of diet composition, physical activity and lifestyle on RG level variability in relation to genetic effects. Finally, through MR, we identified a causal effect of glucose levels and T2D on lung function, demonstrating the utility of this approach for corroborating findings from observational studies and elevating lung dysfunction as a new complication of diabetes.

## Online content

Any methods, additional references, Nature Portfolio reporting summaries, source data, extended data, supplementary information,

acknowledgements, peer review information; details of author contributions and competing interests; and statements of data and code availability are available at https://doi.org/10.1038/s41588-023-01462-3.

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

Vasiliki Lagou[1,2,3,93], Longda Jiang[4,5,93], Anna Ulrich[5,6,93], Liudmila Zudina[5,6], Karla Sofia Gutiérrez González[7,8], Zhanna Balkhiyarova[5,6,9], Alessia Faggian[5,6,10], Jared G. Maina[6,11], Shiqian Chen[12], Petar V. Todorov[13], Sodbo Sharapov[14,15], Alessia David[16], Letizia Marullo[17], Reedik Mägi[18], Roxana-Maria Rujan[19], Emma Ahlqvist[20], Gudmar Thorleifsson[21], He Gao[22], Evangelos Evangelou[22,23], Beben Benyamin[24,25,26], Robert A. Scott[27], Aaron Isaacs[7,28,29], Jing Hua Zhao[30], Sara M. Willems[7], Toby Johnson[31], Christian Gieger[32,33,34], Harald Grallert[32,34], Christa Meisinger[35], Martina Müller-Nurasyid[36,37,38,39], Rona J. Strawbridge[40,41,42], Anuj Goel[1,43], Denis Rybin[44], Eva Albrecht[36], Anne U. Jackson[45], Heather M. Stringham[45], Ivan R. Corrêa Jr.[46], Eric Farber-Eger[47], Valgerdur Steinthorsdottir[21], André G. Uitterlinden[7,48], Patricia B. Munroe[31,49], Morris J. Brown[31],

Julian Schmidberger[50], Oddgeir Holmen[51], Barbara Thorand[33,34], Kristian Hveem[52], Tom Wilsgaard[53,54], Karen L. Mohlke[55], Zhe Wang[56], GWA-PA Consortium*, Aleksey Shmeliov[6], Marcel den Hoed[57], Ruth J. F. Loos[13,56,58], Wolfgang Kratzer[50], Mark Haenle[50], Wolfgang Koenig[59,60,61], Bernhard O. Boehm[62], Tricia M. Tan[12], Alejandra Tomas[63], Victoria Salem[64], Inês Barroso[65], Jaakko Tuomilehto[66,67,68], Michael Boehnke[45], Jose C. Florez[69,70,71], Anders Hamsten[40,41], Hugh Watkins[1,43], Inger Njølstad[53,54], H.-Erich Wichmann[33], Mark J. Caulfield[31,49], Kay-Tee Khaw[30], Cornelia M. van Duijn[7,72,73], Albert Hofman[7,74], Nicholas J. Wareham[27], Claudia Langenberg[27,75,76], John B. Whitfield[77], Nicholas G. Martin[77], Grant Montgomery[77,78], Chiara Scapoli[79], Ioanna Tzoulaki[22,23], Paul Elliott[22,80,81], Unnur Thorsteinsdottir[21,82], Kari Stefansson[21,82], Evan L. Brittain[83], Mark I. McCarthy[1,84,92], Philippe Froguel[5,11], Patrick M. Sexton[85,86], Denise Wootten[85,86], Leif Groop[20,87], Josée Dupuis[44,88], James B. Meigs[70,71,89], Giuseppe Deganutti[19], Ayse Demirkan[6,9,90], Tune H. Pers[13], Christopher A. Reynolds[19,91], Yurii S. Aulchenko[7,14,15], Marika A. Kaakinen[5,6,9,94] ✉, Ben Jones[12,94] ✉, Inga Prokopenko[6,9,11,94] ✉ & Meta-Analysis of Glucose and Insulin-Related Traits Consortium (MAGIC)*

[1]Wellcome Centre for Human Genetics, University of Oxford, Oxford, UK. [2]Human Genetics, Wellcome Sanger Institute, Hinxton, UK. [3]VIB-KU Leuven Center for Brain and Disease Research, Leuven, Belgium. [4]Institute for Molecular Bioscience, The University of Queensland, Brisbane, Queensland, Australia. [5]Department of Metabolism, Digestion and Reproduction, Imperial College London, London, UK. [6]Department of Clinical and Experimental Medicine, School of Biosciences and Medicine, University of Surrey, Guildford, UK. [7]Department of Epidemiology, Erasmus Medical Center, Rotterdam, the Netherlands. [8]Department of Molecular Diagnostics, Clinical Laboratory, Clinica Biblica Hospital, San José, Costa Rica. [9]People-Centred Artificial Intelligence Institute, University of Surrey, Guildford, UK. [10]Laboratory for Artificial Biology, Department of Cellular, Computational and Integrative Biology, University of Trento, Trento, Italy. [11]UMR 8199—EGID, Institut Pasteur de Lille, CNRS, University of Lille, Lille, France. [12]Section of Endocrinology and Investigative Medicine, Imperial College London, London, UK. [13]Novo Nordisk Foundation Center for Basic Metabolic Research, University of Copenhagen, Copenhagen, Denmark. [14]Laboratory of Glycogenomics, Institute of Cytology and Genetics SD RAS, Novosibirsk, Russia. [15]MSU Institute for Artificial Intelligence, Lomonosov Moscow State University, Moscow, Russia. [16]Centre for Bioinformatics and System Biology, Department of Life Sciences, Imperial College London, London, UK. [17]Department of Evolutionary Biology, Genetic Section, University of Ferrara, Ferrara, Italy. [18]Estonian Genome Centre, Institute of Genomics, University of Tartu, Tartu, Estonia. [19]Centre for Sports, Exercise and Life Sciences, Coventry University, Conventry, UK. [20]Lund University Diabetes Centre, Department of Clinical Sciences Malmö, Lund University, Malmö, Sweden. [21]deCODE genetics/Amgen, Inc., Reykjavik, Iceland. [22]Department of Epidemiology and Biostatistics, School of Public Health, Imperial College London, London, UK. [23]Department of Hygiene and Epidemiology, University of Ioannina Medical School, Ioannina, Greece. [24]Australian Centre for Precision Health, University of South Australia, Adelaide, South Australia, Australia. [25]Allied Health and Human Performance, University of South Australia, Adelaide, South Australia, Australia. [26]South Australian Health and Medical Research Institute, Adelaide, South Australia, Australia. [27]MRC Epidemiology Unit, Institute of Metabolic Science, University of Cambridge, Cambridge, UK. [28]CARIM School for Cardiovascular Diseases and Maastricht Centre for Systems Biology (MaCSBio), Maastricht University, Maastricht, the Netherlands. [29]Department of Physiology, Maastricht University, Maastricht, the Netherlands. [30]Department of Public Health and Primary Care, University of Cambridge, Cambridge, UK. [31]Clinical Pharmacology, William Harvey Research Institute, Barts and The London School of Medicine and Dentistry, Queen Mary University of London, London, UK. [32]Research Unit of Molecular Epidemiology, Institute of Epidemiology, Helmholtz Zentrum München Research Center for Environmental Health, Neuherberg, Germany. [33]Institute of Epidemiology, Helmholtz Zentrum München, German Research Center for Environmental Health, Neuherberg, Germany. [34]German Center for Diabetes Research (DZD), Neuherberg, Germany. [35]Epidemiology, Faculty of Medicine, University of Augsburg, Augsburg, Germany. [36]Institute of Genetic Epidemiology, Helmholtz Zentrum München, German Research Center for Environmental Health, Neuherberg, Germany. [37]IBE, Faculty of Medicine, LMU Munich, Munich, Germany. [38]Institute of Medical Biostatistics, Epidemiology and Informatics (IMBEI), University Medical Center, Johannes Gutenberg University, Mainz, Germany. [39]Department of Medicine I, University Hospital Grosshadern, Ludwig-Maximilians-University, Munich, Germany. [40]Cardiovascular Medicine Unit, Department of Medicine, Solna, Karolinska Institutet, Stockholm, Sweden. [41]Center for Molecular Medicine, Karolinska University Hospital Solna, Stockholm, Sweden. [42]School of Health and Wellbeing, University of Glasgow, Glasgow, UK. [43]Cardiovascular Medicine, Radcliffe Department of Medicine, University of Oxford, Oxford, UK. [44]Department of Biostatistics, Boston University School of Public Health, Boston, MA, USA. [45]Department of Biostatistics and Center for Statistical Genetics, University of Michigan, Ann Arbor, MI, USA. [46]New England Biolabs, Ipswich, MA, USA. [47]Vanderbilt Institute for Clinical and Translational Research and Vanderbilt Translational and Clinical Cardiovascular Research Center, Nashville, TN, USA. [48]Department of Internal Medicine, Erasmus Medical Center, Rotterdam, the Netherlands. [49]NIHR Barts Cardiovascular Biomedical Research Centre, Barts and The London School of Medicine and Dentistry, Queen Mary University of London, London, UK. [50]Department of Internal Medicine I, Ulm University Medical Centre, Ulm, Germany. [51]Department of Public Health and General Practice, Norwegian University of Science and Technology, Trondheim, Norway. [52]K G Jebsen Centre for Genetic Epdiemiology, Department of Public Health and General Practice, Norwegian University of Science and Technology, Trondheim, Norway. [53]Department of Community Medicine, Faculty of Health Sciences, University of Tromsø, Tromsø, Norway. [54]Department of Clinical Medicine, Faculty of Health Sciences, University of Tromsø, Tromsø, Norway. [55]Department of Genetics, University of North Carolina, Chapel Hill, NC, USA. [56]The Charles Bronfman Institute for Personalized Medicine, Icahn School of Medicine at Mount Sinai, New York City, NY, USA. [57]The Beijer Laboratory and Department of Immunology, Genetics and Pathology, Uppsala University and SciLifeLab, Uppsala, Sweden. [58]The Mindich Child Health and Development Institute, Icahn School of Medicine at Mount Sinai, New York City, NY, USA. [59]Deutsches Herzzentrum München, Technische Universität München, Munich, Germany. [60]German Centre for Cardiovascular Research (DZHK), Partner Site Munich Heart Alliance, Munich, Germany. [61]Institute of Epidemiology and Medical Biometry, University of Ulm, Ulm, Germany. [62]Lee Kong Chian School of Medicine, Nanyang Technological University Singapore, Singapore and Department of Endocrinology, Tan Tock Seng Hospital, Singapore City, Singapore. [63]Section of Cell Biology and Functional Genomics, Imperial College London, London, UK. [64]Department of Bioengineering, Imperial College London, South Kensington Campus, London, UK. [65]Exeter Centre of Excellence for Diabetes Research (EXCEED), University of Exeter Medical School, Exeter, UK. [66]Public Health Promotion Unit, Finnish Institute for Health and Welfare, Helsinki, Finland. [67]Department of Public Health, University of Helsinki, Helsinki, Finland. [68]Diabetes Research Unit, King Abdulaziz University, Jeddah, Saudi Arabia. [69]Center for Genomic Medicine and Diabetes Unit, Massachusetts General Hospital, Boston, MA, USA. [70]Programs in Metabolism and

Medical and Population Genetics, Broad Institute, Cambridge, MA, USA. [71]Department of Medicine, Harvard Medical School, Boston, MA, USA. [72]Centre for Medical Systems Biology, Leiden, the Netherlands. [73]Nuffield Department of Population Health, University of Oxford, Oxford, UK. [74]Netherlands Consortium for Healthy Ageing, the Hague, the Netherlands. [75]Computational Medicine, Berlin Institute of Health at Charité—Universitätsmedizin Berlin, Berlin, Germany. [76]Precision Healthcare University Research Institute, Queen Mary University of London, London, UK. [77]QIMR Berghofer Medical Research Institute, Brisbane, Queensland, Australia. [78]Institute for Molecular Bioscience, The University of Queensland, St Lucia, Queensland, Australia. [79]Department of Life Sciences and Biotechnology, University of Ferrara, Ferrara, Italy. [80]MRC Centre for Environment and Health, Imperial College London, London, UK. [81]National Institute for Health Research Imperial College London Biomedical Research Centre, Imperial College London, London, UK. [82]Faculty of Medicine, University of Iceland, Reykjavík, Iceland. [83]Vanderbilt University Medical Center and the Vanderbilt Translational and Clinical Cardiovascular Research Center, Nashville, TN, USA. [84]Oxford Centre for Diabetes, Endocrinology and Metabolism, University of Oxford, Oxford, UK. [85]Drug Discovery Biology, Monash Institute of Pharmaceutical Sciences, Monash University, Parkville, Victoria, Australia. [86]ARC Centre for Cryo-Electron Microscopy of Membrane Proteins, Monash Institute of Pharmaceutical Sciences, Monash University, Parkville, Victoria, Australia. [87]Finnish Institute for Molecular Medicine (FIMM), Helsinki University, Helsinki, Finland. [88]Department of Epidemiology, Biostatistics and Occupational Health, McGill University, Montreal, Quebec, Canada. [89]Division of General Internal Medicine, Massachusetts General Hospital, Boston, MA, USA. [90]Department of Genetics, University Medical Center Groningen, Groningen, the Netherlands. [91]School of Life Sciences, University of Essex, Colchester, UK. [92]Present address: Genentech, South San Francisco, CA, USA. [93]These authors contributed equally: Vasiliki Lagou, Longda Jiang, Anna Ulrich. [94]These authors jointly supervised this work: Marika A. Kaakinen, Ben Jones, Inga Prokopenko. *A full lists of authors and their affiliations appears at the end of the paper.
✉e-mail: m.kaakinen@imperial.ac.uk; ben.jones@imperial.ac.uk; i.prokopenko@surrey.ac.uk

## GWA-PA Consortium

Vasiliki Lagou[1,2,3,93], Zhanna Balkhiyarova[5,6,9], Robert A. Scott[27], Anne U. Jackson[45], Heather M. Stringham[45], André G. Uitterlinden[7,48], Barbara Thorand[33,34], Zhe Wang[56], Marcel den Hoed[57], Ruth J. F. Loos[13,56,58], Michael Boehnke[45], Nicholas J. Wareham[27], Claudia Langenberg[27,75,76], Tune H. Pers[13], Marika A. Kaakinen[5,6,9,94] & Inga Prokopenko[6,9,11,94]

## Meta-Analysis of Glucose and Insulin-Related Traits Consortium (MAGIC)

Vasiliki Lagou[1,2,3,93], Longda Jiang[4,5,93], Anna Ulrich[5,6,93], Liudmila Zudina[5,6], Karla Sofia Gutiérrez González[7,8], Zhanna Balkhiyarova[5,6,9], Alessia Faggian[5,6,10], Jared G. Maina[6,11], Letizia Marullo[17], Reedik Mägi[18], Emma Ahlqvist[20], Gudmar Thorleifsson[21], Beben Benyamin[24,25,26], Robert A. Scott[27], Aaron Isaacs[7,28,29], Jing Hua Zhao[30], Sara M. Willems[7], Toby Johnson[31], Christian Gieger[32,33,34], Harald Grallert[32,34], Christa Meisinger[35], Martina Müller-Nurasyid[36,37,38,39], Rona J. Strawbridge[40,41,42], Anuj Goel[1,43], Denis Rybin[44], Eva Albrecht[36], Anne U. Jackson[45], Heather M. Stringham[45], Valgerdur Steinthorsdottir[21], André G. Uitterlinden[7,48], Patricia B. Munroe[31,49], Morris J. Brown[31], Julian Schmidberger[50], Oddgeir Holmen[51], Barbara Thorand[33,34], Kristian Hveem[52], Tom Wilsgaard[53,54], Karen L. Mohlke[55], Wolfgang Kratzer[50], Mark Haenle[50], Wolfgang Koenig[59,60,61], Bernhard O. Boehm[62], Inês Barroso[65], Jaakko Tuomilehto[66,67,68], Michael Boehnke[45], Jose C. Florez[69,70,71], Anders Hamsten[40,41], Hugh Watkins[1,43], Inger Njølstad[53,54], H.-Erich Wichmann[33], Mark J. Caulfield[31,49], Kay-Tee Khaw[30], Cornelia M. van Duijn[7,72,73], Albert Hofman[7,74], Nicholas J. Wareham[27], Claudia Langenberg[27,75,76], John B. Whitfield[77], Nicholas G. Martin[77], Grant Montgomery[77,78], Unnur Thorsteinsdottir[21,82], Kari Stefansson[21,82], Mark I. McCarthy[1,84,92], Philippe Froguel[5,11], Leif Groop[20,87], Josée Dupuis[44,88], James B. Meigs[70,71,89], Yurii S. Aulchenko[7,14,15], Marika A. Kaakinen[5,6,9,94] & Inga Prokopenko[6,9,11,94]

## Methods

### Ethics

All participating studies were approved by their appropriate institutional review boards or committees, and written informed consent was obtained from all study participants. For all the participating studies, approval was received to use their data in the present work. Study-specific ethics statements are provided in the references listed in Supplementary Table 1.

### Phenotype definition and model selection for RG GWAS

We used RG (mmol l$^{-1}$) measured in plasma or in whole blood (corrected to plasma level using the correction factor of 1.13). Individuals were excluded from the analysis if they had a diagnosis of T2D or were on diabetes treatment (oral or insulin). Individual studies applied further sample exclusions, including pregnancy, fasting plasma glucose ≥7 mmol l$^{-1}$ in a separate visit, when available, and having type 1 diabetes (Supplementary Table 1). Details about RG modeling in the first set of six available cohorts (Supplementary Table 2) can be found in the Supplementary Note. For the GWAS, we included individuals based on the following two RG cut-offs: <20 mmol l$^{-1}$ (20) to account for the effect of extreme RG values and <11.1 mmol l$^{-1}$ (11), which is an established threshold for T2D diagnosis. We then evaluated the following six different models in GWAS according to covariates included and cut-offs used: (1) age (A) and sex (S), RG < 20 mmol l$^{-1}$ (AS20); (2) age, sex and BMI (B), RG < 20 mmol l$^{-1}$ (ASB20); (3) age and sex, RG < 11.1 mmol l$^{-1}$ (AS11); (4) age, sex and BMI, RG < 11.1 mmol l$^{-1}$ (ASB11); (5) age, sex, time since last meal (accounted for as $T$, $T^2$ and $T^3$), RG < 20 mmol l$^{-1}$ (AST20) and (6) age, sex, $T$, $T^2$ and $T^3$ and BMI, RG < 20 mmol l$^{-1}$ (ASTB20). Apart from the above, additional adjustments for study site and geographical covariates were also applied.

### RG meta-analyses

The GWAS meta-analysis of RG consisted of the following five components: (1) 37,239 individuals from ten European ancestry GWAS imputed up to the HapMap 2 reference panel; (2) 3,156 individuals from three European ancestry GWAS with Metabochip coverage; (3) 21,083 individuals from two European ancestry GWAS imputed up to 1000 Genomes reference panel; (4) 380,432 individuals of white European ancestry from the UKBB and (5) 16,983 individuals from the Vanderbilt cohort imputed to the HRC panel (Supplementary Note). We imputed the GWAS meta-analysis summary statistics of each component to all-ancestries 1000 Genomes reference panel[69] using the summary statistics imputation method implemented in the SS-Imp v0.5.5 software[70]. SNPs with imputation quality scores <0.7 were excluded. We then conducted inverse-variance meta-analyses to combine the association summary statistics from all components using METAL v2011-03-25 (ref. 71). We focused our meta-analyses on models AS20 (17 cohorts, $n_{max}$ = 459,772) and AST20 (when time from last meal was available in the cohort; 12 cohorts, $n_{max}$ = 417,290). For the FHS cohort, where no information was available for individuals with RG > 11.1 (an established threshold for 2hGlu concentration, which is a criterion for T2D diagnosis), AS11 model results were used. We also performed a meta-analysis using cohorts with time from the last meal available (AST20 model, 12 cohorts) combined with those lacking this information (AS20, five cohorts) to maximize the association power while taking into account $T$. We termed this analysis as AS20 + AST20 in the following text (17 cohorts, $n_{max}$ = 458,862). A signal was considered to be associated with RG if it reached genome-wide significance ($P < 5.0 \times 10^{-8}$) in the meta-analysis of UKBB and other cohorts in either of our two models of interest (AS20) or (AST20) or in their combination (AS20 + AST20).

Of 133 signals detected in the European ancestry subset (Supplementary Note), 105 were directionally consistent in the UK Biobank and other contributing studies grouped together, providing the discovery validation (Supplementary Table 3). We report the $P$ value from the combined model unless otherwise stated. Full results from

all models are provided in Supplementary Table 3. We checked for nominal significance ($P < 0.05$) and directional consistency of the effect sizes for the selected lead SNPs in the combined model in UKBB results versus other cohort results. We further extended the check between UKBB results and meta-analysis of other cohorts including FG GWAS meta-analysis[64], excluding overlapping cohorts. This meta-analysis conducted in METAL v2011-03-25 was sample size and $P$ value based due to the measures being at different scales (natural logarithm-transformed RG and untransformed FG).

### Cross-ancestry analyses and meta-analysis

We performed GWAS in non-European ancestry populations within UKBB that had a sample size of at least 1,500 individuals. These were Black ($n$ = 7,644), Indian ($n$ = 5,660), Pakistani ($n$ = 1,747) and Chinese ($n$ = 1,503). We further meta-analyzed our European ancestry cohorts with the cross-ancestry UKBB cohorts. The analyses were performed with BOLT-LMM v2.3 (ref. 72) and METAL v2011-03-25.

### Sex-dimorphic analysis

To evaluate sex dimorphism in our results, we meta-analyzed the UKBB and the Vanderbilt cohort with the GWAMA v2.1 software[73], which provides a 2 degrees of freedom (df) test of association assuming different effect sizes between the sexes. We evaluated the evidence for heterogeneity of allelic effects between sexes using Cochran's $Q$ statistic[73,74]. We considered a signal to show evidence of sex dimorphism if the sex-dimorphic $P$ value was $<5.0 \times 10^{-8}$ and if the sex heterogeneity $P$ value (1 df) was <0.05.

### Clumping and conditional analysis

We performed a standard clumping analysis (PLINK v1.90 (ref. 75) criteria $-P \leq 5 \times 10^{-8}$, $r^2 = 0.01$, window-size = 1 Mb, 1000 Genomes Phase 3 data as linkage disequilibrium (LD) reference panel) to select a list of near-independent signals. We then performed a stepwise model selection analysis (approximate conditional analysis) to replicate the analysis using GCTA v1.93.0 (ref. 76) with the following parameters: $P \leq 5 \times 10^{-8}$ and window-size = 1 Mb. We further checked for additional distinct signals by using a region-wide threshold of $P \leq 1.0 \times 10^{-5}$ for statistical significance. For validation and comparison, we also performed direct conditional analyses using BOLT-LMM v2.3 (Supplementary Note). We filtered the direct conditional analysis results and BOLT-LMM results by checking the LD between all the variants within the same locus and keeping only independent signals ($r^2 < 0.01$). LD was calculated from European reference haplotypes from the 1000 Genomes Project using LDlinkR v1.1.2 library.

### GLP-1R pharmacological and structural analysis

**Mini-G$_s$ recruitment assay.** Where stable cell lines were used (that is, Fig. 2a,b), WT or variant T-REx-SNAP-GLP-1R-SmBiT cells (Supplementary Note) were seeded in 12-well plates and transfected with 1 μg per well LgBiT-mini-G$_s$[23] (a gift from N. Lambert, Medical College of Georgia). The following day, GLP-1R expression was induced by the addition of tetracycline (0.2 μg ml$^{-1}$) to the culture medium for 24 h. For transient transfection assays (that is, Fig. 2j), HEK293T cells in poly-D-lysine-coated white 96-well plates were transfected using Lipofectamine 2000 with 0.05 μg per well WT or variant SNAP-GLP-1R-SmBiT plus 0.05 μg per well LgBiT-mini-G$_s$ and the assay performed 24 h later. Cells were then resuspended in Hank's balanced salt solution + furimazine (Promega) diluted 1:50 and seeded in 96-well half-area white plates, or the same reagent added to adherent cells for transient transfection assays. Baseline luminescence was measured over 5 min using a Flexstation 3 plate reader at 37 °C before the addition of ligand or vehicle. Agonists were applied at a series of concentrations spanning the response range. After agonist addition, luminescent signal was serially recorded over 30 min, and ligand-induced effects were quantified by subtracting individual well baselines. Signals

were corrected for differences in cell number as determined by bicinchoninic acid assay.

**Analysis of pharmacological data.** Technical replicates within the same assay were averaged to give one biological replicate. For concentration-response assays (Fig. 2a,b), ligand-induced responses were analyzed by three-parameter fitting in Prism 8.0 (GraphPad Software). As a composite measure of agonism[77], $\log_{10}$-transformed $E_{max}$/half maximal effective concentration ($EC_{50}$) values were obtained for each ligand/variant response. The WT response was subtracted from the variant response to give $\Delta\log(max/EC_{50})$, a measure of gain- or loss-of-function for the variant relative to WT. $\log_{10}$-transformed surface expression levels were obtained for each variant relative to WT; these were then used to correct mini-$G_s$ $\Delta\log(max/EC_{50})$ values for differences in variant GLP-1R surface expression levels, by subtraction with error propagation. GLP-1R internalization responses were already normalized to surface expression within each assay. Statistical significance between WT and variant responses was inferred if the 95% confidence intervals for $\Delta\log(max/EC_{50})$ did not cross zero[77]. Changes to the profile of receptor response between mini-$G_s$ recruitment and GLP-1R internalization were inferred if $P < 0.05$ with unpaired $t$ test analysis, with Holm–Sidak correction for multiple comparisons. For transient transfection assays (Fig. 2j), responses were normalized to WT response and $\log_{10}$ transformed to give $Log\ \Delta$ responses. Additionally, the impact of differences in the surface expression on functional responses was determined by subtracting the log-transformed normalized expression level from the log-transformed normalized response.

**Variance explained in RG effects by mini-$G_s$ recruitment at coding GLP1R variants.** RG (AST20 model) effects estimated in the UKBB study at 16 independent ($r^2 < 0.02$) coding *GLP1R* variants (Supplementary Table 11) were regressed on mini-$G_s$ coupling in response to glucagon-like peptide-1 (GLP-1) stimulation (corrected for surface expression) giving more weight to variants with higher minor allele frequency.

**Computational methods including MD simulations.** The active state structure of GLP-1R in complex with OXM[29] and $G_s$ protein was used to simulate WT GLP-1R and G168S, A316T and R421W. The WT systems and variants were prepared for MD simulations and equilibrated as reported[78]. AceMD3 3.3.0 (ref. 79) was used for production runs (four MD replicas of 500 ns each). AquaMMapS v1 analysis[80] was performed on 10 ns-long MD simulations of GLP-1R(WT) and GLP-1R(A316T) in complex with OXM, with all the α carbons restrained; coordinates were written every 10 ps of simulation.

**Credible set analysis**
After selecting the signals with each region based on different meta-analysis results from AS20, AST20 and AS20 + AST20 models, we further performed a credible set analysis to obtain a list of potential causal variants for each of the 133 selected signals (Supplementary Note). We also calculated credible sets for the cross-ancestry meta-analysis and compared the results between the European ancestry-only and cross-ancestry meta-analyses.

**DEPICT analysis**
DEPICT uses GWAS summary statistics and computes a prioritization of genes in associated loci, which are used to prioritize tissues via enrichment analysis. DEPICT v1_rel 194 was used with default settings and RG GWAS summary statistics as input against a genetic background of SNPsnap data[81] derived from the 1000 Genomes Project Phase 3 (ref. 82) to prioritize genes (Supplementary Note).

**CELLECT analysis**
CELLECT[35] v1.0.0 and Cell type EXpression-specificity[35] v1.0.0 are two toolkits for genetic identification of likely etiologic cell types using

GWAS summary statistics and scRNA-seq data. Tabula Muris gene expression data[83], a scRNA-seq dataset derived from 20 organs from adult male and female mice, was preprocessed as described in the Supplementary Note.

**Genetically regulated gene expression analysis**
We used MetaXcan (S-PrediXcan) v0.6.10 (ref. 84) to identify genes whose genetically predicted gene expression levels are associated with RG in a number of tissues. The tested tissues were chosen based on their involvement in glucose metabolism. Those were adipose visceral omentum, adipose subcutaneous, skeletal muscle, liver, pancreas and whole blood. Additionally, we tested ileum, transverse colon, sigmoid colon and adrenal gland because they were highlighted by DEPICT analysis. The models for the tissues of interest were trained with GTEx Version 7 transcriptome data from individuals of European ancestry[85]. The tissue transcriptome models and 1000 Genomes[86] based covariance matrices of the SNPs used within each model were downloaded from PredictDB Data Repository. The association statistics between predicted gene expression and RG were estimated from the effects and their standard errors coming from the AS20 + AST20 model. Only statistically significant associations after Bonferroni correction for the number of genes tested across all tissues ($P \le 9.0 \times 10^{-7}$) were included in the table. Genes, where less than 80% of the SNPs used in the model were found in the GWAS summary statistics, were excluded due to the low reliability of the association result.

**GARFIELD analysis**
We applied the GWAS analysis of regulatory or functional information enrichment with LD correction (GARFIELD) tool v2 (ref. 87) on the RG AS20 + AST20 meta-analysis results to assess the enrichment of the RG-associated variants within functional and regulatory features. GARFIELD integrates various types of data from a number of publicly available cell lines. Those include genetic annotations, chromatin states, DNaseI hypersensitive sites, transcription factor binding sites, FAIRE-seq elements and histone modifications. We considered enrichment to be statistically significant if the RG GWAS $P$ value reached $1 \times 10^{-8}$ and the enrichment analysis $P$ value was $<2.5 \times 10^{-5}$ (Bonferroni corrected for 2,040 annotations).

**Genetic association with gut microbiome**
We assessed the genetic overlap between RG GWAS results and those for gut microbiome. GWAS of microbiome profiles were publicly available and downloaded from https://mibiogen.gcc.rug.nl/. For each of the 210 taxa, the corresponding $P$ values for the 133 RG GWAS SNPs and their proxies were extracted.

**Genetic association with GLP-1 and gastric inhibitory polypeptide (GIP)**
We assessed the genetic overlap between RG GWAS results and those for GLP-1 and GIP measured at 0 min and 120 min. We extracted the results for the 133 RG signals from the GWAS summary statistics for GLP-1 and GIP[88].

**eQTL colocalization analysis**
We further performed colocalization analysis using whole blood gene eQTL data provided by eQTLGen[37] and human pancreatic islets eQTLs provided by TIGER[38] for all 133 RG signals. We used meta-analysis results from AS20, AST20 or AS20 + AST20 depending on the degree of association of each signal. Only *cis*-eQTL data from eQTLGen/TIGER were incorporated to reduce the computational burden. The COLOC2 Bayesian-based method[89] was used to interrogate the potential colocalization between RG GWAS signals and the genetic control of gene expression. First, for each signal, depending on which model (AS20, AST20 or AS20 + AST20) had the lowest GWAS $P$ value, we extracted the RG GWAS test statistics of all SNPs within ±1 Mb region around the

133 RG signals. Then, for each RG signal, we matched the eQTLGen/TIGER results with the RG results and performed COLOC2 analysis evaluating the posterior probability of the following five hypotheses for each region: $H_0$, no association; $H_1$, GWAS association only; $H_2$, eQTL association only; $H_3$, both GWAS and eQTL association, but not colocalized and $H_4$, both GWAS and eQTL association and colocalized. Only GWAS signals with at least one nearby gene/probe reaching posterior probability $(H_4) \geq 0.5$ were reported. We considered signals to have strong evidence of colocalization if posterior probability $(H_4) > 0.7$.

### Genetic association with human blood plasma N-glycosylation
We assessed genetic associations between 133 RG signals and 113 human blood plasma N-glycome traits using previously published genome-wide association summary statistics[90]. The description of the analyzed traits and details of the association analysis can be found elsewhere[48]. We considered associations to be significant when $P < 0.05/113/133 = 3.3 \times 10^{-6}$ (after Bonferroni correction). Association was considered suggestive when $P < 10^{-4}$.

### Genetic correlation analysis
We investigate the shared genetic component between RG and other traits, including glycemic ones, by performing genetic correlation analysis using the bivariate LD score regression method (LDSC v1.0.0)[91]. To reduce multiple testing burden, only the GWAS results of the AS20 + AST20 model were used. We used GWAS summary statistics available in LDhub[92] and the Meta-Analysis of Glucose and Insulin-related Traits Consortium (MAGIC) website (https://www.magicinvestigators.org) for several traits including FG/FI[64], HOMA-B/HOMA-IR[93]. In total, 228 different traits were included in the genetic correlation analysis with RG. We considered $P \leq 2.2 \times 10^{-4}$ (Bonferroni correction for 228 traits) as the statistical significant level and $P \leq 0.05$ as the nominal level.

### MR analysis
We applied a bidirectional two-sample MR strategy (Supplementary Note) to investigate causality between RG and lung function, as well as T2D and lung function using independent genetic variants as instruments. We looked for evidence for the presence of a causal effect of RG and T2D on the following two lung function phenotypes: FEV1 and FVC in a two-sample MR setting. Genome-wide summary statistics for the lung function phenotypes were available[94], involving cohorts from the SpiroMeta consortium and the UKBB study. T2D susceptibility variants and their effects were obtained from the largest-to-date T2D GWAS[4].

To avoid confounding due to sample overlap, lung function summary statistics used as outcome data were those estimated in the SpiroMeta consortium alone. Similarly, when testing the effect of lung function on RG, RG genetic effects used as outcome data were estimated in all cohorts except UK Biobank. There was no sample overlap between the lung function and the T2D GWAS, thus allowing the use of T2D effects estimated in all contributing European ancestry studies. Genome-wide T2D summary statistics were available from a previous study[3] to test for the causal effect of lung function on T2D. All analyses were conducted using the R software package TwoSampleMR v0.5.4 (ref. 95).

Causal effects were estimated using the inverse-variance weighted method, which combines the causal estimates of individual instrumental variants (Wald ratios; Supplementary Note) in a random-effects meta-analysis[96]. Instrument heterogeneity $Q$ statistic $P$ values are reported. As a sensitivity analysis, we used MR-Egger regression (Supplementary Note) to test for the presence of horizontal pleiotropy and obtain causal estimates that are more robust to the inclusion of invalid instruments[97].

MVMR is an extension of MR that can be applied with either individual or summary-level data to estimate the effect of multiple, potentially related, exposures on an outcome[98]. We used the MVMR v0.3 R package to test whether the causal effects of RG and T2D on FVC are independent of possible confounders, such as physical activity and smoking. The same instrument selection criteria as described for the main MR analysis were used. CPD was instrumented by 54 (available out of the 58 in total) independent genome-wide significant variants, obtained from the GWAS discussed in ref. 58. LST served as a continuous proxy phenotype for physical activity from the recent study discussed in ref. 59 with 66 (available out of the 88 in total) independent genome-wide significant variants.

### PRS analysis
We tested the ability of the RG genetic effects to predict RG, T2D and HbA1c. We compared that to the predictive power of T2D and FG genetic instruments by computing PRS for RG, T2D and FG and assessing their performance in predicting RG, T2D and HbA1c. PRS analyses require base and target data from independent populations. The base datasets in our analyses were UKBB-only estimates from the present RG GWAS, meta-analysis estimates of 32 studies for T2D[15] and meta-analysis estimates from MAGIC for FG[64]. We used the second largest cohort, the Vanderbilt University Medical Center, as our target dataset. PRS construction and model evaluation (Supplementary Note) were done using the software PRSice v2.2.3 (ref. 99).

### Clustering of the RG signals with results for 45 other phenotypes
We looked up the $z$ scores (regression coefficient $\beta$ divided by the standard error) of the distinct 133 RG signals in publicly available summary statistics of 45 relevant phenotypes (Supplementary Table 23). All variant effects were aligned to the RG risk allele. HapMap 2-based summary statistics were imputed using SS-Imp v0.5.5 (ref. 70) to minimize missingness. Missing summary statistics values were imputed via mean imputation. The resulting variant–trait association matrix was truncated to 2 s.d. to minimize the effect of outliers. We used agglomerative hierarchical clustering with Ward's method to partition the variants into groups by their effects on the considered outcomes. The clustering analysis was performed in R using function hclust() from in-built stats package.

### Reporting summary
Further information on research design is available in the Nature Portfolio Reporting Summary linked to this article.

## Data availability
Meta-analysis summary statistics for the GWAS presented in this manuscript are available on the MAGIC website (magicinvestigators.org) and through the NHGRI-EBI GWAS Catalog (https://www.ebi.ac.uk/gwas/downloads/summary-statistics, GCP ID: GCP000666; with study accession codes for Europeans-only meta-analysis: GCST90271557; cross-ancestry meta-analysis: GCST90271558; and sex-dimorphic meta-analysis: GCST90271559). UK Biobank individual-level data can be obtained through a data access application available at https://www.ukbiobank.ac.uk/. In this study, we made use of data made available by: 1000 Genomes project (https://www.genome.gov/27528684/1000-genomes-project); SNPsnap (https://data.broadinstitute.org/mpg/snpsnap/index.html); Tabula Muris (https://www.czbiohub.org/tabula-muris/); GTEx Consortium (https://gtexportal.org/home/); microbiome GWAS (https://mibiogen.gcc.rug.nl/); Human Gut Microbiome Atlas (https://www.microbiomeatlas.org); eQTLGen Consortium (https://www.eqtlgen.org/); TIGER expression data (http://tiger.bsc.es/) and LDHub database (http://ldsc.broadinstitute.org/ldhub/).

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

## Acknowledgements

This research has been conducted using the UK Biobank Resource, project number 37685. We are supported by the following: the Medical Research Council (grants MR/L01341X/1 to P.E., MR/R010676/1 to B.J., MR/R010676/1 to A.T.); PHE (to P.E.); the UK Dementia Research Institute (to P.E. and I.T.); the Alzheimer's Society (to P.E. and I.T.); the Alzheimer's Research UK (to P.E. and I.T.); the National Health and Medical Research Council (NHMRC) Fellowship Schemes (552498 to B.B., 339446 and 619667 to G.M.); the NHMRC Ideas (grant 1184726 to P.M.S. and D.W.); the NHMRC (grants 1150083 to P.M.S. and D.W., 1154434 to P.M.S., 1155302 to D.W.); the Swedish Research Council (grants 2017-02688, 2020-02191 to E. Ahlqvist and 2019-01417 to M.d.H.); the Swedish Heart-Lung Foundation (grants 20200781 and 20200602 to M.d.H.); the British Heart Foundation (to A.G.), the European Commission (grants LSHM-CT-2007-037273 and HEALTH-F2-2013-601456 to A.G.); the Novo Nordisk Foundation (grants NNF15CC0018486 to A.G. and NNF18CC0034900 to T.H.P.); the Lundbeck Foundation (grant R190-2014-3904 to T.H.P.); VIAgenomics (grant SP/19/2/344612 to A.G.); the Wellcome Trust (grants 090532/Z/09/Z, 203141/Z/16/Z to H.W., 104955/Z/14/Z to A. David, 090532 to M.I.M., 098381 to M.I.M., 106130 to M.I.M., 203141 to M.I.M., 212259 to M.I.M., 205915/Z/17/Z to I.P.); UKRI Innovation-HDR-UK Fellowship (grant MR/S003061/1 to R.J.S.); European Union's Horizon 2020 research and innovation program LONGITOOLS (grant H2020-SC1-2019-874739 to M.A.K., A.U., Z.B. and I.P.); the European Foundation for the Study of Diabetes (to B.J. and M.A.K.); the Imperial Post-CCT Post-Doctoral Fellowship (to B.J.); the Academy of Medical Sciences (to B.J.); the National Institute for Health Research Imperial NIHR Biomedical Research Center (to B.J. and T.M.T.); the Engineering and Physical Sciences Research Council (to B.J.); the Society for Endocrinology (to B.J.); the British Society for Neuroendocrinology (to B.J.); Research England 'Expanding excellence in England' (to I.B.); the Research Foundation-Flanders (to V.L.); the Diabetes UK (to V. Salem, A.T.; BDA, 20/0006307 to I.P.); the Russian Science Foundation (grant 19-15-00115 to S.S.); the NIDDK (grant U01-DK105535 to M.I.M.); European Federation for the Study of Diabetes (to A.T.); the Agence Nationale de la Recherche (PreciDIAB, grant ANR-18-IBHU-0001 to J.G.M. and I.P.); the University of Lille mobility grant (to J.G.M.); the People-Centered Artificial Intelligence Institute, University of Surrey (Z.B., M.A.K., A. Demirkan and I.P.); the World Cancer Research Fund (to I.P.); the World Cancer Research Fund International (grant 2017/1641 to I.P.); the Royal Society (grant IEC\

R2\181075 to I.P. and C.A.R.); the European Union through the 'Fonds européen de développement regional' (FEDER; to I.P.); the 'Conseil Régional des Hauts-de-France' (Hauts-de-France Regional Council; to I.P.); the 'Métropole Européenne de Lille' (MEL, European Metropolis of Lille; to I.P.).

## Author contributions

These authors junior-led the study analyses and write-up: V.L., L.J. and A.U. Central analysis and writing group included: V.L., L.J., A.U., L.Z., K.S.G.G., Z.B., A.F., L.M., A.S., M.A.K., B.J. and I.P. Additional analyses were junior-led by: J.G.M., S.C., P.V.T., S.S., A. David, R.M., R.-M.R., E. Ahlqvist, Z.W., T.M.T., A.T. and V. Salem. GWAS cohort analyses were carried out by: G.T., H.G., E.E., B.B., R.A.S., A.I., J.H.Z., S.M.W., T.J., C.G., H.G., C.M., M.M.-N., R.J.S., A.G., D.R., J.D., Y.S.A. and M.A.K. Metabochip cohort analyses were undertaken by: E. Albrecht, A.U.J. and H.M.S. Cohort sample collection, genotyping, phenotyping or additional analyses were led by: I.R.C., E.F.-E., V. Steinthorsdottir, A.G.U., P.B.M., M.J.B., J.S., O.H., B.T., K.H., T.W., K.L.M., Z.W., M.d.H. and R.J.F.L. Metabochip cohort principal investigators were: W. Kratzer, M.H., W. Koenig and B.O.B. GWAS cohort principal investigators were: J.T., M.B., J.C.F., A. Hamsten, H.W., I.N., H.-E.W., M.J.C., K.T.K., C.M.v.D., A. Hofman, N.J.W., C.L., J.B.W., N.G.M., G.M., I.T., P.E., U.T., K.S., E.L.B. and J.B.M. Additional analyses senior leads were: P.M.S., D.W., L.G., G.D., A. Demirkan, T.H.P. and C.A.R. Senior authors who contributed to paper writing: I.B., C.S., M.I.M., P.F., J.D. and J.B.M. Senior author who contributed to analyses and was a member of the writing group: Y.S.A. Senior authors who led the study design, analyses and write-up were: M.A.K., B.J. and I.P.

## Competing interests

A.T. has received grant funding from Sun Pharmaceuticals and Eli Lilly. J.B.M. is an academic associate for Quest Diagnostics. They make an HbA1c assay. I.R.C. is an employee of New England Biolabs, a manufacturer and vendor of reagents for life science research. M.J.C. is Chief Scientist for Genomics England, a UK Government company. The views expressed in this article are those of the author(s) and not necessarily those of the NHS, the NIHR or the Department of Health. M.I.M. has served on advisory panels for Pfizer, Novo Nordisk and Zoe Global, has received honoraria from Merck, Pfizer, Novo Nordisk and Eli Lilly and research funding from Abbvie, AstraZeneca, Boehringer Ingelheim, Eli Lilly, Janssen, Merck, Novo Nordisk, Pfizer, Roche, Sanofi Aventis, Servier and Takeda. As of June 2019, M.I.M. is an employee of Genentech and a holder of Roche stock. P.M.S. received grant funding from Laboratoires Servier. P.M.S. and D.W. receive funding from Astex Pharmaceuticals and Novo Nordisk. They are both shareholders of Septerna, where P.M.S. is also a founder. P.M.S. is the director and D.W. the Monash Node leader of the Australian Research Council of Australia Center for Cryo-Electron Microscopy of Membrane Proteins that includes the following as Partner Organizations who provide cash or in-kind funding: Astex Pharmaceuticals, AstraZeneca, Boehringer Ingelheim, Catalyst Therapeutics, Dimerix Bioscience, Genentech, Novo Nordisk, Pfizer, Sanofi Aventis, Servier and Thermo Fisher Scientific. T.J. is now a GSK employee. W. Koenig reports consulting fees from AstraZeneca, Novartis, Pfizer, The Medicines Company, DalCor, Kowa, Amgen, Corvidia, Daiichi-Sankyo, Genentech, Novo Nordisk, Esperion, OMEICOS, LIB Therapeutics; speaker honoraria from Amgen, Novartis, Berlin-Chemie, Sanofi and Bristol-Myers Squibb; grants and nonfinancial support from Abbott, Roche Diagnostics, Beckmann and Singulex, all outside the submitted work. Y.S.A. is the owner of Maatschap PolyOmica and PolyKnomics BV, private organizations providing services, research and development in the field of computational and statistical, quantitative and computational (gen)omics. G.T., U.T. and K.S. are employees of deCODE genetics/Amgen. The other authors declare no competing interests.

## Additional information

**Extended data** is available for this paper at https://doi.org/10.1038/s41588-023-01462-3.

**Correspondence and requests for materials** should be addressed to Marika A. Kaakinen, Ben Jones or Inga Prokopenko.

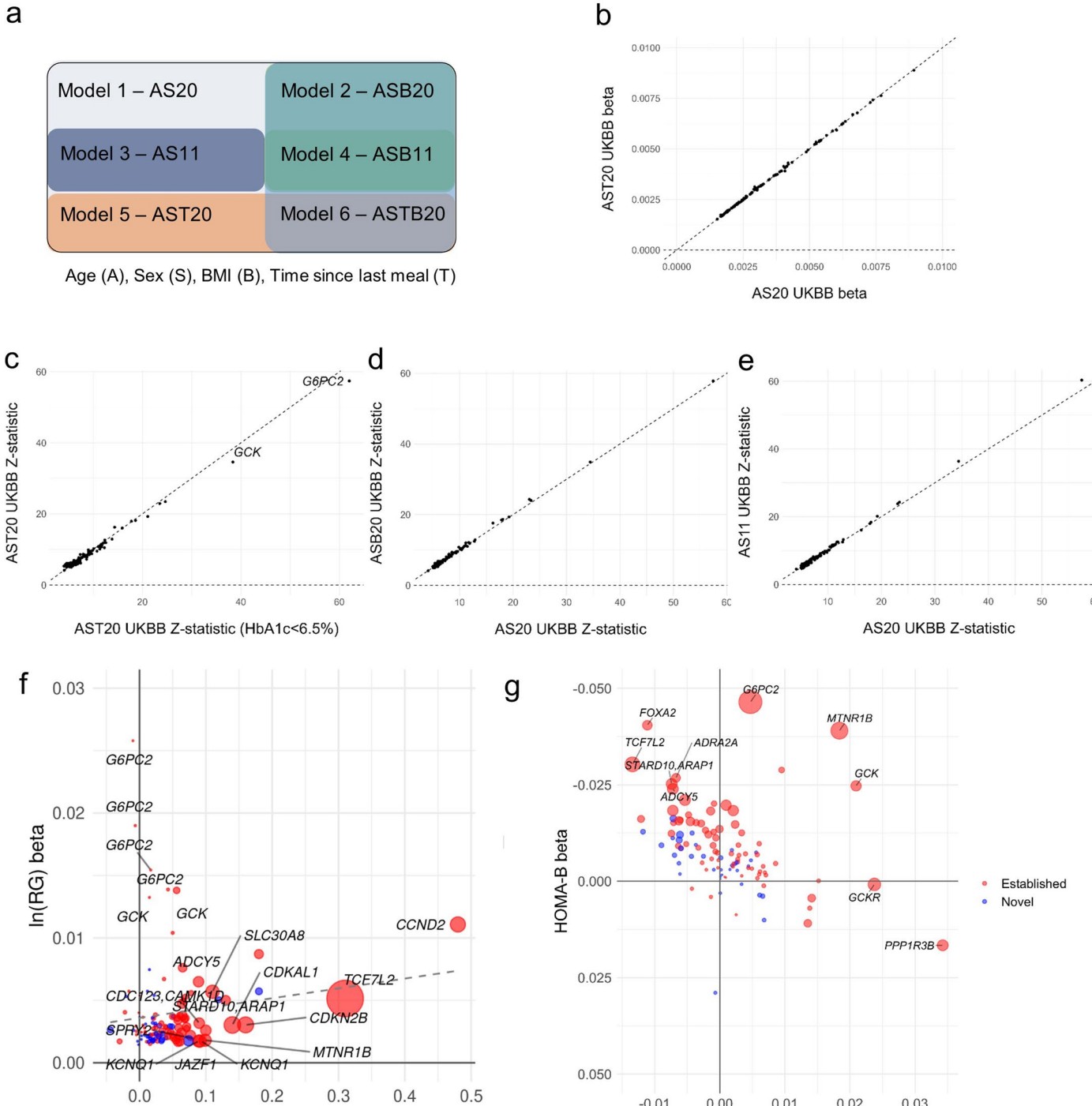

**Extended Data Fig. 1 | RG trait models tested and sensitivity plots showing the correlations between association analyses beta coefficients and Z-scores from RG models in UKBB. a**, The models were labeled according to covariates included and RG cut-offs used. Individuals were included based on two RG cut-offs: <20 mmol/l to account for the effect of extreme RG values (20) and <11.1 mmol/l (11), which is an established threshold for T2D diagnosis. Hence, model 1 – AS20 refers to adjustment for age and sex, using a cut-off of <20 mmol/l, and so forth. **b-e**, For **c**, 4,138 individuals were excluded based on HbA1c ≥ 6.5%, in addition to the self-reported or diagnosed T2D cases. Variants with a heterogeneity *P*-value ≤ 0.05 (beta-coefficient plot) or a *Z*-score difference between the two models compared >3 (*Z*-score plots) are annotated. **f**, An enrichment plot showing the effect of RG signals (AS20 + AST20 model) on T2D.

RG and T2D effect sizes are plotted along the *y*- and *x*-axes, respectively. Point size is proportional to the statistical significance of the variant for T2D, with red color indicating previously established signals and blue novel signals, respectively. The dashed line represents the line of best fit. Variants with T2D *P*-value in the lowest decile are labeled. **g**, An enrichment plot showing the effects of RG signals (AS20 + AST20 model) on HOMA-B and HOMA-IR. The effect sizes on HOMA-B and HOMA-IR are plotted along the *y*- and *x*-axes, respectively. Point size is proportional to the significance of the variant either in HOMA-B or HOMA-IR, depending on which trait has the smaller *P* value. Red color indicates previously established signals and blue indicates novel signals, respectively. Variants with suggestive significance ($P < 5.0 \times 10^{-6}$) are labeled.

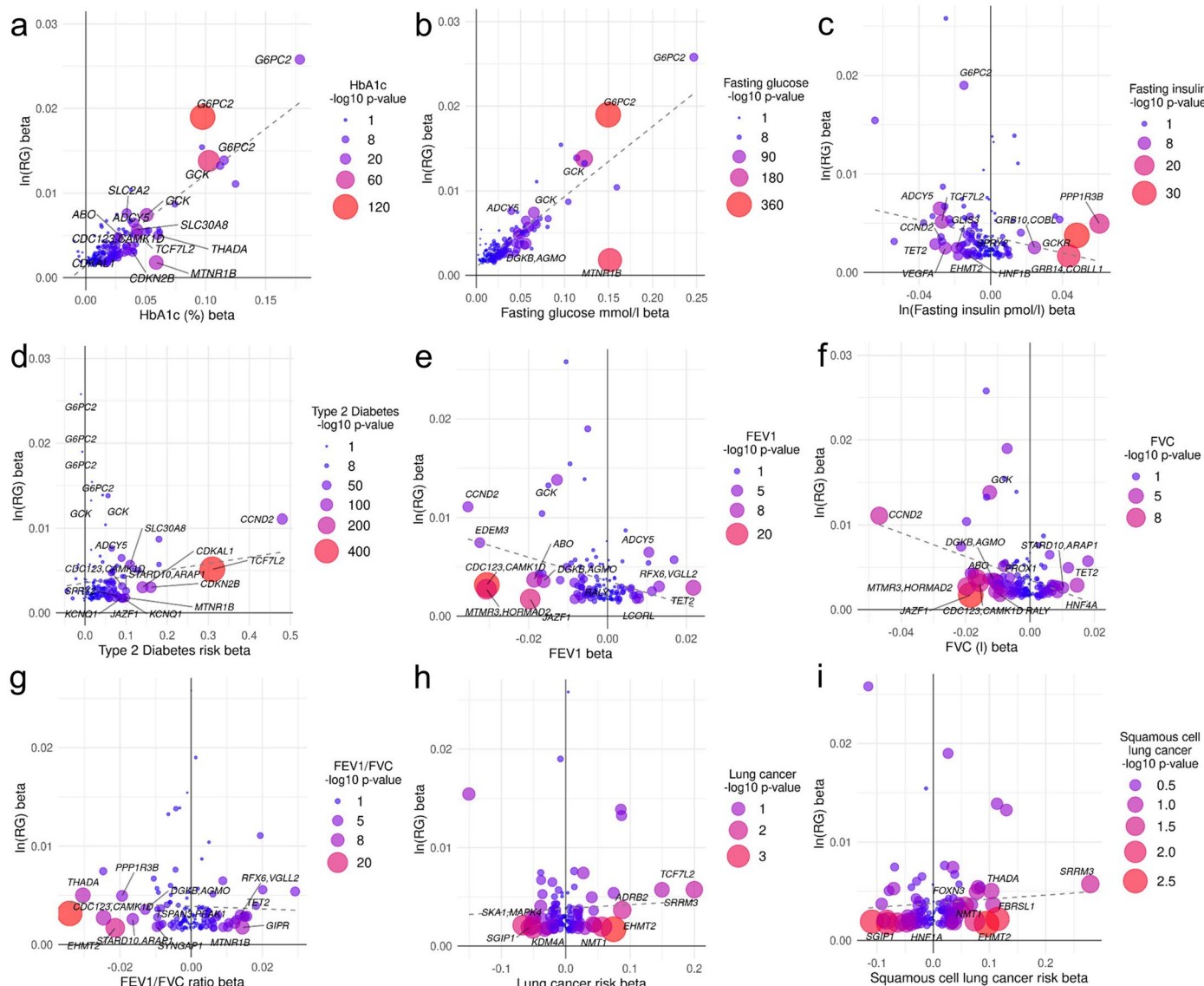

**Extended Data Fig. 2 | Enrichment plots showing the effect of RG signals (AS20 + AST20 model) on glycemic and respiratory-related phenotypes. a-i**, Look-up of effects was done in previously published genome-wide association studies for HbA1c (**a**), fasting glucose (**b**), fasting insulin (**c**), type 2 diabetes (**d**), forced expiratory volume in one second (FEV1) (**e**), forced vital capacity (FVC) (**f**), FEV1/FVC (**g**), lung cancer (**h**) and squamous cell lung cancer (**i**). RG and other phenotype effect sizes are plotted along the *y*- and *x*-axes, respectively. Point size and color are proportional to the significance of the variant in each phenotype, with red indicating higher and blue lower significance, respectively. The dashed line represents the line of best fit. $P < 5.0 \times 10^{-8}$ was considered statistically significant after adjusting for multiple testing. Two-tailed *P*-values are reported. Variants with *P*-values in the lowest decile are labeled.

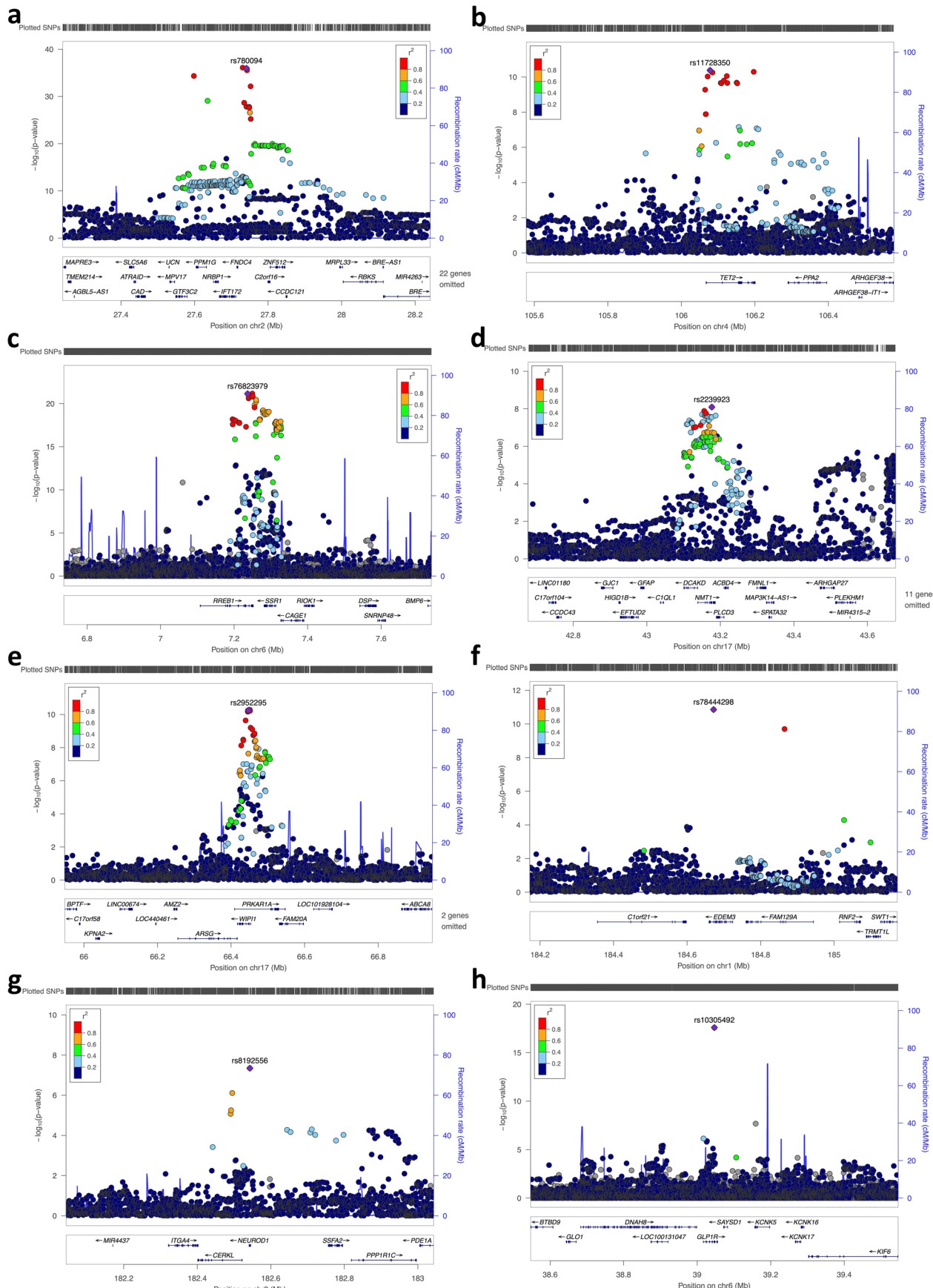

**Extended Data Fig. 3 | LocusZoom plots of common variants in UKBB (Europeans) meta-analysis for RG. a-h**, Plots are shown for *GCKR* (**a**), *TET2* (**b**), *RREB1* (**c**), *NMT1* (**d**) and *WIPI1* (**e**) loci and low-frequency coding variants at *EDEM3* (**f**), *NEUROD1* (**g**) and *GLP1R* (**h**) loci. The *x*-axis shows the chromosomal position, and the *y*-axis shows the uncorrected two-sided −log₁₀ *P* values from the UKBB GWAS conducted using linear mixed-modeling in BOLT-LMM. Horizontal line corresponds to $P = 5 \times 10^{-8}$ and blue peaks show the recombination rate.

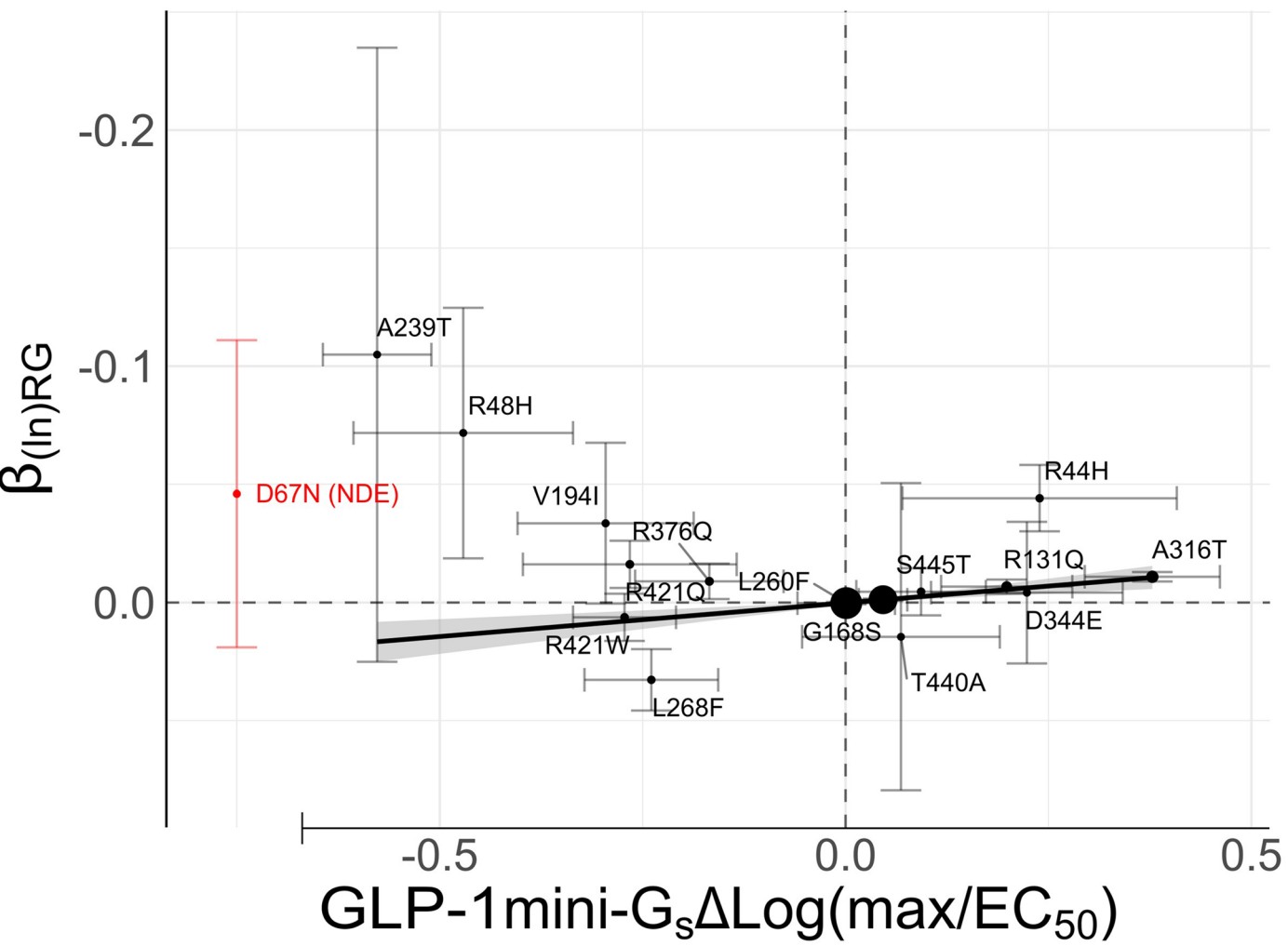

**Extended Data Fig. 4 | Association analysis of GLPR1 receptor function and random glucose effects of coding variants.** Minor allele frequency-weighted linear regression was used to test if mini-$G_s$ response to GLP-1 stimulation significantly predicted point estimates of GLP1R variant effect on RG levels (AST20 $\beta_{RG}$ as estimated in whole-exome sequencing data from the UKBB study). Mini-$G_s$ response to GLP-1 stimulation was corrected for variant surface expression ($n_{max}$ = 22, exact $n$ for each variant is provided in Supplementary Table 11). Error bars extend one standard error above and below the point estimate. Size of the dots is proportional to the weight applied in the regression model (Methods). The regression results (coefficient of determination $R^2$ = 0.56, $F(1, 14)$ = 20.1, $P$ = 5.2 × 10$^{-4}$) suggest that mini-$G_s$ coupling in response to GLP-1 stimulation predicts the effect of these coding variants on RG levels (AST20 $\beta_{RG}$ = − 0.028; 95% CI = −0.042 to −0.015; $P$ = 5.2 × 10$^{-4}$). The gray shaded area around the regression line corresponds to the 95% confidence interval of predictions from the model. Variants in red showed no detectable surface expression (NDE) and are not included in regression analysis.

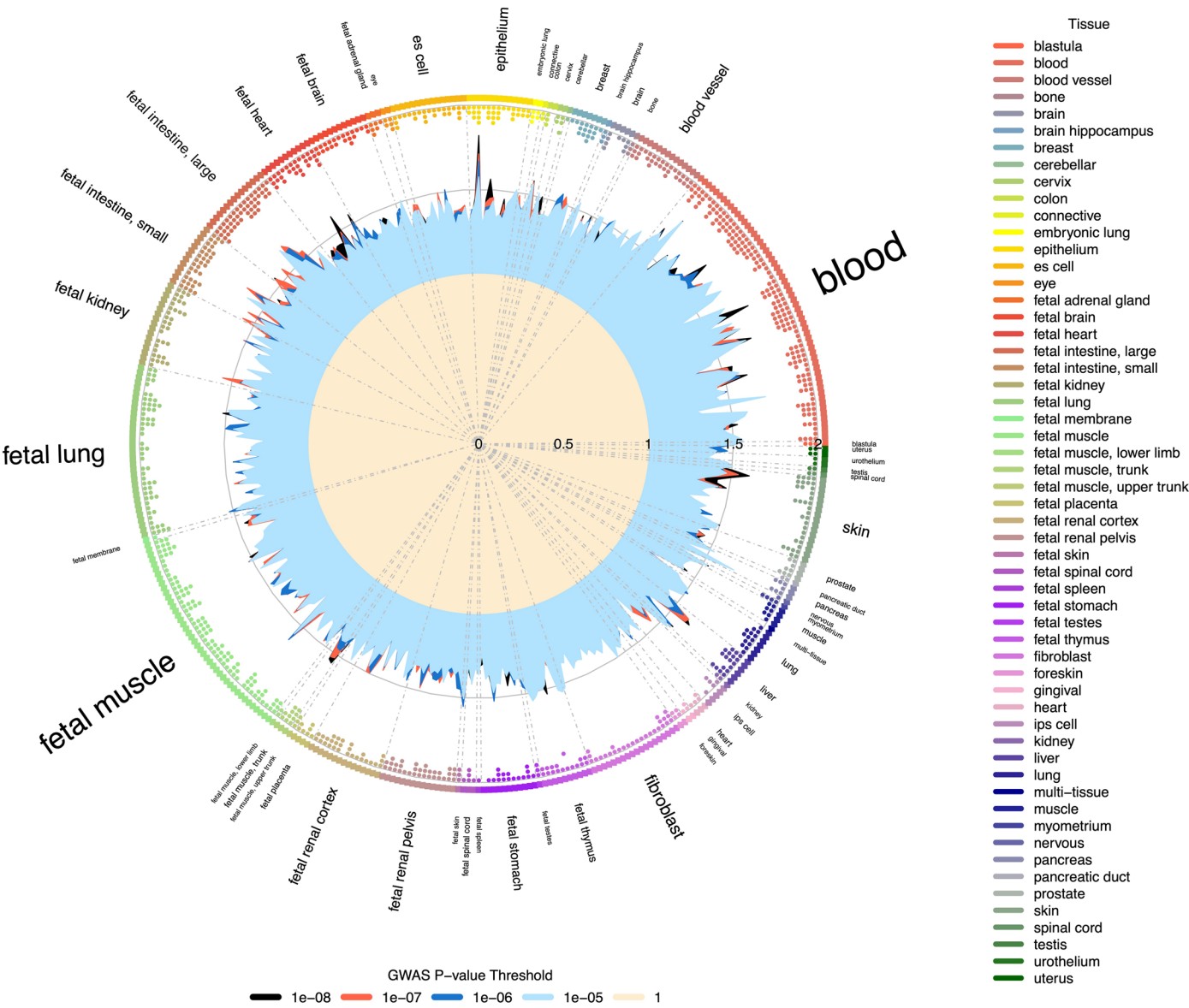

**Extended Data Fig. 5 | Epigenetic annotation of the RG GWAS results using GARFIELD.** The analyses were performed using generalized linear modeling in GARFIELD software. We considered enrichment to be statistically significant if the RG GWAS $P$-value reached $P = 1 \times 10^{-8}$ and the enrichment analysis P-value was $< 2.5 \times 10^{-5}$ (Bonferonni corrected for 2,040 annotations).

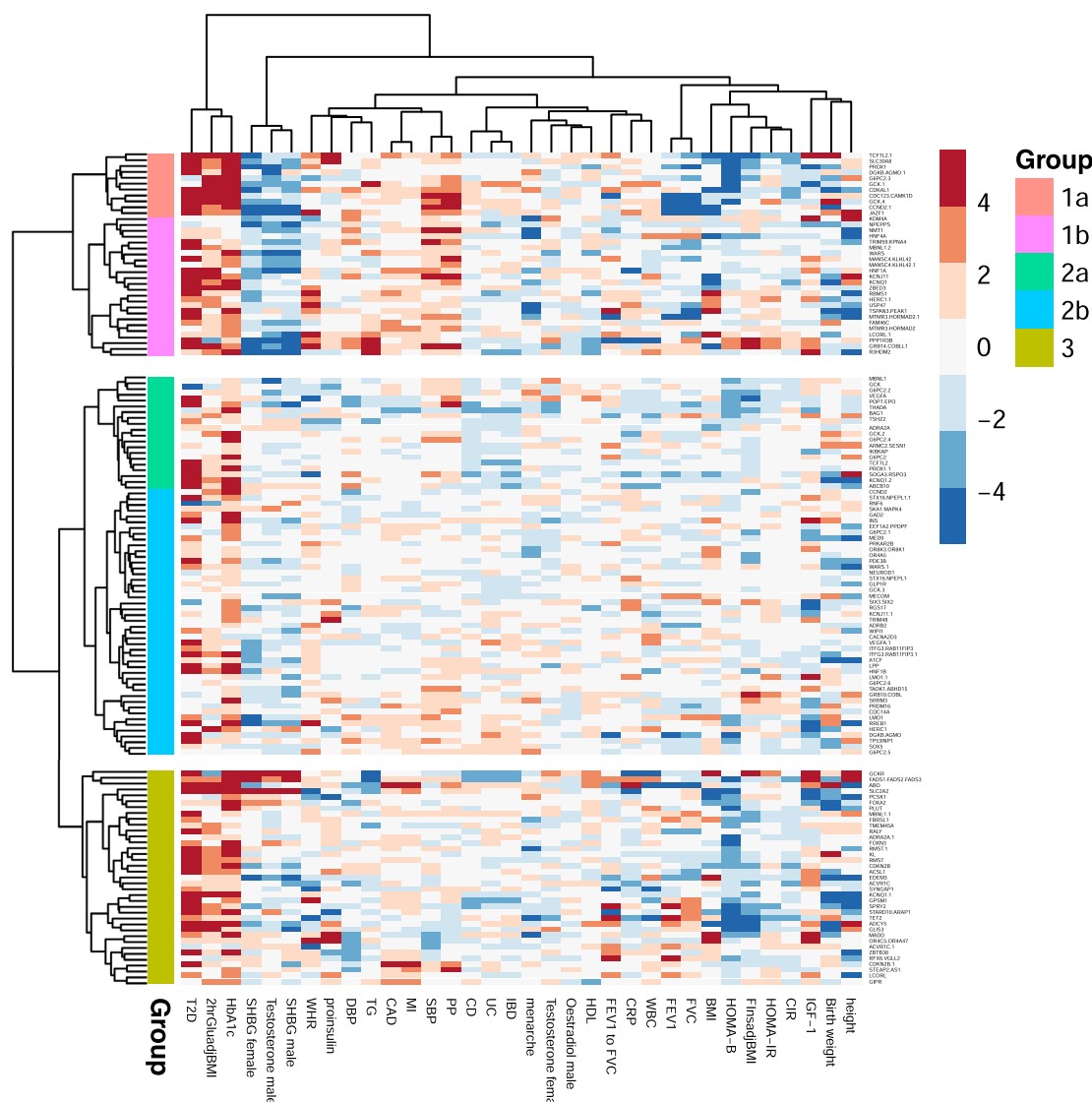

**Extended Data Fig. 6 | Cluster analysis of effects (as *Z*-scores) of the distinct 143 RG signals on 45 relevant phenotypes.** All variant effects were aligned to the RG risk allele. HapMap2 based summary statistics were imputed using SS-Imp v0.5.565 to minimize missingness. Missing summary statistics values were imputed via mean imputation. The heatmap was produced using the Pheatmap package. For visualization, the *Z*-scores were truncated to the value corresponding to genome-wide significance (*Z* = 5.45), and 11 phenotypes with the lowest median absolute *Z*-scores were excluded.

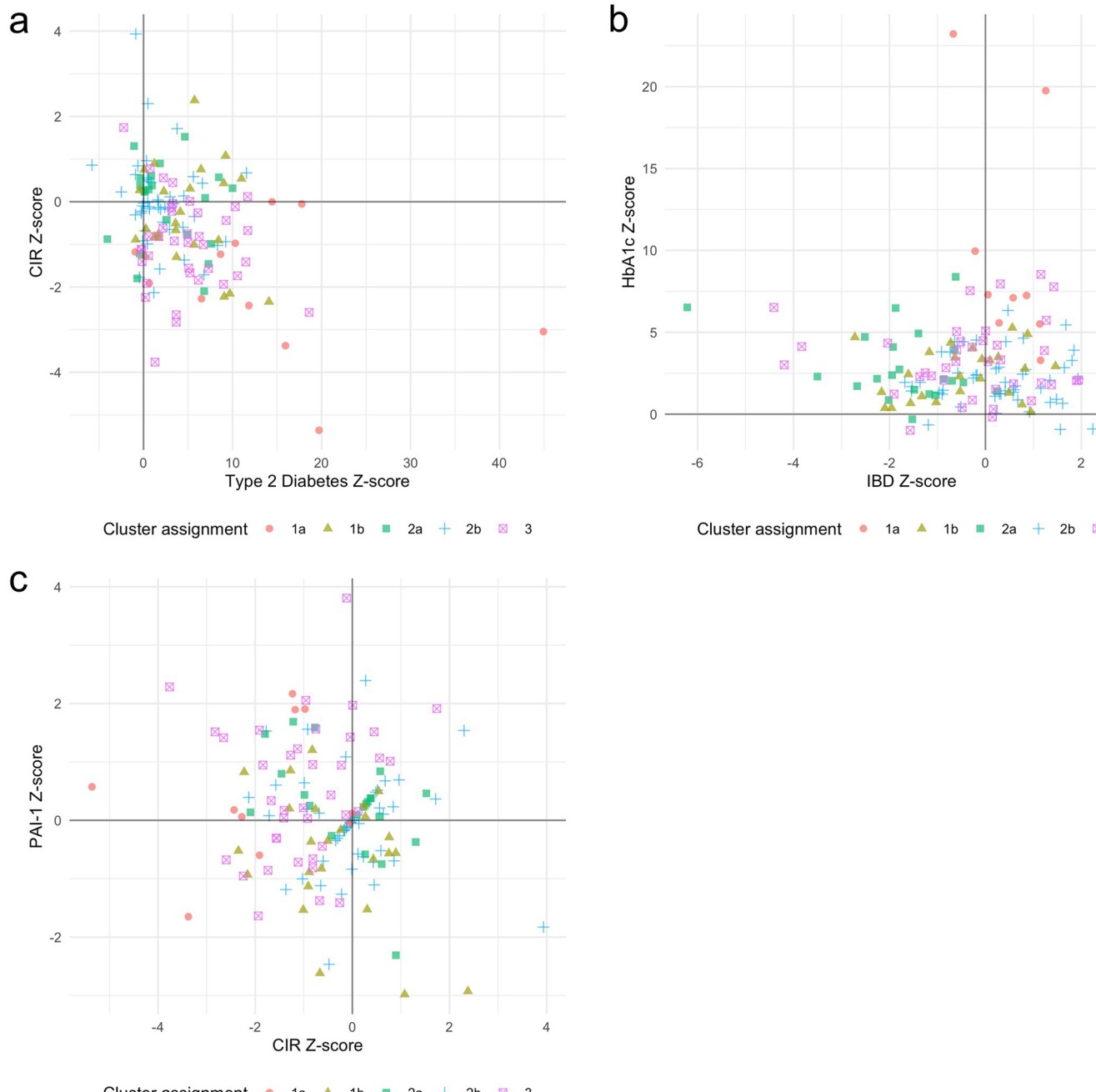

**Extended Data Fig. 7 | Scatter plots of the standardized allelic effect estimates for selected trait pairs.** In each scatter plot, loci were assigned to the groups defined from the cluster analysis and highlighted by different colors. **a**, Corrected insulin response (CIR) vs. type 2 diabetes (T2D) (clusters 1a/b related to metabolic syndrome). **b**, Glycated hemoglobin (HbA1c) vs. inflammatory bowel disease (IBD) (cluster 2a) highlights the effects of loci with a protective role in IBD. **c**, Plasminogen activator inhibitor-1 (PAI-1) vs. CIR (cluster 3) highlights loci linked to insulin secretion defects.

# Reporting Summary

## Statistics

For all statistical analyses, confirm that the following items are present in the figure legend, table legend, main text, or Methods section.

| n/a | Confirmed | |
|---|---|---|
| ☐ | ☒ | The exact sample size (*n*) for each experimental group/condition, given as a discrete number and unit of measurement |
| ☐ | ☒ | A statement on whether measurements were taken from distinct samples or whether the same sample was measured repeatedly |
| ☐ | ☒ | The statistical test(s) used AND whether they are one- or two-sided<br>*Only common tests should be described solely by name; describe more complex techniques in the Methods section.* |
| ☐ | ☒ | A description of all covariates tested |
| ☐ | ☒ | A description of any assumptions or corrections, such as tests of normality and adjustment for multiple comparisons |
| ☐ | ☒ | A full description of the statistical parameters including central tendency (e.g. means) or other basic estimates (e.g. regression coefficient) AND variation (e.g. standard deviation) or associated estimates of uncertainty (e.g. confidence intervals) |
| ☐ | ☒ | For null hypothesis testing, the test statistic (e.g. *F*, *t*, *r*) with confidence intervals, effect sizes, degrees of freedom and *P* value noted<br>*Give P values as exact values whenever suitable.* |
| ☒ | ☐ | For Bayesian analysis, information on the choice of priors and Markov chain Monte Carlo settings |
| ☒ | ☐ | For hierarchical and complex designs, identification of the appropriate level for tests and full reporting of outcomes |
| ☐ | ☒ | Estimates of effect sizes (e.g. Cohen's *d*, Pearson's *r*), indicating how they were calculated |

*Our web collection on statistics for biologists contains articles on many of the points above.*

## Software and code

Policy information about availability of computer code

| | |
|---|---|
| Data collection | No software was used for the data collection. |
| Data analysis | The software and tools used in individual GWAS can be found in Supplementary Table 1 with participating studies characteristics. Software/tools/algorithms included: Minimac2, MACH v1.0, IMPUTE v0.3.1/v1.0.0/v2.3.2/v4.1.2, SNPTEST v1.1.5/v2.5.1, EMMAX vbeta-07Mar2010, LMEKIN v1.8 (R package), Merlin v1.1.2, STATA v11, ProbABEL v0.4.3, BOLT-LMM v2.3, SS-Imp v0.5.5, METAL v2011-03-25, GWAMA v2.1, PLINK v1.07/1.90, GCTA v1.93.0, BaSiC v1, AceMD3 3.3.0, PHANTAST v1, AquaMMapS v1, DEPICT v1_rel 194, CELLECT v1.0.0, CELLEX v1.0.0, MetaXcan (S-PrediXcan) v0.6.10, GARFIELD v2, COLOC2, LDSC v1.0.0, TwoSampleMR v0.5.4 (R package), PRSice v2.2.3, LDlinkR v1.1.2 library, MVMR v0.3 (R package), Prism 8.0 (GraphPad Software), IDTxGen Exome Research Panel v1.0. |

For manuscripts utilizing custom algorithms or software that are central to the research but not yet described in published literature, software must be made available to editors and reviewers. We strongly encourage code deposition in a community repository (e.g. GitHub). See the Nature Portfolio guidelines for submitting code & software for further information.

## Data

Policy information about **availability of data**

All manuscripts must include a **data availability statement**. This statement should provide the following information, where applicable:

- Accession codes, unique identifiers, or web links for publicly available datasets
- A description of any restrictions on data availability
- For clinical datasets or third party data, please ensure that the statement adheres to our **policy**

> Meta-analyses summary statistics for the GWAS presented in this manuscript are available on the MAGIC website (magicinvestigators.org) and through the NHGR1-EBI GWAS Catalog (https://www.ebi.ac.uk/gwas/downloads/summary-statistics, GCP ID: GCP000470). UK Biobank individual-level data can be obtained through a data access application available at https://www.ukbiobank.ac.uk/. In this study we made use of data made available by: 1000 Genomes project https://www.genome.gov/27528684/1000-genomes-project; SNPsnap https://data.broadinstitute.org/mpg/snpsnap/index.html; Tabula Muris https://www.czbiohub.org/tabula-muris/; GTEx Consortium https://gtexportal.org/home/; microbiome GWAS https://mibiogen.gcc.rug.nl/; Human Gut Microbiome Atlas https://www.microbiomeatlas.org; eQTLGen Consortium https://www.eqtlgen.org/; TIGER expression data http://tiger.bsc.es/; LDHub database http://ldsc.broadinstitute.org/ldhub/.

## Human research participants

Policy information about **studies involving human research participants and Sex and Gender in Research.**

| Reporting on sex and gender | As a first step, we have fitted several models in the six cohorts, available to us initially for the modeling of RG, i.e. to identify the relevant set of covariates (including sex) as well as the necessary transformation for RG to be used across all the datasets in the GWAS meta-analysis. We then evaluated six different models in GWAS according to covariates included and cut-offs used: 1) age (A) and sex (S), RG<20 mmol/L (AS20), 2) age, sex and BMI (B), RG<20 mmol/L (ASB20), 3) age and sex, RG<11.1 mmol/L (AS11), 4) age, sex and BMI, RG<11.1 mmol/L (ASB11), 5) age, sex, T, T2 and T3, RG<20 mmol/L (AST20) and 6) age, sex, T, T2 and T3 and BMI, RG<20 mmol/L (ASTB20). To evaluate sex-dimorphism in our results, we meta-analyzed the UKBB and the Vanderbilt cohort with the GMAMA software, which provides a 2 degrees of freedom (df) test of association assuming different effect sizes between the sexes. We considered a signal to show evidence of sex-dimorphism if the 2 df test P-value was <5x10-8 and if the sex heterogeneity P-value (1 df) was <0.05. |
|---|---|
| Population characteristics | Analyses were conducted on non-diabetic females and males of European ancestry and additionally in non-European populations within UKBB (Black, Indian, Pakistani and Chinese). Within each study, individuals were included based on two RG cut-offs: <20 mmol/l (20) to account for the effect of extreme RG values and <11.1 mmol/l, which is an established threshold for T2D diagnosis. Age distribution and percentage for each gender varied between studies. More detailed description of each study collection is provided in Supplementary Table 1, and for the UK Biobank in https://www.ukbiobank.ac.uk/. |
| Recruitment | The majority of studies are population-based cohorts, case-control or family-based studies with related individuals. Subjects were men and women of European, Black, Indian, Pakistani or Chinese ancestry with no diagnosed diabetes. |
| Ethics oversight | No ethical approval was required for the study as it is a meta-analysis of summary statistics obtained from studies that each had ethical approval by local research ethics committees and written consent was obtained from all study participants. Further details about each study ethics approval can be found in the references and websites provided in Supplementary Table 1. |

Note that full information on the approval of the study protocol must also be provided in the manuscript.

# Field-specific reporting

Please select the one below that is the best fit for your research. If you are not sure, read the appropriate sections before making your selection.

☒ Life sciences  ☐ Behavioural & social sciences  ☐ Ecological, evolutionary & environmental sciences

For a reference copy of the document with all sections, see **nature.com/documents/nr-reporting-summary-flat.pdf**

# Life sciences study design

All studies must disclose on these points even when the disclosure is negative.

| Sample size | We aimed to bring together the largest possible sample size for RG with the following collection of samples: (i) 37,239 individuals from 10 European ancestry GWAS imputed up to the HapMap 2 reference panel; (ii) 3,156 individuals from three European ancestry GWAS with Metabochip coverage; (iii) 21,083 individuals from two European ancestry GWAS imputed up to 1000 Genomes reference panel; (iv) 380,432 individuals of white European ancestry from the UKBB, and; (v) 16,983 individuals from the Vanderbilt cohort imputed to the HRC panel Non-European UKBB populations included in the analyses had a sample size of at least 1,500 individuals. These were Black (N=7,644), Indian (N=5,660), Pakistani (N=1,747) and Chinese (N=1,503). Therefore, exact sample size was not predetermined. Maximum sample size was achieved by including all cohorts with RG available. |
|---|---|

| | |
|---|---|
| Data exclusions | Individuals were excluded from the analysis, if they had a diagnosis of T2D or were on diabetes treatment (oral or insulin). Individual studies applied further sample exclusions, including pregnancy, fasting plasma glucose equal to or greater than 7 mmol/l in a separate visit, when available, and having Type 1 Diabetes. Detailed descriptions of study-specific RG measurements are given in Supplementary Table 1. Low-quality SNPs were excluded by the following criteria: call rate <0.95, minor allele frequency (MAF) <0.01, minor allele count <10, Hardy-Weinberg P-value <10−4. After imputation of the GWAS meta-analysis summary statistics, imputed SNPs up to 1000 Genomes reference panel with imputation quality score < 0.7 were excluded. For the GWAS of the UKBB data, non-white non-European individuals and those with discrepancies in genoped and reported sex were excluded. Furthermore, in the UKBB GWAS, variants with MAF<=1% and imputation quality<=0.4 were excluded. |
| Replication | We have assessed the robustness of our RG meta-analysis findings by comparing the direction of effect of 133 signals, detected in the European subset between the UK Biobank (UKBB, 83.8% of the total study size) and other RG contributing studies grouped together. We further extended the check between UKBB results and meta-analysis of other RG contributing cohorts together with fasting glucose GWAS excluding overlapping cohorts (roughly 1/3 of UKBB sample size). Results from these comparisons are presented in Supplementary Table 3. Additionally, we have selected a list of additional distinct signals by carrying out approximate conditional analysis with GCTA and direct conditional analyses on UKBB genotypes with BOLT-LMM. To ensure pharmacological assay results were reproducible, all assays were repeated at least 4 times. The exact number of biological replicates is provided in the relevant supplementary table, along with the relevant measure of dispersion. |
| Randomization | This study meta-analyzed existing data. Therefore, there were no experimental groups and no randomization was required. |
| Blinding | GWAS is a hypothesis-free approach, so in each study contributing to the meta-analysis, researchers assessing glycemic traits, such as random glucose, were blinded to the genotypes that are associated with these outcomes. |

# Reporting for specific materials, systems and methods

We require information from authors about some types of materials, experimental systems and methods used in many studies. Here, indicate whether each material, system or method listed is relevant to your study. If you are not sure if a list item applies to your research, read the appropriate section before selecting a response.

## Materials & experimental systems

| n/a | Involved in the study |
|---|---|
| ☒ | Antibodies |
| ☐ | ☒ Eukaryotic cell lines |
| ☒ | Palaeontology and archaeology |
| ☒ | Animals and other organisms |
| ☒ | Clinical data |
| ☒ | Dual use research of concern |

## Methods

| n/a | Involved in the study |
|---|---|
| ☒ | ChIP-seq |
| ☒ | Flow cytometry |
| ☒ | MRI-based neuroimaging |

## Eukaryotic cell lines

Policy information about cell lines and Sex and Gender in Research

| | |
|---|---|
| Cell line source(s) | Flp-In T-REx-293 cells were obtained from Thermo Fisher. |
| Authentication | The Flp-In T-REx-293 cell line were indirectly authenticated by successful integration of the WT/variant GLP1R insert. |
| Mycoplasma contamination | Cells tested negative for mycoplasma. |
| Commonly misidentified lines (See ICLAC register) | *Name any commonly misidentified cell lines used in the study and provide a rationale for their use.* |

