## [Peer Review File · Nature Genetics]

Peer Review Information

Manuscript Title: Genome-wide association analyses of random glucose levels in 493,036 individuals provide insights into diabetes pathophysiology, complications and treatment stratification.

Corresponding author name(s): Prof Inga Prokopenko, Dr Ben Jones, Dr Marika Kaakinen

Reviewer Comments & Decisions:

Decision Letter, first revision:

18th March 2022

Dear Inga,

Your Article "Random glucose GWAS in 493,036 individuals provides insights into diabetes pathophysiology, complications and treatment stratification" has been seen by two referees. You will see from their comments below that, while they find your work of interest, they have raised several relevant points. We are interested in the possibility of publishing your study in Nature Genetics, but we would like to consider your response to these points in the form of a revised manuscript before we make a final decision on publication.

To guide the scope of the revisions, the editors discuss the referee reports in detail within the team, including with the chief editor, with a view to identifying key priorities that should be addressed in revision, and sometimes overruling referee requests that are deemed beyond the scope of the current study. In this case, we ask that you address all technical points related to the association and Mendelian randomization analyses, improve the presentation of the directional pleiotropy analyses, extend the molecular QTL colocalization analyses, and revise the presentation of the functional studies for clarity. We hope you will find this prioritized set of referee points to be useful when revising your study. Please do not hesitate to get in touch if you would like to discuss these issues further.

We therefore invite you to revise your manuscript taking into account all reviewer and editor comments. Please highlight all changes in the manuscript text file. At this stage, we will need you to upload a copy of the manuscript in MS Word .docx or similar editable format.

*2) If you have not done so already please begin to revise your manuscript so that it conforms to our Article format instructions, available [here](http://www.nature.com/ng/authors/article_types/index.html). Refer also to any guidelines provided in this letter.

*3) Include a revised version of any required Reporting Summary: <https://www.nature.com/documents/nr-reporting-summary.pdf>

[redacted]

We hope to receive your revised manuscript within 8-12 weeks. If you cannot send it within this time, please let us know.

Sincerely,
Kyle

Kyle Vogan, PhD
Senior Editor
Nature Genetics
<https://orcid.org/0000-0001-9565-9665>

Referee expertise:

Referee #1: Genetics, diabetes, complex traits

Referee #2: Genetics, cardiovascular diseases, clinical translation

Reviewers' Comments:

Reviewer #1:

Remarks to the Author:

This paper describes a large GWAS meta-analysis of random glucose in almost 500K samples from 17

studies, of which the UK Biobank was the largest group (~400K). The authors carry out a wide range of interesting analyses that add to our understanding of glucose genetics. The manuscript is well written, and easy to follow despite the large number of conducted analyses. The results are novel and original, and used contemporary methods for analysis. I have the following comments and suggestions for the authors' consideration, mainly concerning secondary signal, rare variant and directional pleiotropy analysis:

The authors very clearly distinguish lead loci from secondary loci (142 loci and 185 signals) in abstract and main results. What I did find confusing is how the number of novel SNPs for glycemic traits were counted (ST3). I have tried regenerate the number 84 novel SNPs number and was unable to, and noticed a couple of peculiarities to the table which are listed below. Foremost, it seems that novelty is noted for both primary loci and secondary signals. I would refrain from using secondary signals for novelty, because they were generated by an approximate method (GCTA on summary statistics) and not formal method (stepwise regression with individual level data). For example, it seems that 16 SNPs in G6PC2 locus seems are being counted 7 times as novel? Perhaps the authors can clarify what they exactly counted as being novel. Their current approach seems generous. GCTA is not a formal method but approximate method, and depending on the data set (and unfortunately even various parameters chosen within its own software) will identify different SNPs. Given that the UK Biobank is the predominant sample (400K of 500K participants), can the authors perform formal secondary signal analysis using stepwise linear regression, where they include the lead SNP as a covariate in the analysis, and then repeat until there are no SNPs left that exceed set significance? From personal experience, GCTA produces vastly different results from individual level conditional analysis. Notwithstanding, I would limit the counting of novel SNP based on the loci (not distinct signals).

Rare variant analysis: On line 248 the authors state that they used the squared correlation coefficient to identify rare variants as being independent ($r^2_{1000GenomesAllEthnicities} < 0.001$). The authors state that the lack of LD between SNPs was chosen to indicate independent signal. Similar to my issue with secondary signals, can the authors perform a conditional analysis and adjust for the lead SNP to empirically confirm independence of the rare variant signal? A low r^2 can still imply a high D' especially when evaluating rare variants where differences in minor allele frequencies contribute to a large reduction in the correlation coefficient with the usually common lead variant.

The authors have performed a directional pleiotropy analysis, of which the z-scores are available in Supplementary Table 23 and Supplementary figure 6A (and not 5A as the authors refer to in the text). The hierarchical cluster plot is not interpretable, the rows and columns are unreadable, and the figure is more or less a yellow blob. I was looking forward to examining that figure, and if executed correctly would be a great main figure to the manuscript. My approach would be the following: While I think the z-scores a great input to denote snp-trait association strength, the extreme minima (e.g. z-score of -33 of GCKR with triglycerides) and maxima (z-score of 44 for TCF7L2 for T2D) currently force all other associations to look relatively 'non-relevant' (e.g. yellow, please reconsider to white color) because they bleak in comparison to the peak associations. Please consider using a cut-off for minima and maxima so that a more nuanced picture of directional pleiotropy may emerge. Limiting to z-score (to for example 5.45, corresponding to $P < 5.10^{-8}$) and categorizing the colors into dark red, red, light red, white, light blue, blue, dark blue (7 categories). Finally, some traits don't seem to be that related to the lead SNPs with exception of an incidental association here or there. It might be an idea to remove them from the figure because they don't contribute to the visualization and cluster assignment. For more customized hierarchical clustering in R, the authors could look into R package 'pheatmap' rather than using hclust().

Minor:

line 201: Adjustment for last meal timing attenuated some effect estimates for 5 loci? Or is this just a selection? Then indicate the total number of loci that were affected. Does the adjustment for meal timing, increase effect estimates for some SNPs?

Line 214: Can the authors specify if this is nominal significance ($P < 0.05$) without or without directional concordance.

Line 233: male-specific and female-specific instead of men-specific and women-specific

Out of 748 rare variants associated with RG above genome-wide 567 significance in the HRC-imputed UKBB data. Is this genome-wide?

AS20+AST20 – not directly clear from the description what exactly has been done here?

Supp Table 2: can the authors indicate which of the models were the final AS20, AST20, AS11, AST11, ASTB11, ASTB20

Supp Table 3: either separate out the independent from the conditionally independent variants, or reconsider limiting novelty assignment to primary SNPs.

rs538547285, in G6PC2 locus, unless independent from lead locus by formal conditional analysis (not approximate), it should be part of locus 14 and not locus 122.

Duplication of rs2800734, and other sex-dimorphic SNPs. Instead of a duplicate SNP entry, maybe add a flag for sex-dimorphism?

rs74832478 entry is confusing. (primary locus 115 and 140, counted twice)

rs147493041 and rs149594447 are listed as locus 126 and locus 127, whereas they are in complete LD in all ancestry 1000 G all ancestries

rs543933126 locus 128 could be argues its part of locus 43 based on distance.

ST 7: orient towards the minor allele, instead of having EAF of 99.999%. Also, adjust for the lead SNP in order to find out whether the rare allele signal is independent from the lead signal.

ST 8 and ST 9: r^2 with lead SNP is always 1, can the authors double-check the veracity of that column.

ST 15a: column name: RG GWAS significant? What does this mean?

ST16: can the authors specify in which tissue the colocalization result is referring to?

Reviewer #2:
Remarks to the Author:

Random glucose GWAS in 493,036 individuals provides insights into diabetes pathophysiology, complications and treatment stratification. Lagou et al

Summary of findings

Random glucose 493K (only 16K non-European), non-diabetic

142 loci, 185 signals

14 sex dimorphic effects

9 through trans ethnic

25 rare variants

underappreciated role of gut

lung function modulated by glucose

Nice to read this paper from a very good group and PI who have been leaders in the field. Overall, I liked the paper but felt it could be improved organizationally and honed down to key messages. It seems like there were many, many different analyses performed and there was insufficient detail on some things (for instance functional assays) and insufficient justification and analyses for others (microbiome). I also had some specific questions.

Questions

Are the glucose levels really random? Is there a plot of the time of day the glucose labs were drawn? Suspect these are almost all between 8am-5pm. They did have info on time from last meal on some which is important.

However, did time from last meal systematically differ by sex or race/ethnicity? This could be a confounder and could reflect occupation or other socio-economic factors rather than underlying biology.

How do you deal with correlations between diet and glucose given confounders? For example, people that tend to eat dense caloric diets especially if they eat more frequently are likely to have higher glucose levels but also higher obesity rates and may be more sedentary. This is not necessarily bad, but certainly is a limitation or caveat.

It can be hard to link loci to genes (ITPR3, RREB1 are examples of loci highlighted in one section of the paper etc) speculatively just based on distance to nearest gene. As the authors hint at (in the functional section) colocalization can be useful in narrowing down the list of causal genes. For all the RG non-coding variants identified, is there colocalization data to support a causal relationship for genes within associated loci (based on eQTL or sQTL RNASeq data from pancreas, gut, fat, brain etc)? The methods section on eQTL seems to indicate that only eQTL data from whole blood was used which seems suboptimal given other publicly available data on metabolically relevant tissues. See recent papers from Mark McCarthy, Anna Gloyn and Stephen Montgomery for instance.

Functional and structural characterisation was also done for several genes including GLP1R which was interesting data. However, the methods for this were not even introduced on p12 making it hard to follow ("our functional approach")...I was left wondering what was the system used for these experiments? Would be best at least introduced here rather than exclusively in the methods. The methods section for these is also a bit hard to follow in terms of critical items. It indicates that both transient or stable cell lines were used but not why this was the case or for which experiments which stable vs transient lines were used. For the mini-g recruitment assays it is not clear what kind of cells were used in the stable lines (lines 604-604). Were these also HEK cells? Same is true for lines

621-622. I also think this Results section, while interesting, does not naturally flow in terms of how it is organized in the paper. I understand the desire to connect these analyses to a more clinically relevant issue and GLP1R targeting drugs are really "hot" right now. But it seemed out of place in how it was introduced and presented.

The microbiome associations p 15 seem almost an afterthought and are not well developed. Pretty speculative. If more is not done, would remove.

The relationship of RG related traits with other traits (cancer, lung function) are interesting. Phenotypes associated with RG (like obesity) are obvious candidates to be associated with lung function. Things that are associated with lung function could obviously impact RG through modification of exercise (people with lower FEV1 may choose to exercise less and therefore get higher RG). In this way the MR analyses are important. Did the RG variants used in the MR analyses exclude variants that were associated with other intermediate phenotypes that are known to affect lung function? Were standard MR analyses protocols followed to rule out pleiotropy etc. I did not see all the standardized checklists for MR now mandated by many journals.

Author Rebuttal, first revision:

We are very thankful to the reviewers, editor and whole editorial team for their time and thoughtful suggestions. We have carefully addressed the reviewers' comments which we believe have substantially improved our report about this MAGIC collaborators effort for Random Glucose. In the below document we provide responses to the review comments point-by-point, using **bold blue text** to highlight the reviewer's question and plain black text for our response. We also provide citations of the introduced changes in *black italic text*, as well as report the location of changes in main text or supplementary material. We have highlighted the changes introduced to the text **in red font** within the marked copy and followed other instructions for the submission.

General:

Referee expertise:

Referee #1: Genetics, diabetes, complex traits

Referee #2: Genetics, cardiovascular diseases, clinical translation

Reviewers' Comments:

Reviewer #1:

Remarks to the Author:

This paper describes a large GWAS meta-analysis of random glucose in almost 500K samples from 17 studies, of which the UK Biobank was the largest group (~400K). The authors carry out a wide range of interesting analyses that add to our understanding of glucose genetics. The manuscript is well written, and easy to follow despite the large number of conducted analyses. The results are novel and original, and used contemporary methods for analysis. I have the following comments and suggestions for the authors' consideration, mainly concerning secondary signal, rare variant and directional pleiotropy analysis:

1. The authors very clearly distinguish lead loci from secondary loci (142 loci and 185 signals) in abstract and main results. What I did find confusing is how the number of novel SNPs for glycemic traits were counted (ST3). I have tried regenerate the number 84 novel SNPs number and was unable to, and noticed a couple of peculiarities to the table which are listed below. Foremost, it seems that novelty is noted for both primary loci and secondary signals. I would refrain from using secondary signals for novelty, because they were generated by an approximate method (GCTA on summary statistics) and not formal method (stepwise regression with individual level data). For example, it seems that 16 SNPs in G6PC2 locus seems are being counted 7 times as novel? Perhaps the authors can clarify what they exactly counted as being novel. Their current approach seems generous. GCTA is not a formal method but approximate method, and depending on the data set (and unfortunately even various parameters chosen within its own software) will identify different SNPs. Given that the UK Biobank is the predominant sample (400K of 500K participants), can the authors perform formal secondary signal analysis using stepwise linear regression, where they include the lead SNP as a covariate in the analysis, and then repeat until there are no SNPs left that exceed set significance? From personal experience, GCTA produces vastly different results from individual level conditional analysis. Notwithstanding, I would limit the counting of novel SNP based on the loci (not distinct signals).

We thank the Reviewer for the suggestion to validate our GCTA conditional analysis results (summary-level approximate approach) by performing an individual-level conditional analysis based

on UKBB genotypes. To carry out the direct conditional analysis using BOLT-LMM, we first conducted standard GWAS for UKBB by conditioning on the 120 primary signals from our meta-analysis, and then performed clumping analyses ($P \leq 1 \times 10^{-5}$, $r^2 = 0.01$, window-size=1Mb) to check whether the number of independent significant loci were consistent with the number of secondary loci for each primary locus (**Extended Data Table 3**). We repeated the GWAS by conditioning on both, the 120 primary signals and the clumped loci from the previous round of conditional analyses until there was no more significant locus for each primary locus.

We have added in **Extended Data Table 3** two columns (I-J) showing: (I) P values for additional signals (from approximate conditional analysis) from direct conditional analysis in UKBB; and (J) description whether signal is “confirmed”, “suggestive” or “not confirmed”. In the individual-level conditional analysis for UKBB we found that the number of independent loci was mostly consistent with the number of additional signals for each region, although the exact lead variant being reported might be different (**Extended Data Table 3**- columns I-J). This is in agreement with the statements from the Reviewer #1 and expected for the following reasons: 1) GCTA conditional analysis was based on the results from our meta-analysis and a subset of UKBB (as a reference panel for linkage disequilibrium estimates) to approximate the real conditional analysis; and 2) GCTA conditional analysis is not a fine-mapping method to define causal variants, but a method to detect the number of additional signals within a region, where such additional signals could indicate either multiple causal variants or imperfect tagging of genotyped variants for the causal variant within a region¹. The latter point also applies to the individual-level conditional analysis we have performed, so direct interpretation of the additional signals as causal variants should be avoided.

In addition to confirming the signals proposed in our meta-analysis, we have observed 54 additional secondary signals by carrying out direct conditional analysis on UKBB. We added these results to a new table (**Extended Data Table 4b**). We also added in the “RG meta-analyses” section of the methods a detailed description about novel loci and signals counting, which was missing from the previous version. We believe it would be useful for the readers: *“A locus was considered novel if it contained only novel RG signals. A signal was considered novel if there were no previously established signals for glycaemic traits² within +/- 400 kb of the RG signal in LD ($r^2 \geq 0.01$) with this signal (the look up for glycaemic traits was done on the 1st of June 2021). Additional signals were excluded from the novel loci count. However, these are annotated in **Extended Data Table 3**. All signals, including rare, cross-ancestry and sex-dimorphic, were included in the count and those significant in multiple analyses were counted only once.”*

After applying above definition, we reassigned several previously believed novel loci as established. Additionally, we have corrected any peculiarities in **Extended Data Table 3** (also mentioned below in minor changes). We updated the text with a new loci count: *“We discovered 137 RG loci represented by 178 distinct signals, including 14 with sex-dimorphic effects, 9 cross-ancestry and 17 low/rare frequency signals. Of these, 56 loci are novel for glycaemic traits.”*

In **Extended Data Table 3** we still provide novel status for the additional signals. However, it is easy to calculate the total number of novel primary signals by filtering by “Novel locus for glycaemic traits” column. In summary, the updated counts are: 74 novel signals and 56 novel loci for glycaemic traits. These numbers can also be obtained by filtering the corresponding columns in **Extended Data Table 3**.

We changed how information about previous associations with T2D is presented. We renamed “Novel for T2D” column to “Established for Type 2 Diabetes” and updated accordingly the “RG meta-analyses” section of the methods: *“A signal was considered established for T2D if a lead or additional signal was located within +/- 400 kb of a previously reported association”*.

2. Rare variant analysis: On line 248 the authors state that they used the squared correlation coefficient to identify rare variants as being independent ($r^2_{1000GenomesAllEthnicities} < 0.001$). The authors state that the lack of LD between SNPs was chosen to indicate independent signal. Similar to my issue with secondary signals, can the authors perform a conditional analysis and adjust for the lead SNP to empirically confirm independence of the rare variant signal? A low r^2 can still imply a high D' especially when evaluating rare variants where differences in minor allele frequencies contribute to a large reduction in the correlation coefficient with the usually common lead variant.

As suggested by the Reviewer, we adjusted the rare variant association analyses for the 120 common lead variants, similarly to the conditional analyses described above. These additional analyses confirmed our results on the independence of the rare variant signals from common variant signals for most of the SNPs we reported initially – out of the 22 only four (one exonic and two intronic variants within *G6PC2* and one intronic variant within *CAMK2B*, near *GCK*) became non-significant after this adjustment. We report the P -values of these direct conditional analyses on **Extended Data Table 3** - column I. We removed those four non-independent variants from **Extended Data Table 3**. Additionally, we removed one rare variant that is in complete LD with another one as the Reviewer pointed out in Minor point 11, leaving the total count of rare variants to be 17 instead of 22. We also added description to the Methods about this additional check, please see p.26:

“Out of 748 rare variants, associated with RG at genome-wide significance in the HRC-imputed UKBB data, 45 were available and 22 were validated by the UKBB WES data (Extended Data Table 7). Each of 45 variants were tested separately for their effect on RG by fitting the AST20 RG linear regression model including the first six principal components using the software PLINKv1.90. We considered a rare variant validated if it had the same direction of effect on RG and reached a nominal threshold of significance ($P \leq 0.05$). Furthermore, we tested the 22 validated variants for their independence from the 120 common lead variants in direct conditional analyses by using the 120 variants as covariates in the association analysis. Out of the 22 variants, 18 remained significant at both, regional ($P < 10^{-5}$) and genome-wide ($P < 5 \times 10^{-8}$) thresholds (Extended Data table 3). However, with additional checks on LD between the 17 variants, we identified that two were in complete LD in Africans. Amongst those two, the less significant was removed from the set of validated variants.”

3. The authors have performed a directional pleiotropy analysis, of which the z-scores are available in Supplementary Table 23 and Supplementary figure 6A (and not 5A as the authors refer to in the text). The hierarchical cluster plot is not interpretable, the rows and columns are unreadable, and the figure is more or less a yellow blob. I was looking forward to examining that figure, and if executed correctly would be a great main figure to the manuscript. My approach would be the following: While I think the z-scores a great input to denote snp-trait association strength, the extreme minima (e.g. z-score of -33 of GCKR with triglycerides) and maxima (z-score of 44 for TCF7L2 for T2D) currently force all other associations to look relatively ‘non-relevant’ (e.g. yellow, please reconsider to white color) because they bleak in comparison to the peak associations. Please consider using a cut-off for minima and maxima so that a more nuanced picture of directional pleiotropy may emerge. Limiting to z-score (to for example 5.45, corresponding to $P < 5.10^{-8}$) and categorizing the colors into dark red, red, light red, white, light blue, blue, dark blue (7 categories). Finally, some traits don’t seem to be that related to the lead SNPs with exception of an incidental association here or there. It might be an idea to remove them from the figure because they don’t contribute to the visualization and cluster assignment. For more customized hierarchical clustering in R, the authors could look into R package ‘pheatmap’ rather than using `hclust()`.

We thank the Reviewer for the suggestion to improve the interpretability of the heatmap plot for the directional pleiotropy analysis. We modified the methods section describing clustering and generated a new heatmap (below, **Extended Data Figure 6a** and separate pdf file with figure provided) that incorporates the Reviewer’s suggestions. According to Reviewer #1 suggestion, we limited z-score to 5.45, corresponding to $P < 5.10^{-8}$ genome-wide significance, and categorised the colours into dark red, red, light red, white, light blue, blue, dark blue (7 categories). We also agree with Reviewer #1 about limiting the visualised phenotypes to those affected by the lead RG-associated SNPs (with exception of an incidental association, as Reviewer commented). We removed such phenotypes from the figure and limited the set of visualised phenotypes contributing to the cluster assignment to 34. The new text in the “Clustering of the RG signals with results for 45 other phenotypes” section is now as follows:

*“We looked up the Z-scores (regression coefficient beta divided by the standard error) of the distinct 143 RG signals in publicly available summary statistics of 45 relevant phenotypes (**Extended Data Table 24**). All variant effects were aligned to the RG risk allele. HapMap2 based summary statistics were imputed using SS-Imp v0.5.565 to minimise missingness. Missing summary statistics values were imputed via mean imputation. The resulting variant-trait association matrix was truncated to 2 standard deviations to minimise the effect of outliers. We used agglomerative hierarchical clustering with Ward’s method to partition the variants into groups by their effects on the considered outcomes. The clustering analysis was performed in R using function hclust() from in-built stats package.”*

The legend for the figure is as follows: “The heatmap was produced using Pheatmap package. For visualisation, the z-scores were truncated to the value corresponding to genome-wide significance (5.45) and 11 of phenotypes with the lowest median absolute z-scores were excluded.”

Moreover, we would like to clarify that in many cases the mean imputation of missing values results in “pale” cells on heatmap.

This suggestion from the Reviewer #1 was very helpful in streamlining the visualisation of the evaluation of directional pleiotropy, thus supporting better the Main text highlights of the loci cluster effects, which remained unchanged.

Minor:

1. Line 201: Adjustment for last meal timing attenuated some effect estimates for 5 loci? Or is this just a selection? Then indicate the total number of loci that were affected. Does the adjustment for meal timing, increase effect estimates for some SNPs?

Please, refer to **Extended Data Figure 1b** where we highlight the five loci with heterogeneity P-value ≤ 0.05 (beta-coefficient plot) or Z-score >3 difference between the two models compared (Z-score plots). Full details are in the **Extended Data Figure 1b** legend. The same figure highlights that the adjustment for the time since last meal did not increase effect estimates at any of the RG signals.

2. Line 214: Can the authors specify if this is nominal significance ($P < 0.05$) without or without directional concordance.

We have now specified in the text that nominal associations in other ethnicities for signals, identified in Europeans, were also directionally concordant but one signal, as per below citation: “A number of signals identified in Europeans showed nominal significance ($P < 0.05$) in other ethnicities, including novel *USP47* in the African, *FAM46C* and *ACVR1C* in Indian, *TRIM59/KPNA4* and *ZC3H13* in Chinese (**Extended Data Table 3**). All such signals, but *rs540524* at *G6PC2*, were directionally concordant across ethnicities. At *GCK*, *rs2908286* ($r^2_{1000GenomesAllEthnicities} = 0.83$ with *rs2971670* lead in Europeans) was genome-wide significant in the African descent individuals alone (**Extended Data Table 5**).”

3. Line 233: male-specific and female-specific instead of men-specific and women-specific

We made the suggested change in the text: “We first discovered sex-dimorphism at 14 RG loci, including male-specific *PRDM16*, *RSPO3*, and female-specific *SGIP1*, *SRRM3*, and *SLC43A2* (**Methods, Table 1, Figure 1a, Extended Data Table 3**).” Additionally, we checked the text and amended the references to the sex using the terms suggested by the Reviewer #1.

4. Out of 748 rare variants associated with RG above genome-wide significance in the HRC-imputed UKBB data. Is this genome-wide?

Yes, rare-variant analysis was carried out across the genome. We realise the description about rare variant analysis was not sufficient in the Methods section and have updated it on p.22:

“For the GWAS of the UKBB data we excluded non-white non-European individuals and those with discrepancies in genotyped and reported sex. For the RG definition, we used the same criteria as in the other studies described above. To control for population structure, we adjusted the analyses for six first principal components. The GWAS was performed using the BOLT-LMM v2.3 software restricting the

analyses to variants with $MAF > 1\%$ and imputation quality > 0.4 . Additionally, we performed rare variant analysis across the genome on UKBB data for variants with $MAF < 1\%$ and imputation quality > 0.4 .”

5. AS20+AST20 – not directly clear from the description what exactly has been done here?

We have rewritten the description of AS20+AST20 meta-analysis to make this part of the methods clearer: “We have also performed meta-analysis using cohorts with time from last meal available (AST20 model, 12 cohorts) combined with those lacking this information (AS20, 5 cohorts) to maximise the association power while taking into account T . We termed this analysis as AS20+AST20 in the following text (17 cohorts, $N_{max}=480,250$).”

6. Supp Table 2: can the authors indicate which of the models were the final AS20, AST20, AS11, AST11, ASTB11, ASTB20

We would like to clarify about the models presented in **Extended Data Table 2. Evaluation of the effects of different sets of covariates and transformations for RG in the initial set of six available cohorts.**

They present ten different models fitted in the six cohorts, available to us initially for the modelling of RG, i.e. to identify the relevant set of covariates as well as the necessary transformation for RG to be used across all the datasets in the GWAS meta-analysis. We fitted the models using untransformed (models 1-5) and natural log-transformed (models 6-10) RG as the outcome without applying any specific trait cut-off. Hence, these models are not related to the final models termed as AS20, AST20, AS11, AST11, ASTB11, ASTB20 in the manuscript. These models have only guided the selection of covariates and RG transformation to be used in the GWAS analyses. In parallel, we checked the distribution of RG in these cohorts and decided on two cut-offs for subsequent analyses: < 20 mmol/l to account for the effect of extreme RG values observed in some of these six cohorts and < 11.1 mmol/l (11), which is an established threshold for T2D diagnosis. Therefore, the six models used in GWAS analyses were constructed to accommodate our observations regarding both best predictors and RG distribution. As such, the GWAS models do not directly correspond to those presented in ST2. We changed **Extended Data Table 2** title: “*Extended Data Table 2. Evaluation of the effects of different sets of covariates and transformations for RG in the initial set of six available cohorts.*”

7. Supp Table 3: either separate out the independent from the conditionally independent variants, or reconsider limiting novelty assignment to primary SNPs.

Please refer to our answer to question 1, where we describe the new columns added to **Extended Data Table 3**, showing which secondary signals from approximate conditional analysis were confirmed by individual-level conditional analysis in UKBB. We believe the readers can easily filter the corresponding column for lead and additional signals in this table and explore additional UKBB secondary signals presented in **Extended Data Table 4b**.

8. rs538547285, in G6PC2 locus, unless independent from lead locus by formal conditional analysis (not approximate), it should be part of locus 14 and not locus 122.

The reviewer is correct. Please refer to our answer to question 2. Since this rare variant signal was not independent from the common variant signals as demonstrated by this direct conditional analysis, we have removed it completely from **Extended Data Table 3**.

9. Duplication of rs2800734, and other sex-dimorphic SNPs. Instead of a duplicate SNP entry, maybe add a flag for sex-dimorphism?

We appreciate this suggestion and have removed all duplicate rows from **Extended Data Table 3**. Instead, we have added two new columns to indicate cross-ancestry and sex-dimorphic signals (columns K-L).

10. rs74832478 entry is confusing. (primary locus 115 and 140, counted twice)

We would like to thank the reviewer for spotting this double counting for rs74832478. We corrected this entry in **Extended Data Table 3**.

11. rs147493041 and rs149594447 are listed as locus 126 and locus 127, whereas they are in complete LD in all ancestry 1000 G all ancestries

We thank the Reviewer for this correction. Our rare variant analysis and clumping were done in Europeans, where the LD between these two variants is 0. After additional checks, we note that in Africans there is indeed one individual within the 1000G data that shares the minor allele at these two variants. Therefore, we removed the less significant variant from **Extended Data Table 3**. We also added corresponding note in the Methods. The full description is given in our answer to question number 2.

12. rs543933126 locus 128 could be argues its part of locus 43 based on distance.

The reviewer is correct that once we conditioned all the rare variant association analyses on the common lead variants, the effect of rs543933126 became non-significant, specifically given the effect at *GCK* locus (locus 53). Since this rare variant signal was not independent from the common variant signals as demonstrated by this direct conditional analysis, we have removed it completely from **Extended Data Table 3**.

13. ST7: orient towards the minor allele, instead of having EAF of 99.999%. Also, adjust for the lead SNP in order to find out whether the rare allele signal is independent from the lead signal.

We report the effect sizes for the minor allele as suggested by Reviewer. Please see our response to Question 1 about the direct conditional analysis.

14. ST8 and ST9: r2 with lead SNP is always 1, can the authors double-check the veracity of that column.

We corrected the columns showing r2 between lead and credible variants in **Extended Data Table 8** and **Extended Data Table 9**. We are now reporting r2 between these set of variants in EUR (**Extended Data Table 8**) and additional ethnicities (**Extended Data Table 9**) based on 1000 Genomes Phase 3.

15. ST15a: column name: RG GWAS significant? What does this mean?

ST15a includes all significant associations from MetaXcan across all tested tissues. The column "RG GWAS significant" in ST15a captures which of these associations are also genome-wide significant in the RG meta-analysis. For better interpretability, we have renamed the column into "GW-significant in RG M-A".

16. ST16: can the authors specify in which tissue the colocalization result is referring to?

We have now specified in the title of ST16 the tissues of colocalization: "*Extended Data Table 16. Results from the eQTLGen whole blood and TIGER human pancreatic islets co-localization analysis of the RG signals.*"

Reviewer #2:
Remarks to the Author:

Random glucose GWAS in 493,036 individuals provides insights into diabetes pathophysiology, complications and treatment stratification. Lagou et al

Summary of findings

Random glucose 493K (only 16K non-European), non-diabetic
142 loci, 185 signals
14 sex dimorphic effects
9 through trans ethnic
25 rare variants
underappreciated role of gut
lung function modulated by glucose

Nice to read this paper from a very good group and PI who have been leaders in the field. Overall, I liked the paper but felt it could be improved organizationally and honed down to key messages. It seems like there were many, many different analyses performed and there was insufficient detail on some things (for instance functional assays) and insufficient justification and analyses for others (microbiome). I also had some specific questions.

Questions

1. Are the glucose levels really random? Is there a plot of the time of day the glucose labs were drawn? Suspect these are almost all between 8am-5pm. They did have info on time from last meal on some which is important. However, did time from last meal systematically differ by sex or race/ethnicity? This could be a confounder and could reflect occupation or other socio-economic factors rather than underlying biology.

We would like the reviewer to refer to **Extended Data Table 1** where we report mean (sd) of time since last meal for men and women (when available) in each participating study. To provide a visual aspect of these values for comparison purposes, we have generated the following plots.

In the figure above, the first plot shows the means and standard deviations between sexes across all participating studies. These measures overlap and span over a large time since last meal period to blood sampling from each study. This plot is representative of what is usually referred to as random glucose. We additionally clarify that ethnicity was self-reported and within contributing studies these measures were not taken as part of OGTT tests – this is confirmed by standard deviations provided. In such sample collections, the participants report the time passed since they had last eaten, while the time of blood sampling is typically not available for analysis. The density plots

generated within UKBB data demonstrate no differences in the distribution of time since last meal, when comparing different ethnicities and when additionally stratifying by sex. As discussed in our responses to Reviewer #1 questions, we performed detailed modelling of RG phenotype and compared models with and without adjustment for the time since last meal. We report the differences in genetic effects on RG between models in the main text. These differences are represented only by a small number of loci having a change in the allelic effect size estimates. Most of these loci do have a biological explanation for such a change in effect estimates. Additionally, we would like to bring attention of the Reviewer to analysis of genetic correlations between RG and other phenotypes, especially glycaemic traits. Given the estimates of genetic correlation (Figure 4), we are confident that the modelling of RG, proposed in our manuscript, takes into account the necessary covariates, while losing estimated ~5% in power, when the time since last meal is not recorded. We appreciate, there might be additional confounder of blood sampling time. We are looking forward to more detailed explorations of this phenotype in the future in even larger studies with potential confounders evaluated, when such data would become available. However, for this manuscript, our task was to demonstrate that RG is an important and valid phenotype for investigation in GWAS meta-analyses.

2. How do you deal with correlations between diet and glucose given confounders? For example, people that tend to eat dense caloric diets especially if they eat more frequently are likely to have higher glucose levels but also higher obesity rates and may be more sedentary. This is not necessarily bad, but certainly is a limitation or caveat.

The trajectories of glucose levels after the meal is an interesting and yet not well-explored subject within the GWAS. One important factor is that in this analysis we included individuals without known diabetes, where obesity is an important risk factor. Therefore, in this study analyses, some individuals, on the trajectory of developing diabetes, while having a certain BMI, could have been included. Our analyses, comparing models with and without BMI adjustment have shown no difference between the effect estimates for the associated loci, in line with previous studies of fasting glucose from MAGIC.

Major efforts have been put to describe fasting glycaemic trait variability in the past ~15 years. Both within the MAGIC, which we are representing, and within various large-scale biobanks, GWAS meta-analyses were published on phenotypes, related to OGTT measures and various additional standard glycaemic trait measures, including intravenous tests. These detailed measures of glucose metabolism efforts demonstrated that, depending on time after glucose intake, there is a well-defined and timed biological sequence of biological processes, maintaining glucose homeostasis. However, the sample sizes, required to identify genome-wide significant associations for such traits are very large, and many previous GWAS efforts have highlighted only dozens of loci for more detailed measures of glucose metabolism, frequently overlapping between related biological processes, such as insulin secretion or action. The meal composition is a complex factor to account for within GWAS, and within subgroups of a sample. Such stratification requires sufficient power for locus discovery. With smaller sample sizes within subgroups (less than 40,000 individuals) it is a power-limiting exercise to either adjust for many confounders or perform a stratified analysis, to account for diet composition. With the success stories from large biobanks and record linkage, in the near future, we will be able to model the food composition and sedentary/active lifestyle better for well-powered studies of glucose levels variability in individuals without diabetes. Such investigations would *implement* multi-phenotype and high-dimensional analysis approaches to disentangle the primary genetic effects, pleiotropy and other more complex genetic relationships.

To accommodate the reviewer's comment, we included the following phrase in the Discussion section: *"In the near future, larger well-phenotyped datasets will enable high-dimensional GWAS*

investigations, disentangling the role of diet composition, physical activity or lifestyle on RG level variability in relation to genetic effects.”

3. It can be hard to link loci to genes (ITPR3, RREB1 are examples of loci highlighted in one section of the paper etc) speculatively just based on distance to nearest gene. As the authors hint at (in the functional section) colocalization can be useful in narrowing down the list of causal genes. For all the RG non-coding variants identified, is there colocalization data to support a causal relationship for genes within associated loci (based on eqtl or sqtl RNASeq data from pancreas, gut, fat, brain etc)? The methods section on eQTL seems to indicate that only eQTL data from whole blood was used which seems suboptimal given other publicly available data on metabolically relevant tissues. See recent papers from Mark McCarthy, Anna Gloyn and Stephen Montgomery for instance.

We followed the same strategy for all tissues and further performed colocalization analyses using human pancreatic islet eQTLs provided by TIGER (up to 514 individuals)³. We have identified 19 loci with strong links (posterior probability H4, (i.e. both traits are associated and share the same single causal variant) to pancreatic tissue gene expression data. Apart from the shared loci with eQTLGen (e.g. *MADD*, *PPDPF*), we have also defined additional novel loci which are more closely linked to pancreas, like the *SIX2* and *SIX3* genes. We added these new eQTL analysis results to the existing **Extended Data Table 16** and updated the colocalization section of the methods: “*We further performed co-localization analysis using whole blood gene expression-QTL (eQTL) data provided by eQTLGen and human pancreatic islets eQTLs provided by TIGER along with AS20+AST20 meta-analysis results.*” Further, we added new wording to the main text: “*Similar analyses of pancreatic islets regulatory variation in the TIGER dataset defined 19 loci with strong statistical support for colocalization of the effects on RG and tissue expression of *ADCY5*, *RNF6*, *SIX2*, *SIX3/SIX3-AS1* and *STARD10*, in addition to *MADD* and *PPDPF*, with latter overlapping in whole blood.*”

4. Functional and structural characterisation was also done for several genes including *GLP1R* which was interesting data. However, the methods for this were not even introduced on p12 making it hard to follow (“our functional approach”)...I was left wondering what was the system used for these experiments? Would be best at least introduced here rather than exclusively in the methods. The methods section for these is also a bit hard to follow in terms of critical items. It indicates that both transient or stable cell lines were used but not why this was the case or for which experiments which stable vs transient lines were used. For the mini-g recruitment assays it is not clear what kind of cells were used in the stable lines (lines 604-604). Were these also HEK cells? Same is true for lines 621-622. I also think this Results section, while interesting, does not naturally flow in terms of how it is organized in the paper. I understand the desire to connect these analyses to a more clinically relevant issue and *GLP1R* targeting drugs are really “hot” right now. But it seemed out of place in how it was introduced and presented.

We rewrote this part of the manuscript (under “Functional and structural characterisation of RG-associated *GLP1R* coding variants provides a framework for T2D treatment stratification”; p12) to provide essential details of the approach. This should give the reader more information about the system, implemented for the assays and used to functionally evaluate *GLP-1R* variants. Full details are in the Methods section. Nomenclature for *GLP1R* amino acid residue numbering is also explained in the revised manuscript. We also expanded on specific points, e.g., explaining the rationale for testing *GLP-1R* endocytosis, which is related to the pleiotropic behaviours of this receptor and biased agonism, rather than just G protein responses. The reviewer asked about the reasons for doing some experiments in stable cell lines and others using transient transfection. Most of the *GLP1R* functional data was generated using full concentration responses which warranted developing stable cell lines

to ensure we had enough cellular material for these assays (most of Figure 2). We did also want to give a snapshot of the broader impact of *GLP1R* coding variants on functional response, but as many of these variants are very rare it did not merit the extra work of generating stable cell lines for what is essentially a screen at a single agonist concentration (Figure 2j). We have adjusted the main text Methods section appropriately. Due to the large volume these changes are not shown here, please see the respective parts of the manuscript.

5. The microbiome associations p 15 seem almost an afterthought and are not well developed. Pretty speculative. If more is not done, would remove.

We expanded the description of microbiome results in the “Functional annotation of RG associations and intestinal health” section:

“Prompted by multiple analyses highlighting a potential role for the digestive tract in glucose regulation, we assessed the overlap between our signals and those from the latest gut microbiome GWAS⁴ (Methods) and identified three genera sharing signals and direction of effect with RG at two loci: Collinsella and LachnospiraceaeFCS020 at ABO-FUT2 and Slackia at G6PC2 (Figure 1a, Extended Data Table 18). The ABO-FUT2 locus effects on RG could be mediated by abundance of Collinsella/LachnospiraceaeFCS02, producing glucose from lactose and galactose⁵. Collinsella genus affects gut permeability IL-17A⁶ and shows higher abundance in individuals with T2D compared to those with normal glucose tolerance and individuals with pre-diabetes^{7,8}. Moreover, weight loss decreases Collinsella among obese individuals with T2D⁹. Higher prevalence of Lachnospiraceae family is associated with metabolic disorders, while genus LachnospiraceaeFCS02 abundance shows inverse correlation with serum triglycerides¹⁰. Slackia abundance is increased in individuals with T2D as well⁷. However, the mechanism of their enrichment is yet to be studied. This multi-omic annotation provided strong evidence about the RG links to intestinal health.”

6. The relationship of RG related traits with other traits (cancer, lung function) are interesting. Phenotypes associated with RG (like obesity) are obvious candidates to be associated with lung function. Things that are associated with lung function could obviously impact RG through modification of exercise (people with lower FEV1 may choose to exercise less and therefore get higher RG). In this way the MR analyses are important. Did the RG variants used in the MR analyses exclude variants that were associated with other intermediate phenotypes that are known to affect lung function? Were standard MR analyses protocols followed to rule out pleiotropy etc. I did not see all the standardized checklists for MR now mandated by many journals.

We tested the association of RG and lung function for causality in both directions, where we found no evidence for a causal effect in the reverse direction (from lung function to RG) despite comparable instrument strength/power. We state in the main text “suggested a causal effect of RG and T2D on lung function, including FEV1 ($\beta_{MR-RG}=-0.60$, $P=0.0015$; $\beta_{MR-T2D}=-0.049$, $P=1.27 \times 10^{-13}$) and FVC ($\beta_{MR-RG}=-0.61$, $P=3.5 \times 10^{-4}$; $\beta_{MR-T2D}=-0.062$, $P=1.42 \times 10^{-21}$), **but not vice versa** (RG $\beta_{MR-FEV1}=-0.0048$, $P=0.42$; $\beta_{MR-FVC}=-0.01$, $P=0.17$ and T2D ($\beta_{MR-FEV1}=-0.18$, $P=0.040$; $\beta_{MR-FVC}=-0.21$, $P=0.040$)” and refer to the table of full Mendelian Randomisation results. We believe the lack of causal association does not warrant the testing of further causal hypotheses, such as the effect of lung function on RG via exercise or other intermediate phenotypes (still, we provide results of such testing below). Even though pleiotropy cannot be fully ruled out, we report in the MR results table (Extended Data Table 22) the MR-Egger intercept which provides a measure of bias from pleiotropy. An Egger intercept term that differs from zero is indicative of overall directional pleiotropy. This intercept term does not differ from zero in either RG-FVC or RG-FEV1 MR analyses. The scatter plots below illustrate the RG-FEV1, RG-FVC, T2D-FEV1 and T2D-FVC MR analyses. On upper panel (RG plots), SNPs with exposure

(RG) $\beta > 0.008$ are labelled with nearest gene name. On lower panel (T2D plots), SNPs in LD with a coding variant as annotated by Vujkovic et al.¹¹ are labelled with gene names.

We also expanded the description of the MR-Egger analysis in the methods section: *“The MR-Egger causal estimate is valid as long as the pleiotropic effects of the instruments are independently distributed from their genetic associations with the risk factor. If the MR-Egger intercept term is close to zero, then the MR-Egger causal estimate will be close to the IVW estimate. However, even if the two causal estimates are similar, inferences from the two methods can differ if the MR-Egger estimate is imprecise. In such cases, the MR-Egger method provides no additional evidence for a causal effect, but it does not contradict evidence for a causal effect from a conventional (in this case IVW) MR analysis either. Causal effect estimates of RG on lung function are given in units of standard deviation per mmol/L (natural log transformed) RG increase. Similarly, causal effect estimates of lung function measures on RG are expressed in natural log transformed mmol/L RG per standard deviation increase in lung function. We used the STROBE-MR reporting guideline for MR studies to facilitate the readers` evaluation of our results.”*

Furthermore, we provide in **Extended Data Table 21** the list of variants used as instruments in the aforementioned MR analyses, and we included in the submission a file with the STROBE-MR report as mentioned above.

Finally, we explored the possibility that RG exerts its causal effect on lung function via exercise by fitting a multivariable Mendelian Randomisation (MVMR, using *MVMR* package⁴²), model with RG and leisure screen time (LST, a strong genetic instrument proxy for physical activity) as exposure. Moreover, we tested whether smoking - a factor known to reduce lung function - could confound the causal association between glucose levels and lung function. We applied MVMR, estimating the

direct causal effects of smoking and glucose in an inverse-variance weighted MVMR model. Fifty-four independent genome-wide significant variants were used as proxies for cigarettes per day, CPD¹³, 66 variants for LST¹⁴, and 139 variants as proxies for RG in the combined model (**Extended Data Table 22b**, also shown below). For smoking, both RG and cigarettes per day are causally linked to FVC ($\beta_{MR-RG}=-0.64$, $P=2.19 \times 10^{-4}$; $\beta_{MR-Cigarette}=-0.083$, $P=2.39 \times 10^{-9}$) and FEV1 ($\beta_{MR-RG}=-0.64$, $P=5.70 \times 10^{-4}$; $\beta_{MR-Cigarette}=-0.098$, $P=8.10 \times 10^{-11}$), thus demonstrating that causal effect of RG on lung function phenotypes is independent of the smoked cigarettes per day effect on the same phenotypes. Similarly, both RG and LST are causally related to FVC. We observed an analogous set of causal effects for T2D and FVC relationship. Overall, these multivariable MR analyses indicate that RG and T2D might cause deterioration of lung function independent of physical activity and smoking, while the latter two also show causal effects independent of RG and T2D on lung function, as would be expected.

We added the following information to the main text: *“External factors, such as smoking or sedentary lifestyle, could cause lung function to decline, independent of RG and T2D effects. We implemented multivariable MR (Methods) and show (Extended Data Table 22b) that RG and T2D causal effects on FVC are independent of both, cigarettes smoked per day, i.e., proxy for smoking¹³ and leisure screen time, i.e., proxy for physical activity¹⁴.”*

Extended Data Table 22b. Multivariable Mendelian Randomisation analyses assessing causal effects of RG and T2D on lung function via exercise or smoking.

Outcome	Exposures	N SNPs	MVMR beta	MVMR SE	MVMR P-value
FVC	RG	136	-0.6426	0.1705	0.000219
	CPD	54	-0.0834	0.0133	2.39×10^{-9}
FVC	RG	135	-0.5178	0.1583	0.00126
	LST	66	-0.0955	0.0281	0.00080
FVC	T2D	421	-0.0543	0.0085	4.25×10^{-10}
	CPD	54	-0.0867	0.0143	2.88×10^{-9}
FVC	T2D	421	-0.0538	0.0085	5.74×10^{-10}
	LST	65	-0.0846	0.0292	0.00395

We added the description of MVMR analyses to the **Methods** section: *“Multivariable MR (MVMR) is an extension of MR that can be applied with either individual or summary level data to estimate the effect of multiple, potentially related, exposures on an outcome¹². We used the MVMR (v0.3) R package to test whether the causal effects of RG and T2D on forced vital capacity (FVC) are independent of possible confounders, such as physical activity and smoking. The same instrument selection criteria as described for the main MR analysis were used. Cigarettes smoked per day (CPD) was instrumented by 54 (available out of the 58 in total) independent genome-wide significant variants, obtained from the genome-wide association study of Liu et al¹³. Leisure screen time (LST) served as a continuous proxy phenotype for physical activity from the recent study of Wang et al. with 66 (available out of the 88 in total) independent genome-wide significant variants¹⁴.”*

References

1. Yang, J. et al. Conditional and joint multiple-SNP analysis of GWAS summary statistics identifies additional variants influencing complex traits. *Nat Genet* **44**, 369-75, S1-3 (2012).

2. Chen, J. *et al.* The trans-ancestral genomic architecture of glyceic traits. *Nat Genet* **53**, 840-860 (2021).
3. Alonso, L. *et al.* TIGER: The gene expression regulatory variation landscape of human pancreatic islets. *Cell Rep* **37**, 109807 (2021).
4. Kurilshikov, A. *et al.* Genetics of human gut microbiome composition. *bioRxiv*, 2020.06.26.173724 (2020).
5. Lopera-Maya, E.A. *et al.* Effect of host genetics on the gut microbiome in 7,738 participants of the Dutch Microbiome Project. *bioRxiv*, 2020.12.09.417642 (2020).
6. Carmichael, A.J., Arroyo, C.M. & Cockerham, L.G. Reaction of disodium cromoglycate with hydrated electrons. *Free Radic Biol Med* **4**, 215-8 (1988).
7. Human Gut Microbiome Atlas available from <https://www.microbiomeatlas.org>.
8. Zhang, X. *et al.* Human gut microbiota changes reveal the progression of glucose intolerance. *PLoS One* **8**, e71108 (2013).
9. Frost, F. *et al.* A structured weight loss program increases gut microbiota phylogenetic diversity and reduces levels of *Collinsella* in obese type 2 diabetics: A pilot study. *PLoS One* **14**, e0219489 (2019).
10. Vojinovic, D. *et al.* Relationship between gut microbiota and circulating metabolites in population-based cohorts. *Nat Commun* **10**, 5813 (2019).
11. Vujkovic, M. *et al.* Discovery of 318 new risk loci for type 2 diabetes and related vascular outcomes among 1.4 million participants in a multi-ancestry meta-analysis. *Nat Genet* **52**, 680-691 (2020).
12. Sanderson, E., Spiller, W. & Bowden, J. Testing and correcting for weak and pleiotropic instruments in two-sample multivariable Mendelian randomization. *Stat Med* **40**, 5434-5452 (2021).
13. Liu, M. *et al.* Association studies of up to 1.2 million individuals yield new insights into the genetic etiology of tobacco and alcohol use. *Nat Genet* **51**, 237-244 (2019).
14. Hoed, M. *et al.* *Physical activity and sedentary behavior; mechanistic insights and role in disease prevention*, (2021).

Decision Letter, second revision:

11th August 2022

Dear Inga,

Your revised manuscript entitled "Random glucose GWAS in 493,036 individuals provides insights into diabetes pathophysiology, complications and treatment stratification" (NG-A58434R2) has been seen by the original referees. As you will see from their comments below, they find that the paper has improved in revision, and therefore we will be happy in principle to publish it in Nature Genetics as an Article pending final revisions to address Reviewer #2's remaining concerns and to comply with our editorial and formatting guidelines.

We are now performing detailed checks on your paper and we will send you a checklist detailing our editorial and formatting requirements soon. Please do not upload the final materials and make any revisions until you receive this additional information from us.

Thank you again for your interest in Nature Genetics. Please do not hesitate to contact me if you have any questions.

Sincerely,
Kyle

Reviewer #1 (Remarks to the Author):

The authors have done an amazing job at addressing my comments; I am impressed with the level of detail and clarification that they have provided and can't think of any open standing issues.

Reviewer #2 (Remarks to the Author):

The Authors have responded to many of my queries. It is improved.

The paper remains extremely complex with many different messages. It undoubtedly reflects a multidisciplinary team's effort over many years which is commendable. Nevertheless, I find the complexity of the paper and the many different messages remains challenging.

As a crude example, the Discussion is < 1 double spaced page and the methods are close to 20 pages. This is not overly friendly to a non-expert general audience. The Authors really should do a better job in the Discussion of helping the reader to understand context for how important this work and prior literature and how this is a great advance.

In terms of specific issues, I think the potential interactions of the RG loci with the microbiome remains the weakest part of the paper, remaining somewhat speculative as written in the Results.

Author Response, second revision:

We are very thankful to the reviewers, editor, and whole editorial team for their time, valuable comments, and thoughtful suggestions. In the below document, we respond to the review comments point-by-point, using **bold blue text** to highlight the reviewer's question and plain black text for our response.

Reviewer #1:

The authors have done an amazing job at addressing my comments; I am impressed with the level of detail and clarification that they have provided and can't think of any open standing issues.

Answer: We would like to thank the reviewer for the valuable comments and suggestions that led to the improvement of the manuscript.

Reviewer #2:

The Authors have responded to many of my queries. It is improved.

Answer: We would like to thank the reviewer for helping us to improve the manuscript with excellent comments and suggestions.

The paper remains extremely complex with many different messages. It undoubtedly reflects a multidisciplinary team's effort over many years which is commendable. Nevertheless, I find the complexity of the paper and the many different messages remains challenging.

Answer: We are very thankful to the reviewer for their comment and appreciate the complexity of the proposed manuscript. The authorship of this manuscript felt that the quality of *Nature Genetics* papers must be high, and the readership audience is experienced and deeply specialised. It is our strong belief that the manuscript is wrapped up around an important subject, where the information is presented coherently, and conducted experiments and analyses unwrap logically, representing combined endeavour. We do believe that splitting this manuscript would have led to less impactful messages, with each separate part lacking the synergetic effect. Therefore, we are convinced that this manuscript is well presenting itself as a single piece of work.

As a crude example, the Discussion is < 1 double spaced page and the methods are close to 20 pages. This is not overly friendly to a non-expert general audience. The Authors really should do a better job in the Discussion of helping the reader to understand context for how important this work and prior literature and has this is a great advance.

Answer: This work is indeed a multidisciplinary research effort aiming to achieve triangulation and this explains the size of methods. We have now moved almost half of the methods to Supplementary Note, keeping only the important information in the main text, additionally following the journal's recommendation.

In terms of specific issues, I think the potential interactions of the RG loci with the microbiome remains the weakest part of the paper, remaining somewhat speculative as written in the Results.

Answer: We appreciate the question about the potential interaction between the RG loci and gut microbiome composition. However, we would like to highlight that nowhere in the manuscript we claimed such an interaction. The manuscript states specifically that we evaluated the enrichment (lines 397-411) in DNA associations with gut microbiome species abundancies. We have not claimed the interaction as such since the underlying mechanisms of it would have required a different type of statistical and possibly not only in silico evaluation. The description of the observed enrichment analyses attains the rigorous statistical threshold clearly stated by us in the manuscript and follows a standard methodology, including correction for multiple testing. We described the observed results and suggested the biological/pathophysiological rationale for such findings to the best of our knowledge and scientific literature available.

Prompted by the suggestion from the reviewer about a potential deeper investigation of the RG levels over the abundance of microbial species in the human gut in individuals from the general population, we performed Mendelian randomisation analyses to investigate the causal effect of RG levels on the abundance of three microbial species in the gut. While we cannot exclude false negative results due to the relatively low power of microbial species GWAS summary statistics (reference 42, main text), we detected no significant causal effect of higher RG levels on the abundance of three genera, including *Collinsella*, *LachnospiraceaeFCS02*, and *Slackia* (Table below).

Table. Two-sample Mendelian randomisation analyses assessing causal effects of RG on abundance of three microbial species in the human gut.

Exposure	Outcome	MR method	N SNPs	Beta	SE	P value
genus Collinsella	RG	MR-Egger	89	-0.090	0.63	0.89
genus Collinsella	RG	Weighted median	89	-0.99	0.64	0.12
genus Collinsella	RG	Inverse variance weighted	89	-0.020	0.41	0.96
genus Collinsella	RG	Simple mode	89	-0.79	1.40	0.58
genus Collinsella	RG	Weighted mode	89	-0.65	0.58	0.27
genus LachnospiraceaeFCS02	RG	MR Egger	88	-0.34	0.64	0.60
genus LachnospiraceaeFCS02	RG	Weighted median	88	-0.062	0.68	0.93
genus LachnospiraceaeFCS02	RG	Inverse variance weighted	88	-0.18	0.42	0.67
genus LachnospiraceaeFCS02	RG	Simple mode	88	-1.74	1.35	0.20
genus LachnospiraceaeFCS02	RG	Weighted mode	88	-0.29	0.56	0.61
genus Slackia	RG	MR-Egger	87	0.58	1.04	0.58
genus Slackia	RG	Weighted median	87	0.57	0.94	0.54
genus Slackia	RG	Inverse variance weighted	87	0.98	0.68	0.15
genus Slackia	RG	Simple mode	87	-2.72	1.97	0.17
genus Slackia	RG	Weighted mode	87	0.75	0.87	0.39

Final Decision Letter:

27th June 2023

Dear Inga,

I am delighted to say that your manuscript "Genome-wide association analyses of random glucose levels in 476,326 individuals provide insights into diabetes pathophysiology, complications and treatment stratification" has been accepted for publication in an upcoming issue of Nature Genetics.

Your paper will be published online after we receive your corrections and will appear in print in the next available issue. You can find out your date of online publication by contacting the Nature Press Office (press@nature.com) after sending your e-proof corrections. Now is the time to inform your Public Relations or Press Office about your paper, as they might be interested in promoting its publication. This will allow them time to prepare an accurate and satisfactory press release. Include your manuscript tracking number (NG-A58434R3) and the name of the journal, which they will need when they contact our Press Office.

Before your paper is published online, we will be distributing a press release to news organizations worldwide, which may very well include details of your work. We are happy for your institution or funding agency to prepare its own press release, but it must mention the embargo date and Nature Genetics. Our Press Office may contact you closer to the time of publication, but if you or your Press Office have any enquiries in the meantime, please contact press@nature.com.

Please note that Nature Genetics is a Transformative Journal (TJ). Authors may publish their research with us through the traditional subscription access route or make their paper immediately open access through payment of an article-processing charge (APC). Authors will not be required to make a final decision about access to their article until it has been accepted. [Find out more about Transformative Journals](https://www.springernature.com/gp/open-research/transformative-journals)

Authors may need to take specific actions to achieve [compliance](https://www.springernature.com/gp/open-research/funding/policy-compliance-faqs) with funder and institutional open access mandates. If your research is supported by a funder that requires immediate open access (e.g. according to [Plan S principles](https://www.springernature.com/gp/open-research/plan-s-compliance)), then you should select the gold OA route, and we will direct you to the compliant route where possible. For authors selecting the subscription publication route, the journal's standard licensing terms will need to be accepted, including [self-archiving and license to publish](https://www.nature.com/nature-portfolio/editorial-policies/self-archiving-and-license-to-publish). Those licensing terms will supersede any other terms that the author or any third party may assert apply to any version of the manuscript.

Please note that Nature Portfolio offers an immediate open access option only for papers that were first submitted after 1 January 2021.

If you have not already done so, we invite you to upload the step-by-step protocols used in this

manuscript to the Protocols Exchange, part of our on-line web resource, natureprotocols.com. If you complete the upload by the time you receive your manuscript proofs, we can insert links in your article that lead directly to the protocol details. Your protocol will be made freely available upon publication of your paper. By participating in natureprotocols.com, you are enabling researchers to more readily reproduce or adapt the methodology you use. Natureprotocols.com is fully searchable, providing your protocols and paper with increased utility and visibility. Please submit your protocol to <https://protocolexchange.researchsquare.com/>. After entering your nature.com username and password you will need to enter your manuscript number (NG-A58434R3). Further information can be found at <https://www.nature.com/nature-portfolio/editorial-policies/reporting-standards#protocols>

Sincerely,
Kyle

Kyle Vogan, PhD
Senior Editor
Nature Genetics
<https://orcid.org/0000-0001-9565-9665>